   

# Cystinosin regulates Na⁺/H⁺ exchanger 3 trafficking and function in kidney proximal tubular cells

Veenita Khare [1], Jean-Claude Farré[2], Mouad Ait Kbaich [3], Céline J Rocca [1], Cynthia Tang[1], Xuan Ma [1], Kavya Biederman [1], Ioli Mathur [1], Rafael A Badell-Grau [1], Anusha Sivakumar[1], Rola Chen [1], Sergio D Catz[3] & Stephanie Cherqui [1✉]

## Abstract

Cystinosis is a systemic lysosomal storage disease resulting from mutations in the *CTNS* gene encoding the lysosomal cystine transporter cystinosin, leading to cystine accumulation in all organs. Despite cystinosin's ubiquitous expression, renal Fanconi syndrome (FS) is the first clinical manifestation of cystinosis, which is not prevented by cystine reduction therapy with cysteamine. Here, we report a novel interaction of cystinosin and sodium/hydrogen (Na⁺/H⁺) exchanger proteins in the endosomes of yeast and mammalian cells. NHE3 is a major absorptive sodium transporter at the apical membrane of proximal tubular cells (PTCs). Cystinosin is required for the correct subcellular localization and trafficking of NHE3 and for sodium uptake. Introducing *CTNS* successfully rescues these defects in CTNS- deficient PTCs, whereas CTNS-LKG, encoding the lysosomal and plasma membrane isoform of cystinosin, did not. NHE3 mislocalization was confirmed in *Ctns*⁻/⁻ mice and cystinosis patient kidney. Transplantation of wild-type hematopoietic stem and progenitor cells in *Ctns*⁻/⁻ mice restores NHE3 expression at the brush border membrane and improves FS-related phenotypes. This study uncovers an evolutionary conserved novel role of cystinosin in NHE3 trafficking, offering insights into FS pathogenesis and potential new therapeutic avenues.

**Keywords** Cystinosis; Na⁺/H⁺ Exchanger 3; Proximal Tubular Cells; Renal Fanconi Syndrome; Hematopoietic Stem and Progenitor Cells
**Subject Categories** Membranes & Trafficking; Molecular Biology of Disease; Organelles

## Introduction

Cystinosis is an autosomal recessive lysosomal storage disorder caused by mutations in the *CTNS* gene which encodes for the cystine transporter protein, cystinosin (Kalatzis, 2001). While *CTNS* is ubiquitously expressed and cystine accumulates in all tissues, the kidney is the primary organ affected which shows the earliest manifestation of the disease phenotype with renal Fanconi syndrome (FS) at 6–18 months of age (Cherqui and Courtoy, 2017). The renal FS manifests by generalized dysfunction of the proximal tubules resulting in the ineffective reabsorption of essential nutrients such as glucose, uric acid, phosphate, amino acids, and bicarbonate (HCO₃⁻) (Cherqui and Courtoy, 2017; Elmonem et al, 2016). Interestingly, the onset of FS in cystinosis patients occurs almost concurrently with dedifferentiation of proximal tubular cells (PTCs) often prior structural abnormalities, such as cystine crystal formation or accumulation can be detected, suggesting that the pathophysiology of cystinosis involves mechanism beyond cystine accumulation (Mahoney and Striker, 2000). Cystinosis is currently the leading cause of inherited renal Fanconi syndrome (FS) in children, representing up to 20% of patients with hereditary tubular disorders (Cherqui and Courtoy, 2017; Elmonem et al, 2016). Patients affected with cystinosis also develop a progressive loss of glomerular function eventually leading to renal failure in the second or third decade. The non-renal manifestations of cystinosis have become apparent, and they develop severe impairment of other organs including the heart, thyroid, muscle, pancreas, eye, and central nervous system (Nesterova and Gahl, 2008).

The mouse model of cystinosis, the *Ctns*⁻/⁻ mice, accumulates cystine and cystine crystals, pathognomonic of cystinosis, in all tissues (Cherqui et al, 2002). The pure strain of C57BL/6 *Ctns*⁻/⁻ mice develop kidney pathology, in particular, a renal Fanconi syndrome, albeit less severe than in humans, and eventually end-stage renal failure (Nevo et al, 2010). We previously reported that syngeneic wild-type (WT) hematopoietic stem and progenitor cell (HSPC) transplantation in myeloablated *Ctns*⁻/⁻ mice resulted in HSPC-derived engraftment within the kidney, improvement of the renal Fanconi syndrome, and long-term preservation of kidney function and structure (Syres et al, 2009; Yeagy et al, 2011). The mechanism underlying this therapeutic effect involves the differentiation of WT HSPCs into macrophages, which provide "healthy lysosomes" carrying the functional protein cystinosin to the host diseased cells via extension of tunneling nanotubes (TNTs) that can even cross the basement membrane to deliver cystinosin to the PTCs (Naphade et al, 2015). We developed an autologous HSPC transplantation after ex vivo gene-correction of the HSPCs using a lentiviral vector containing the *CTNS* cDNA (Dull et al, 1998; Harrison et al, 2013), and conducted a phase 1/2 clinical trial

[1]Department of Pediatrics, University of California, San Diego, La Jolla, CA, USA. [2]Division of Biological Sciences, University of California, San Diego, La Jolla, CA, USA. [3]Department of Molecular Medicine and Molecular and Cellular Biology, The Scripps Research Institute, La Jolla, CA, USA. ✉E-mail: scherqui@ucsd.edu

including six adult patients (ClinicalTrials.gov Identifier: NCT03897361) (Cherqui, 2021). While the impact of the therapy on the kidney disease could not be evaluated in this trial because the patients had too advance kidney disease or kidney transplants, the overall outcomes are promising.

The current substrate reduction therapy for cystinosis, cysteamine, delays disease progression but does not prevent renal Fanconi syndrome, eventually leading to end-stage renal failure (Cherqui, 2012). This strongly suggests that cystinosin has another specific function in the proximal tubular cells beyond lysosomal cystine-proton cotransporter. In recent years, studies have revealed novel roles of cystinosin in PTCs such as its involvement in autophagy (Andrzejewska et al, 2016; Luciani et al, 2018; Festa et al, 2018; Berquez et al, 2023), including chaperone-mediated autophagy (Napolitano et al, 2015), TFEB expression (Rega et al, 2016), and inflammation (Lobry et al, 2019). Notably, none of these defective pathways were rescued by cysteamine (Andrzejewska et al, 2016; Napolitano et al, 2015; Rega et al, 2016). Furthermore, there are two isoforms of cystinosin protein due to alternate splicing of *CTNS* within exon 12 resulting in the exclusion of the GYDQL motif at the carboxyl-terminal and the longer *CTNS-LKG* isoform. This variant is expressed in various cellular compartments, but its specific function is unknown (Taranta et al, 2008).

In this study, we report a novel interaction between cystinosin and members of the sodium-hydrogen (Na$^+$/H$^+$) exchanger family (NHEs). The Na$^+$/H$^+$ exchanger genes (SLC9 family), which include 10 isoforms with different tissue and cellular localization, play an essential role in maintaining cellular homeostasis, regulating factors such as cell pH, volume, and sodium concentration (Nakamura et al, 2005; Orlowski and Grinstein, 2004; Slepkov et al, 2007). The Na$^+$/H$^+$ exchanger isoform 3 (NHE3) is of particular interest because of its localization at the apical membrane of proximal tubules in the kidney and its role in the absorption of water and sodium (Nwia et al, 2022). NHE3 is also crucial for preventing metabolic acidosis by enabling bicarbonate reabsorption (Li et al, 2013). We showed that cystinosin and NHE3 interact and colocalize at the endosomes in the yeast model and human PTCs, and that cystinosin is necessary for its trafficking to the brush border in human PTCs. We also demonstrated that cystinosin, but not cystinosinLKG, is able to rescue the defective vesicular trafficking, and sodium and albumin uptake in *CTNS*-deficient PTCs. Notably, mislocalization of NHE3 was confirmed in *Ctns*$^{-/-}$ murine and cystinosis patient kidneys, and WT HSPC transplant was able to rescue its localization to the brush border. Together, this study provides a new role of cystinosin in the PTCs, supporting a molecular mechanism underlying the renal FS in cystinosis and supporting new therapeutic targets for this pathology.

## Results

### Ers1 colocalizes and interacts with Nhx1 in the endosomes in yeasts

Ers1 is well conserved and shares structural and functional similarities with human cystinosin (Fig. EV1A,B) (Gao et al, 2005). A high-throughput screen using modified split-ubiquitin technology in *S. cerevisiae* previously identified an interaction

between the yeast cystinosin homolog, Ers1, and the NHE homolog protein, Nhx1 (Miller et al, 2005). The functions of mammalian NHE exhibit conservation in yeast as well, where Nhx1 transports both K$^+$ and Na$^+$, facilitating vacuolar Na$^+$ sequestration by coupling its movement to the proton gradient generated by vacuolar H$^+$-ATPase, thereby playing a role in pH regulation within luminal and cytoplasmic compartments and enhancing salt and osmotic tolerance (Brett et al, 2005; Nass and Rao, 1999; Nass et al, 1997). To confirm the interaction between Ers1 and Nhx1, we used the methylotrophic yeast *Pichia pastoris*, where the two cystinosin isoforms found in higher eukaryotes, exist but are encoded by two distinct genes: Ers1S (short), homologous to cystinosin, and Ers1L (long), homologous to cystinosinLKG, unlike *S. cerevisiae* that only has the short isoform, Ers1S (Fig. EV1A,B). Like *S. cerevisiae*, both forms of *P. pastoris* Ers1 structure are well conserved but lack the luminal N-terminal domain (NTD) (Fig. EV1A,B). This suggests that *P. pastoris* has evolved a divergent mechanism to achieve the same functional outcome, making it an excellent model to study cystinosis. To investigate the potential interaction, we performed the in vivo live-cell imaging assay, Bimolecular Fluorescence Complementation (BiFC), for direct visualization and localization of protein-protein interactions. Two non-fluorescent fragments of the Venus fluorescent protein (Venus-N and Venus-C) were fused with Ers1S and Nhx1 to create Ers1S-Venus-N (Ers1S-VN) and Nhx1-Venus-C (Nhx1-VC), respectively. Fluorescent signal generated by complementation of these two non-fluorescent fragments was observed in small vesicles in the cytoplasm and confirmed the interaction of Ers1S and Nhx1 in these cellular compartments (Figs. 1A and EV1C). Budding yeast possess a minimal endomembrane system lacking a distinct early endosome (EE), and the trans-Golgi network (TGN) serves as the EE (Day et al, 2018). In WT cells, about half of Nhx1-GFP puncta colocalized with the late endosome marker Vps8 (Fig. 1E). Interestingly, BiFC reconstitution assay revealed that the majority of Ers1S-VN+Nhx1-VC dots colocalized with Vps8, with only occasional overlap with the TGN/EE marker Sec7 (Figs. 1E and EV1C). This indicates that the detectable Ers1S-Nhx1 interaction occurs predominantly at the late endosome (Figs. 1A,E and EV1C). These results are consistent with the endosomal localization of Nhx1 reported in *S. cerevisiae* (Nass and Rao, 1998).

We then examined the subcellular localization of Ers1S using Ers1S-GFP fusion protein in ΔErs1S *P. pastoris* strain and conducted pulse-chase experiments with the dye FM4-64, which enters the yeast via endocytosis and selectively stains endocytic intermediates, including early and late endosomes, as well as the vacuole, with red fluorescence (Vida and Emr, 1995). By fluorescence microscopy, we observed Ers1S-GFP colocalizing with endocytic intermediates labeled with FM4-64 after a short pulse-chase (45 min) but showed surprisingly no localization with the vacuole at either 45 or 105 min of the chase (Fig. 1B).

We speculated that Ers1 might play a role in Nhx1 localization and/or function at the endosome, impacting proper trafficking within the endocytic pathway. Hence, we investigated whether Ers1 contributes to cellular trafficking of Nhx1 by generating various mutants of Ers1S/L, including the double *ers1* deletion strain (Δ*ers1S* Δ*ers1L*), and visualizing Nhx1-GFP and Vps8-2xmCherry, the LE marker. As expected, Nhx1 colocalized with Vps8 in wild-type cells; however, colocalization was also observed in single *ers1* (Δ*ers1S* or Δ*ers1L*) and double (Δ*ers1S* Δ*ers1L*) mutant cells.

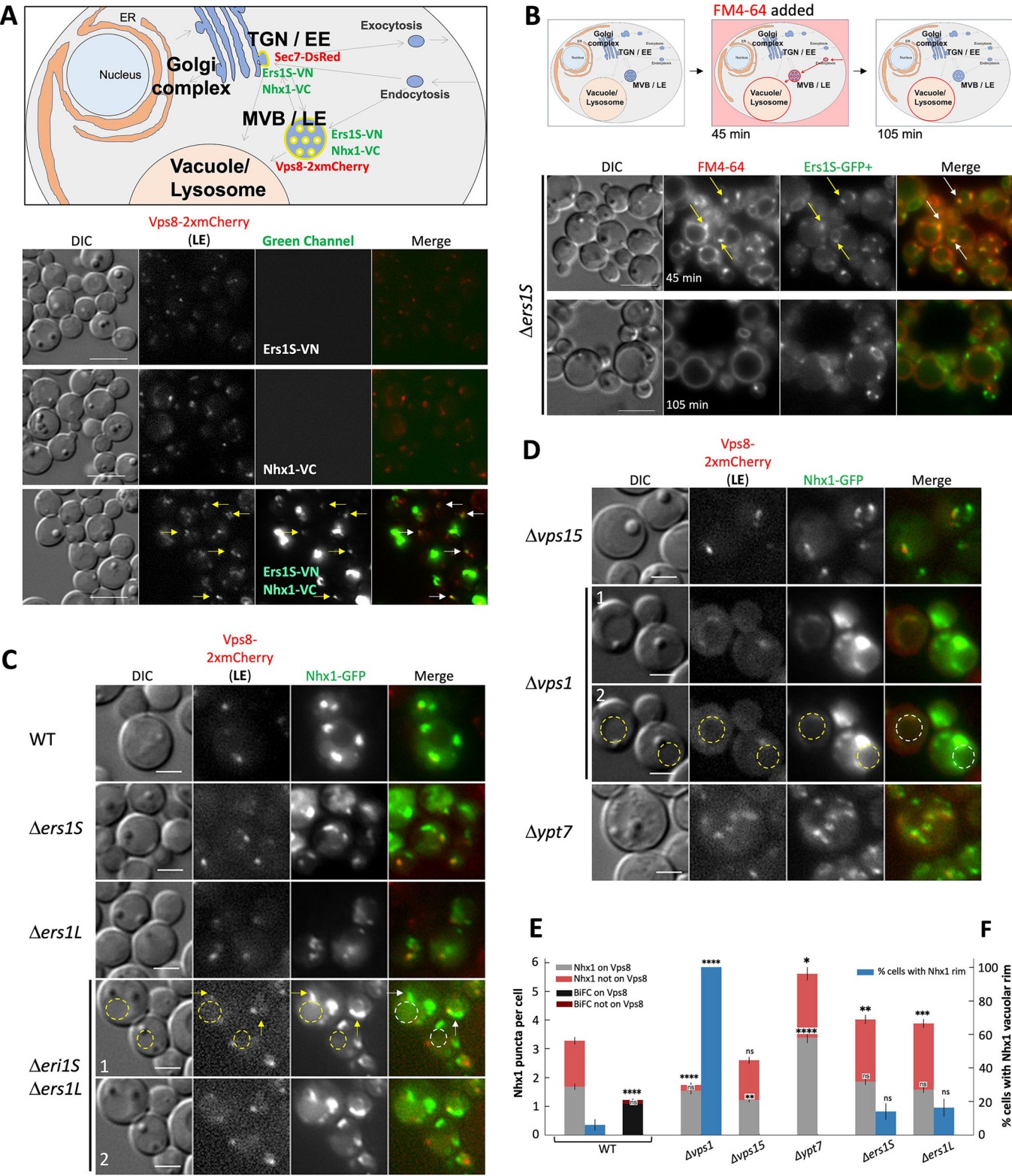

Interestingly, two major differences were observed for Nhx1 localizations in the Δ*ers1S* Δ*ers1L* double mutant, first Nhx1 protruded beyond Vps8 (yellow arrows; Fig. 1C) and second a large fraction of Nhx1 localized into the vacuolar matrix (yellow circle; Fig. 1C). This suggests that the double deletion of Ers1 (S and L)

might affect late endosome/multivesicular body (MVB) morphology. In the absence of both Ers1 form, Nhx1 might be degraded by vacuolar proteases, consistent with Fig. 1C (yellow circle).

With the goal of understanding the mechanism of Ers1's trafficking role in Nhx1, we investigated the localization of Nhx1 in

◀ **Figure 1. Ers1 colocalizes and interacts with Nhx1 in the endosomes in yeast cells.**

(A) Schematic representation of the BiFC assay in *P. pastoris* displaying an interaction between Ers1S and Nhx1, and their colocalization with the early and the late endosome marker, Sec7-DsRed (TGN/EE) and Vps8-2xmCherry (MVB/LE) (Upper & Lower panel) (white arrows and Fig. EV1C). Negative controls, the single Venus moieties (Ers1S-VN and Nhx1-VC) do not fluoresce in the green or red channel (Lower panel) (yellow arrows). (B) Schematic representation of the vital stain FM4-64 internalization and transport to the vacuole/lysosome in yeast. Pulse-chase of FM4-64 followed by a short incubation (45 min) stains endocytic intermediates, such as early and late endosome and the vacuole, whereas a longer incubation (105 min) labels mainly the vacuole (Upper panel). Fluorescent microscopy shows colocalization of Ers1S-GFP with endocytic intermediates labeled with FM4-64 (45 min) but no colocalization with the vacuole at either 45 or 105 min. Cells deleted for ERS1S gene were complemented with a plasmid comprising Ers1S-GFP under regulation of the Ers1 promoter. Arrows indicate colocalization of Ers1S-GFP with endocytic intermediates (Lower panel). (C, D) Localization of Nhx1-GFP expressed under the Nhx1 promoter in different mutant strains including Δers1S and Δers1L mutants (C) or different mutants of the endosomal trafficking pathway (D) (dashed circles indicate the vacuole and arrows indicate Nhx1-GFP localization protruding from the late endosome). Panels 1 and 2 show the same images, with and without annotations. Scale bars: 5 μm (A, B); 2 μm (C, D). (E) Nhx1 puncta per cell separated into those colocalizing with Vps8 (gray) and those not colocalizing (red). Also shown is the quantification of Ers1S-VN+Nhx1-VC BiFC puncta colocalization with Vps8. (F) Percentage of cells in which Nhx1 decorated the vacuolar rim (blue). This panel summarizes mislocalization by separating endosomal versus non-endosomal Nhx1. Sample sizes (cells): WT $n = 50$, Δvps15 $n = 50$, Δvps1 $n = 50$, Δypt7 $n = 36$, Δers1S $n = 50$, Δers1L $n = 50$ (≥2 independent experiments). Bars represents mean ± standard error of mean (SEM). Mann–Whitney U test was used for both "on Vps8" and "not on Vps8" comparisons (E) and Fisher's exact test for (F). Asterisks indicate statistical significance relative to WT, For (E) Δvps15_on_Vps8, $P = 0.001$; Δypt7_not on_Vps8, $P = 0.038$; Δers1S_not on_Vps8, $P = 0.006$; Δers1L_not on_Vps8, $P = 0.001$ (*$P < 0.05$; **$P < 0.01$; ***$P < 0.001$; ****$P < 0.0001$; ns, not significant). Source data are available online for this figure.

several single mutants implicated in vesicular trafficking in yeast, such as Δvps1 (dynamin GTPase homolog implicated in several steps of the endocytic pathways, homolog of mammalian dynamin-1) (Rooij et al, 2010), Δvps15 (member of phosphatidylinositol 3-kinase complex required for autophagy and endosomal vacuolar protein sorting, homolog of mammalian p150) (Stack et al, 1993), and Δypt7 (GTPase required for vesicular fusion with the vacuole, homolog of mammalian RAB7) (Wichmann et al, 1992).

In Δvps15 cells, we observed fewer Vps8 puncta, but the relative proportion of Nhx1 puncta colocalizing with Vps8 remained similar to WT (Fig. 1D,E), consistent with the role of the PI3-kinase complex in maintaining endosome identity rather than directly altering Nhx1 targeting (Stack et al, 1993). In Δypt7 cells, we observed numerous small puncta of both Nhx1 and Vps8 that were still highly colocalized (Fig. 1D,E), reflecting increased fragmentation due to loss of Rab7-mediated vacuole fusion (Wichmann et al, 1992). By contrast, in Δvps1 cells, Nhx1 accumulated around the vacuole/lysosome (yellow circle, Fig. 1D,E), similar to what has been reported for Snc1 (mammalian homolog of vertebrate synaptic vesicle-associated membrane proteins [VAMPs] or synaptobrevins) (Lukehart et al, 2013), Vps10 (mammalian homolog of sortilin) (Arlt et al, 2015), and Pep12 (mammalian homolog of syntaxins) (Hayden et al, 2013) in yeast. These data confirm that Nhx1 traffics through the endosomal system and suggests that Vps1-dependent recycling helps return Nhx1 from the vacuole-proximal region back into endosomal compartments. Quantification of Nhx1-GFP and Vps8-2xmCherry colocalization revealed these distinct patterns across mutants (Figs. 1C–F and EV1D–G). In Δers1 strains, Vps8 puncta persisted, but Nhx1 was mislocalized, with more non-endosomal localization and occasionally spreading onto the vacuolar rim in some cells. Quantification further showed that both Ers1S and Ers1L mutants reduced the fraction of Nhx1 puncta colocalizing with Vps8+ endosomes (Fig. 1E), supporting a role of Ers1 in trafficking Nhx1 to late endosomes and maintaining its proper endosomal localization.

Finally, we aimed to confirm whether the deletion of Ers1 in *P. pastoris*, similar to *S. cerevisiae* (Simpkins et al, 2016), did not lead to cystine accumulation. We studied *P. pastoris* strains lacking Ers1S, Ers1L, or both, under both growth conditions (glucose medium) and nitrogen starvation conditions (starvation condition), where cells arrest the cell cycle and induce autophagy to recycle material for cell survival (Fig. EV1H). Under growth conditions, we confirmed that *P. pastoris* behaved similarly to *S. cerevisiae*, with

*ERS1* deletion strains not accumulating cystine. Surprisingly, the same result was observed under nitrogen starvation conditions. Nitrogen starvation in yeast induces autophagy and increases amino acid content in the vacuole. Altogether, these results support that Ers1 protein does not function as a cystine transporter in the yeast and is involved in the vesicular trafficking of Nhx1.

## Cystinosin interacts and localizes with NHE3 & NHE2, but not with NHE1

To investigate potential interaction between NHE3 and cystinosin in mammalian cells, we generated stable HEK293T and CTNS-deficient HK-2 cell lines with lentiviral vectors (LV) expressing *NHE3* fused with the green fluorescent protein (*GFP*; LV-NHE3-GFP), along with *CTNS* and *CTNS-LKG* fused with *DsRed* reporter gene (LV-CTNS-DsRed; LV-CTNS-LKG-DsRed). Subsequent immunoprecipitation (IP) assays using GFP or DsRed tags from whole cell lysates demonstrated distinct co-precipitation of NHE3-GFP with both cystinosin-DsRed and cystinosinLKG-DsRed, and vice versa (Figs. 2A and EV2A). Additionally, positive colocalization patterns between cystinosin-DsRed and NHE3-GFP and between cystinosinLKG-DsRed and NHE3-GFP observed by fluorescent microscopy within distinct intracellular vesicles (Fig. 2B). Subsequently, we aimed to determine whether this interaction was specific to NHE3 so we examined two other isoforms, NHE1, expressed ubiquitously, and NHE2, expressed primarily in the gastrointestinal tract and kidney (Slepkov et al, 2007; Tse et al, 1993; Orlowski and Grinstein, 1997). Intriguingly, we found that NHE2-GFP, but not NHE1-GFP, co-precipitated with cystinosin-DsRed and cystinosinLKG-DsRed, and vice versa (Figs. 2C and EV2B). Altogether, these results demonstrated the interaction of cystinosin and NHE3 and NHE2 and underscored the specificity of cystinosin interaction with some Na+/H+ exchanger proteins.

## CTNS−/− PTC lines exhibit decreased expression, mislocalization of NHE3 and impaired sodium content and albumin uptake

To determine whether *CTNS* influences the expression of the NHE isoforms under investigation, we measured their endogenous expression levels in the proximal tubular cell line HK2 wild-type

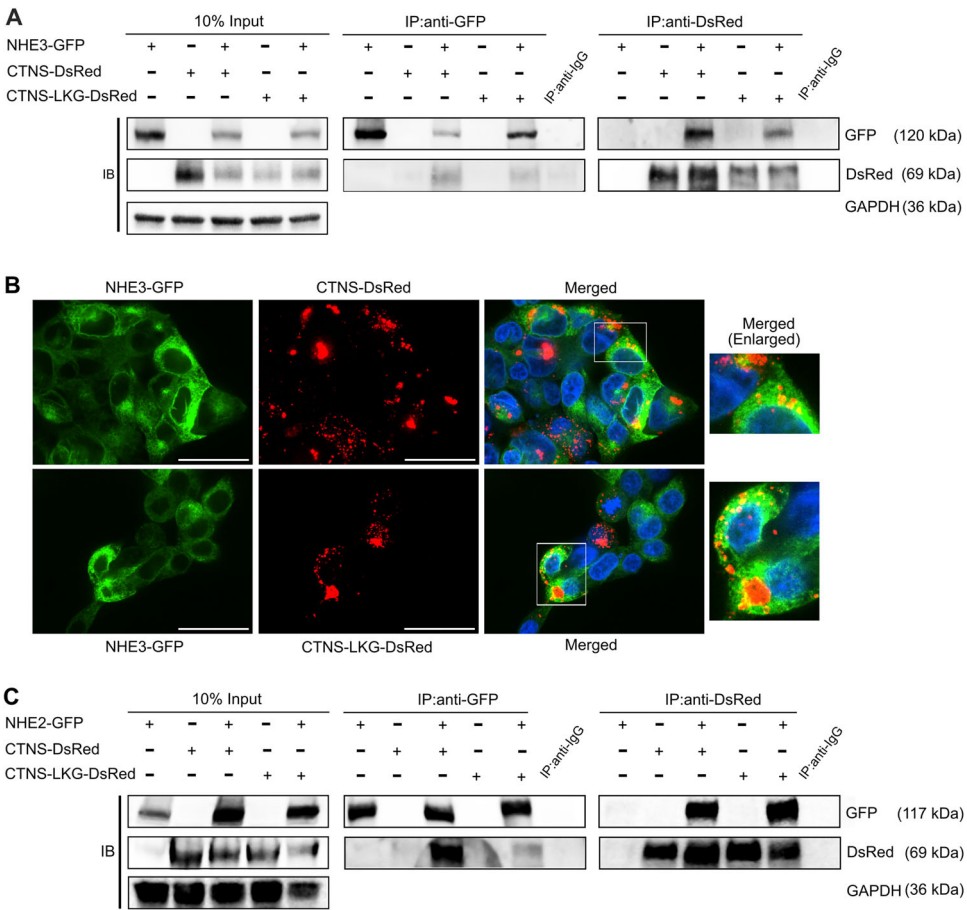

**Figure 2. Cystinosin interacts with NHE3 and NHE2.**

(A) Immunoprecipitation assay performed in HEK293T cells stably expressing NHE3, cystinosin, and cystinosinLKG. Pull down using anti-GFP and anti-DsRed demonstrate the interaction between NHE3 and both cystinosin and cystinosinLKG. (B) Representative immunofluorescence images of HEK293T cells stably expressing NHE3-GFP and cystinosin-DsRed, and NHE3-GFP and cystinosinLKG-DsRed depicting colocalization of NHE3 with cystinosin, and NHE3 with cystinosinLKG in intracellular vesicles. Scale bars: 40 μm. (C) Immunoprecipitation assay performed in HEK293T cells stably expressing NHE2, cystinosin, and cystinosinLKG showing interaction between NHE2 and both cystinosin and cystinosinLKG. For both (A, C) input lysates show proper expression of the proteins with glyceraldehyde-3-phosphate dehydrogenase (GAPDH) used as a loading control. Pull down with anti-IgG was used as a negative control. A representative image from n = 3 biological repeats is shown. Source data are available online for this figure.

(WT) and *CTNS*$^{-/-}$ cells. At the mRNA level, quantitative real-time polymerase chain reaction (PCR) analysis did not reveal major changes in *NHE3* and *NHE1* transcripts in *CTNS*$^{-/-}$ HK2 cell lines compared to WT (Fig. 3A). However, interestingly, significant increase in *NHE2* transcripts was observed in *CTNS*$^{-/-}$ cell compared to WT. As expected, *CTNS* transcripts were not detected in *CTNS*$^{-/-}$ HK2 cells, which accumulate large amount of cystine (Zhang et al, 2019), confirming their genetic status. Western blot analysis conducted on total cell extracts revealed a significant decrease in NHE3 expression as well as a significant increase in NHE2 expression in *CTNS*$^{-/-}$ cells compared to WT cells. No alteration in the protein expression of NHE1 was observed (Fig. 3B).

To characterize the subcellular localization of NHE3 in PTCs, we transduced the WT and *CTNS*$^{-/-}$ HK-2 cells with LV-NHE3-GFP. Western blot analysis confirmed comparable NHE3-GFP expression levels in both cell types (Fig. EV3A). We studied NHE3-GFP localization pattern with lysosomes (LAMP1), early

endosomes (EEA1), Golgi (GM130), and endoplasmic reticulum (ER) (Figs. 3C,D and EV3B,C). We observed significant differences in the localization pattern of the NHE3 protein, being localized primarily in the ER and to a lesser extent in the lysosome, Golgi and early endosomes under *CTNS*$^{-/-}$ condition as opposed to WT. These data suggest that the absence of cystinosin causes trafficking defects inducing the accumulation of NHE3 in the ER (Figs. 3C,D and EV3B,C). Also, NHE3 mislocalization was direct consequence of cystinosin loss and not cystine storage, as evidence by cysteamine treatment of WT and *CTNS*$^{-/-}$ HK2 cells that did not correct the defect (Fig. EV3D).

Finally, we investigated the impact of the absence of cystinosin on NHE3 function. NHE3 is involved in sodium uptake in PTCs and plays a role in albumin endocytosis (Gekle et al, 1999). To assess intracellular sodium content the cells were incubated with Sodium Green™ Tetraacetate for 8 min. To induce the expression of megalin and cubilin in HK2 cells, we treated them with albumin555 for 16 h as previously described (Yang et al, 2008; Liu et al, 2015).

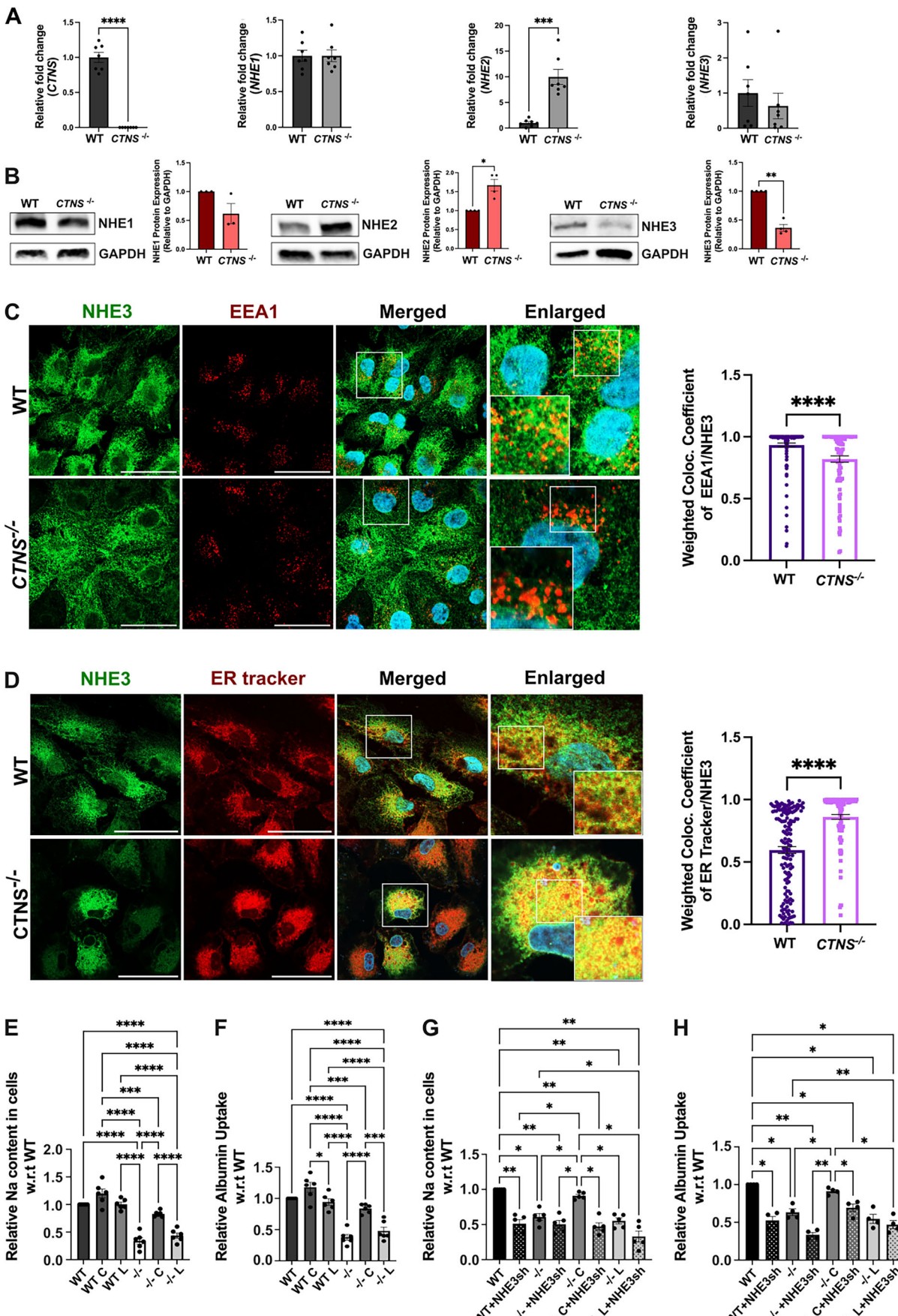

**Figure 3. Cystinosin deficiency affects NHE3 expression, causing its mislocalization and impairing sodium and albumin uptake in human WT and $CTNS^{-/-}$ HK2 proximal tubular cells.**

(A) *CTNS*, *NHE1*, *NHE2*, and *NHE3* expression measured with quantitative polymerase chain reaction. (B) Representative Western blot analysis of NHE1, NHE2 and NHE3 expression with GAPDH used as a loading control. (C, D) Representative immunofluorescence images of colocalization of NHE3-GFP (green) with the early endosome marker EEA1(red) (C) and with the endoplasmic reticulum (ER tracker; red). (D) The corresponding quantification showed a defect in NHE3 trafficking that accumulated in the ER under $CTNS^{-/-}$ condition. Scale bars: 50 μm (C, D). (E, F) Sodium (Na) and albumin uptake studies in WT and $CTNS^{-/-}$ HK2 cells (−/−), and in cells transduced with LV-CTNS (C) or LV-CTNS-LKG (L) measured using Sodium Green™ Tetraacetate and albumin555 by flow cytometry. (G, H) Na and albumin uptake in WT and $CTNS^{-/-}$ HK2 cells (−/−), and in $CTNS^{-/-}$ HK2 cells transduced with LV-CTNS (C) or LV-CTNS-LKG (L), in control and NHE3 shRNA treated cells. For data in figure (A, B, E–H) each dot represents an independent biological replicate (A, $n = 7$; B, $n = 3$ for NHE1, $n = 4$ for NHE2 and NHE3; E, $n = 6$; F, $n = 6$, G, $n = 5$ & H, $n = 4$). The data in (C, D) represent the means ± SEM from $n = 3$ biological repeats. Bar graphs are presented as the mean ± SEM and for (A–D) two-tailed Student's t-test and for (E–H) One-way analysis of variance (ANOVA) was used. For (A) $P = 0.0007$ for *NHE2*; (B) $P = 0.0231$ for NHE2, $P = 0.0017$ for NHE3. For (E) $P = 0.0001$ for WT C vs. −/− C. For (F) $P = 0.0304$, $P = 0.0007$ and $P = 0.0004$ for WT C vs. WT L, WT C vs. −/− C and −/− C vs. −/− L, respectively. For (G) $P = 0.0085$, $P = 0.0107$, $P = 0.0068$, $P = 0.0043$, $P = 0.0038$, $P = 0.0082$, $P = 0.0259$, $P = 0.0479$, $P = 0.0156$, $P = 0.0183$, $P = 0.0293$, $P = 0.0261$ and $P = 0.0196$ for WT vs. WT+NHE3sh, WT vs. −/−, WT vs. −/− + NHE3sh, WT vs. −/− C + NHE3sh, WT vs. −/− L, WT vs. −/− L + NHE3sh, WT + NHE3sh vs. −/− C, −/− vs. −/− C, −/− vs. −/− L + NHE3sh, −/− + NHE3sh vs. −/− C, −/− C vs. −/− C + NHE3sh, −/− C vs. −/− L, and −/− C vs. −/− L + NHE3sh, respectively. For (H) $P = 0.0168$, $P = 0.0232$, $P = 0.0015$, $P = 0.0492$, $P = 0.0293$, $P = 0.0142$, $P = 0.0431$, $P = 0.0071$, $P = 0.0095$, $P = 0.0270$ and $P = 0.0201$ for WT vs. WT+NHE3sh, WT vs. −/−, WT vs. −/− + NHE3sh, WT vs. −/− C + NHE3sh, WT vs. −/− L, WT vs. −/− L + NHE3sh, −/− vs. −/− C, −/− vs. −/− L + NHE3sh, −/− + NHE3sh vs. −/− C, −/− C vs. −/− C + NHE3sh and −/− C vs. −/− L + NHE3sh, respectively. *$P < 0.05$; **$P < 0.01$; ***$P < 0.001$; ****$P < 0.0001$. Source data are available online for this figure.

We observed that 16-h albumin increased megalin expression in the WT cells, but not in the $CTNS^{-/-}$ HK2 cells (Fig. EV4A), consistent with the reported defects in megalin trafficking and expression in cystinosis cells (Zhang et al, 2019; Raggi et al, 2014; Gaide Chevronnay et al, 2014). Intracellular sodium and albumin uptake were then measured using flow cytometry analysis. We observed significantly lower intracellular sodium and albumin uptake in the $CTNS^{-/-}$ cells compared to WT for 16 h treatment. Time-course analysis of albumin uptake was reduced in $CTNS^{-/-}$ cells as early as 1-h post-stimulation, and this defect persisted overtime (Fig. EV4B). Interestingly, the defect was rescued when the cells were transduced with LV-CTNS, but not with LV-CTNS-LKG (Fig. 3E,F). To demonstrate that the defective uptake in $CTNS^{-/-}$ HK2 cells involves NHE3, these assays were conducted under NHE3 knockdown (shRNA; Figs. 3G,H and EV4C) and in the presence of the NHE3 inhibitor EIPA (Gekle et al, 2004, 2001) (Fig. EV4D,E). Both treatments resulted in a significant decrease in sodium content and albumin uptake in WT HK2 cells. In contrast, no significant decrease was detected in $CTNS^{-/-}$ HK2 cells treated or not with NHE3 shRNA or EIPA. This lack of effect persisted even after *CTNS* transduction in $CTNS^{-/-}$ HK2 cells treated with EIPA/NHE3 shRNA, indicating that the absence of cystinosin produces effects comparable to NHE3 inhibition or knockdown (Figs. 3G,H and EV4D,E). These results demonstrate the key role played by cystinosin in the regulation of NHE3 expression, localization and function.

## Cystinosin is involved in the cellular trafficking of NHE3 in PTCs

To confirm the defective trafficking of NHE3 in cystinosis PTCs, we studied the localization and expression of NHE3 protein in human healthy- (referred as Normal) and patient- (referred as CT) derived PTCs after transduction with LV-NHE3-GFP. NHE3 undergoes trafficking between the plasma membrane and endosomes (Gekle et al, 2002). We hypothesized that the defective vesicular trafficking of Nhx1 observed in Ers1-defective yeasts might also occur in mammals. To further explore the vesicular dynamics of NHE3 and determine if its mislocalization resulted from defective trafficking, we measured its vesicular transport using pseudo-Total Internal

Reflection Fluorescence microscopy (pTIRFM, oblique illumination). In pseudo-TIRFM, laser illumination is adjusted to impinge on the coverslip at an angle of incidence that is slightly less than the critical angle required for complete internal reflection. This technique facilitates imaging the cell surface by detecting the trafficking events in proximity to the plasma membrane while also imaging deeper areas in the cell than traditional TIRFM (the sample is illuminated up to ~1 μm depth) (Mudrakola et al, 2009). Thus, pTIRFM allows complete visualization of endosomes, lysosomes and other subcellular organelles but still maintaining the high signal-to-noise ratio of traditional TIRFM (Mudrakola et al, 2009). In this experiment, vesicles were monitored for a period of 60 s. The dynamic studies show that cystinotic cells are characterized by a significant decrease in the number of $NHE3^+$ organelles moving at high speed (>0.5 μm/s) and a concomitant increase in the number of these organelles with no or restricted movement (<0.14 μm/s) in cystinotic cells (Fig. 4B,C). These data suggest defective NHE3 trafficking in patient-derived PTCs (Fig. 4A,B). Remarkably, transduction of CT PTCs with LV-CTNS but not with LV-CTNS-LKG significantly reduced the number of $NHE3^+$ vesicles showing restricted movement in cystinotic cells (Fig. 4B), suggesting that cystinosin, but not cystinosinLKG, regulates NHE3 trafficking (Fig. 4A–C). Although representative still images are provided, the data from the accompanying video (Movies EV1–EV4) offers better insight into these trafficking dynamics.

Furthermore, sodium content in the CT cells was significantly lower in comparison to healthy cells, confirming a significant defect in sodium uptake in cystinosis patient PTCs compared to control (Fig. 4D). This defect was restored upon transduction with LV-CTNS, but not LV-CTNS-LKG. Altogether, these results show that cystinosin, but not cystinosinLKG, is involved in NHE3 vesicular trafficking and subcellular localization, impacting NHE3 function in PTCs.

## In vivo study of NHE3 expression in $Ctns^{-/-}$ mice and potential rescue by HSPC transplantation

To confirm the in vitro results on the impact of cystinosin deficiency on NHE3, we studied NHE3 expression in the mouse

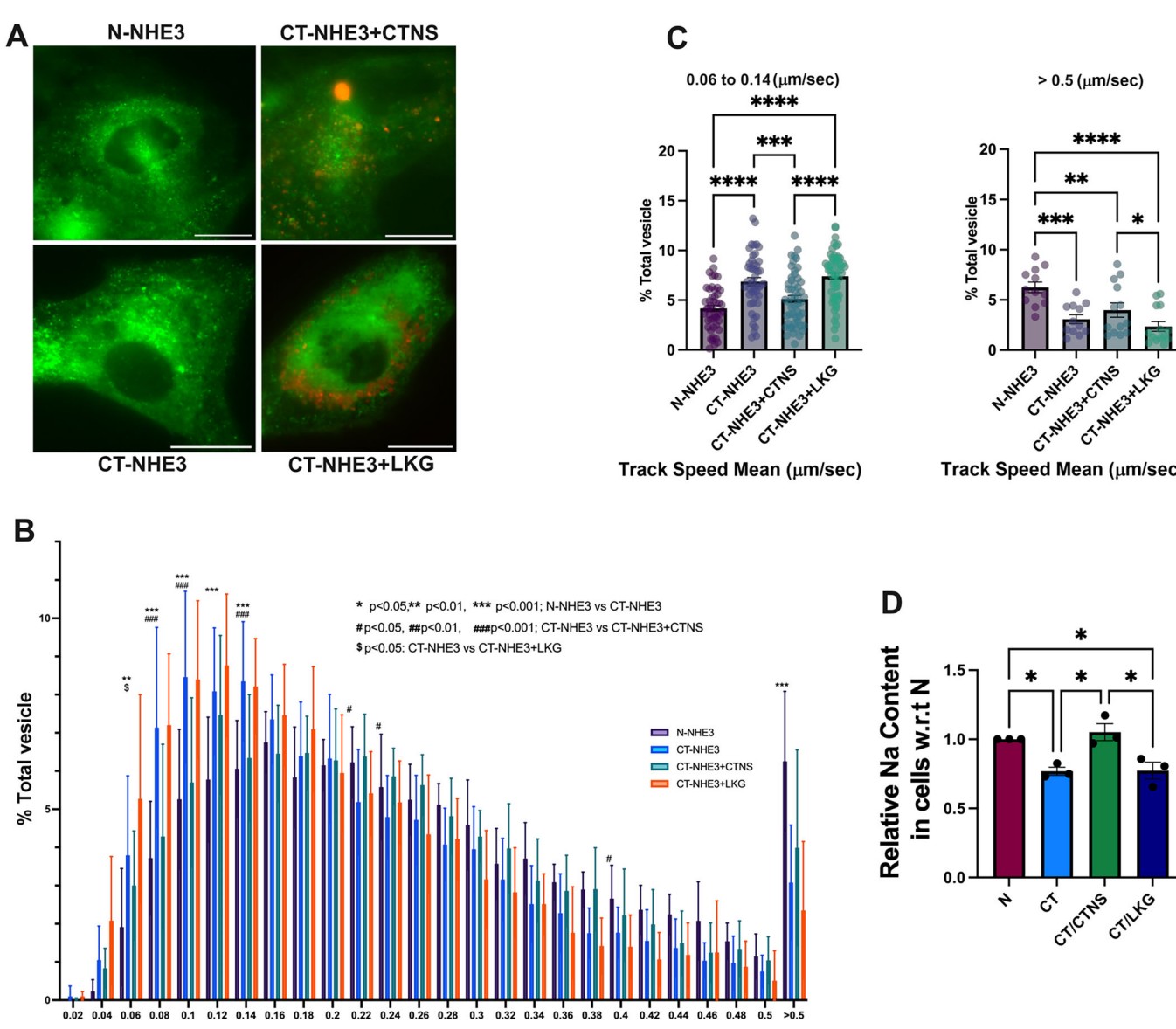

**Figure 4. NHE3 trafficking, and sodium uptake are defective in cystinosis patient's PTCs.**

(A) Representative immunofluorescence images of NHE3-GFP in human normal (N), cystinosis (CT) PTCs and CT cells transduced with LV-CTNS-DsRed and LV-CTNS-LKG-DsRed. Scale bars: 20 μm. (B) Total Internal Reflection Fluorescence microscopy was used to study GFP-tagged NHE3 trafficking in normal (N) and cystinosis patient (CT) PTCs. Rescue of NHE3 trafficking in CT PTCs was achieved with the transduction with LV-CTNS but not with LV-CTNS-LKG. (C) Bar graph showing comparison of slower (0.06–0.14 μm/s) and faster (>0.5μm/s) vesicles. (D) Intracellular sodium content in the N, CT PTCs and CT transduced with LV-CTNS or LV-CTNS-LKG following treatment with Sodium Green™ Tetraacetate as determined by flow cytometry. The data (B) represents the mean ± SD from $n = 3$ biological repeats. Two-way ANOVA was used, and the statistics is shown on the figure. For N-NHE3 vs. CT-NHE3, $P = 0.0043$, $P = 0.0001$ and $P = 0.0001$ for Track speed mean 0.06, 0.12 and 0.14 (μm/s), respectively. For CT-NHE3 + CTNS vs. CT-NHE3 + LKG, $P = 0.0162$, $P = 0.0124$, $P = 0.0003$, $P = 0.0130$, $P = 0.03$, $P = 0.0269$, $P = 0.0342$, $P = 0.0041$ and $P = 0.0016$ for Track speed mean 0.04, 0.12, 0.14, 0.26, 0.3, 0.32, 0.36, 0.38 and >0.5 (μm/s), respectively. For CT-NHE3 vs. CT-NHE3 + CTNS, $P = 0.0001$, $P = 0.0221$, $P = 0.0407$ and $P = 0.027$, for Track speed mean 0.14, 0.22, 0.24 and 0.38 (μm/s), respectively. For CT-NHE3 vs. CT-NHE3 + LKG, $P = 0.0052$ for Track speed mean 0.06 (μm/s). The data (C, D) represents the mean ± SEM from $n = 3$ biological repeats. For (C, D) One-way analysis of variance (ANOVA) was used. For (C) Right Panel, N-NHE3 vs. CT-NHE3 $P = 0.0003$, N-NHE3 vs. CT-NHE3 + CTNS $P = 0.0062$, CT-NHE3 + CTNS vs. CT-NHE3 + LKG $P = 0.0364$. Left Panel, CT-NHE3 vs. CT-NHE3 + CTNS $P = 0.0007$. For (D) N vs. CT $P = 0.0296$, N vs. CT/LKG $P = 0.0324$, CT vs. CT/CTNS $P = 0.0101$, CT/ CTNS vs. CT/LKG $P = 0.0109$. *$P < 0.05$; **$P < 0.01$; ***$P < 0.001$; ****$P < 0.0001$. Source data are available online for this figure.

model of cystinosis and assessed whether any defect could be restored following the transplantation of GFP+ WT HSPCs. We studied different groups of mice composed of age-matched sex-matched WT, untreated *Ctns*−/− mice, *Ctns*−/− mice transplanted with *Ctns*−/− HSPCs (Mock) and *Ctns*−/− mice transplanted with WT GFP+ HSPCs (Test) (Fig. 5A; Table 4). The mice were transplanted at 2 months of age and sacrificed 6 months later. We first investigated *Nhe2* and *Nhe3* expression in the kidney, and no

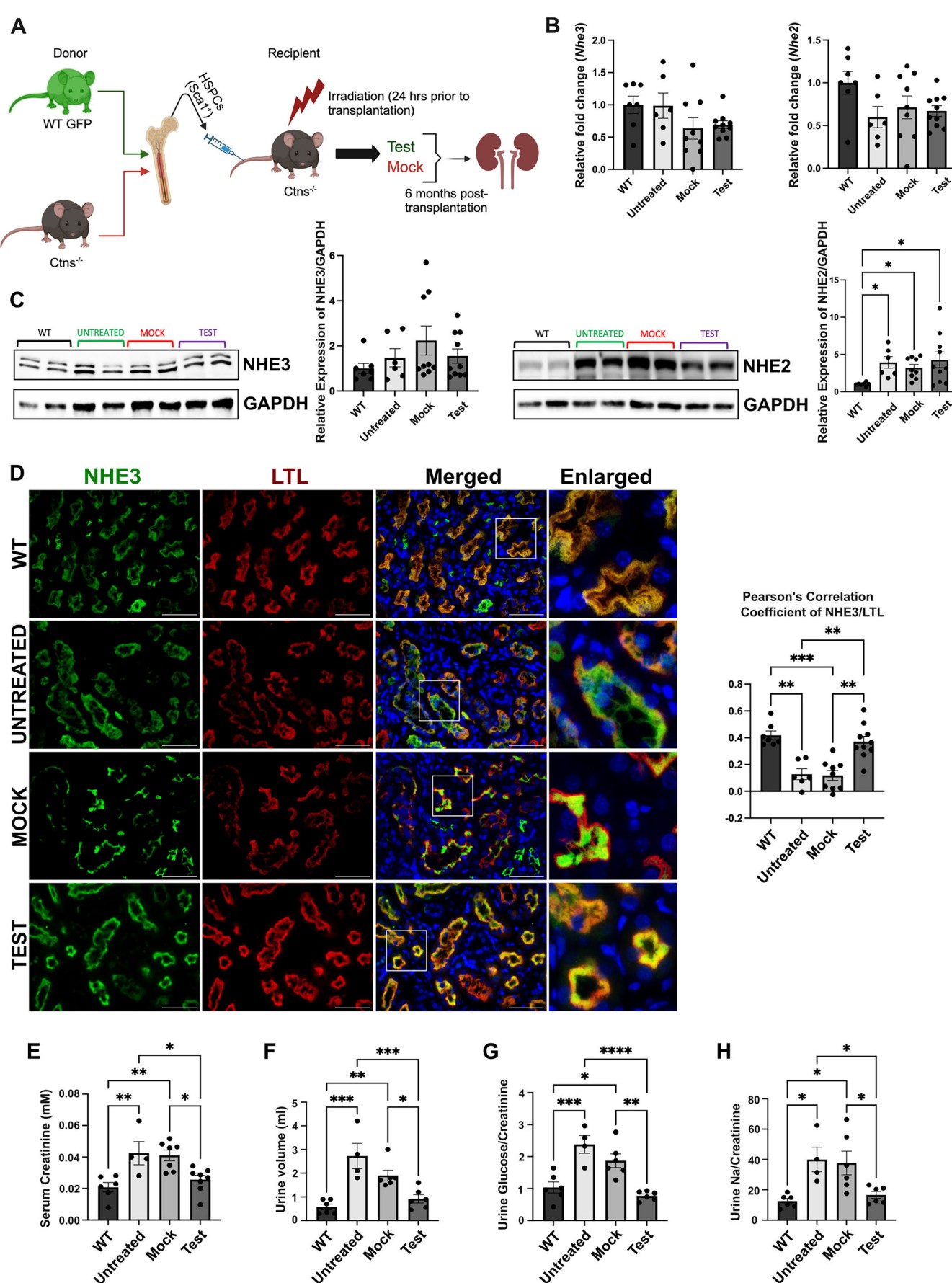

**Figure 5. NHE3 localization, but not expression, is affected in *Ctns*⁻/⁻ murine kidneys, and improves/normalizes following WT HSPC transplantation.**

(A) Schematic representation of WT HSPC transplantation procedure. The schematic was created with BioRender.com. (B) *Nhe2* and *Nhe3* expression measured by quantitative PCR in WT mice, untreated *Ctns*⁻/⁻ mice (untreated), *Ctns*⁻/⁻ mice transplanted with *Ctns*⁻/⁻ HSPCs (Mock), and *Ctns*⁻/⁻ mice transplanted with WT HSPCs (Test mice). (C) Representative western blot analysis and quantitative analysis of NHE3 and NHE2 expression with GAPDH used as a loading control. (D) Representative immunofluorescence images of kidney sections from WT, Ctns⁻/⁻, Mock and Test mice, stained with anti-NHE3 (green) and anti-Lotus tetragonolobus lectin (LTL-Rhodamine conjugated) (red). Corresponding colocalization quantification between NHE3 and LTL was performed using Pearson Correlation Coefficient. Scale bars: 50 μm. (E–H) Renal function in WT, untreated, mock-treated and Test mice was assessed by measuring serum creatinine level (mM) (E), and 24 h-urine volume (ml) (F), Glucose/Creatinine concentration (G), and sodium/creatinine concentration (H). Bar graphs are presented as the mean ± SEM. For data in figure (B–H), each dot represents an individual mouse. For (B–D) WT *n* = 7, Untreated *n* = 6, Mock *n* = 9, Test *n* = 10. For (E) WT *n* = 6, Untreated *n* = 4, Mock *n* = 7, Test *n* = 8. For (F–H) WT *n* = 6, Untreated *n* = 4, Mock *n* = 6, Test *n* = 6. One-way analysis of variance (ANOVA) was used. For Nhe2 (C) *P* = 0.0156, *P* = 0.0300 and *P* = 0.0101 for WT vs. Untreated, Mock and Test, respectively. For (D) *P* = 0.0013, *P* = 0.0002, *P* = 0.0056 and *P* = 0.0014 for WT vs. Untreated, WT vs. Mock, Untreated vs. Test and Mock vs. Test, respectively. For (E) *P* = 0.0088, *P* = 0.0045, *P* = 0.0371 and *P* = 0.0224 for WT vs. Untreated, WT vs. Mock, Untreated vs. Test and Mock vs. Test, respectively. For (F) *P* = 0.0001, *P* = 0.0055, *P* = 0.0009 and *P* = 0.0448 for WT vs. Untreated, WT vs. Mock, Untreated vs. Test and Mock vs. Test, respectively. For (G) *P* = 0.0006, *P* = 0.0156 and *P* = 0.0016 for WT vs. Untreated, WT vs. Mock and Mock vs. Test, respectively. For (H) *P* = 0.0162, *P* = 0.0137, *P* = 0.0457 and *P* = 0.0438 for WT vs. Untreated, WT vs. Mock, Untreated vs. Test and Mock vs. Test, respectively. *P < 0.05; **P < 0.01; ***P < 0.001; ****P < 0.0001. Source data are available online for this figure.

significant change was observed at the transcript level for both genes in any of the groups (Fig. 5B). NHE3 protein expression was also unchanged between the groups, but interestingly, NHE2 expression was significantly increased in the *Ctns*⁻/⁻ mouse untreated, Mock, and Test kidneys compared to WT (Fig. 5C) as observed in the HK2 cell lines. NHE3 protein is expressed in the brush border of the proximal tubular cells (Biemesderfer et al, 1993). We assessed the expression of NHE3 protein by immunofluorescence in the murine kidneys, using Lotus tetragonolobus lectin (LTL) as a brush border-specific marker. NHE3 appeared mislocalized in the proximal tubules of the untreated and Mock *Ctns*⁻/⁻ kidneys as demonstrated by Pearson's correlation coefficient (Fig. 5D). In contrast, significantly improved colocalization between NHE3 and LTL was observed in the Test group compared to untreated and Mock *Ctns*⁻/⁻ mice (Figs. 5D and EV5A). In addition, NHE3 and the ER marker, SERCA2-ATPase, exhibited a negative correlation in WT kidney, consistent with proper trafficking (Fig. EV5B). In contrast, untreated and Mock-transplanted *Ctns*⁻/⁻ kidneys showed a shift towards a positive correlation, indicating abnormal retention of NHE3 within the ER (Fig. EV5B). Remarkably, HSPCs transplantation restored normal NHE3 trafficking and localization, reducing its ER entrapment, as demonstrated by Pearson's correlation analysis (Fig. EV5B).

To demonstrate that correction of the cellular localization of NHE3 correlated with functional improvement of the renal FS in the WT HSPC-transplanted *Ctns*⁻/⁻ mice (Test group), we assessed key FS parameters in 24-h urine samples and evaluated kidney function by measuring serum creatinine. Test mice exhibited significantly decreased serum creatinine as well as polyuria, glucosuria and sodium loss compared to untreated and mock-treated *Ctns*⁻/⁻ mice (Fig. 5E–H). These results provide in vivo evidence that cystinosin deficiency leads to NHE3 mislocalization from the brush border to the ER, and causes renal Fanconi syndrome, which can be corrected by transplantation of *Ctns*-expressing HSPCs.

### Human cystinosis patient kidney exhibit mislocalized NHE3 expression

We verified the impact of the absence of cystinosin on NHE3 localization in human kidney. Formalin-fixed paraffin-embedded kidney biopsy tissue was obtained from a patient affected with cystinosis and donor control, and sections were stained for NHE3 and LTL marker. In the normal kidney sample, we observed a high degree of colocalization between NHE3 and LTL as demonstrated by Pearson's correlation coefficient (Fig. 6). However, in the patient tissue, NHE3 and LTL colocalization was significantly decreased (Fig. 6). Furthermore, while NHE3 and the ER marker SERCA2-ATPase were negatively correlated in control tissue, positive correlation was observed in the cystinosis patient kidney (Fig. EV6). This data confirms that NHE3 is mislocalized from the brush border to the ER in the kidneys of patients affected with cystinosis.

## Discussion

In this study, we report a novel interaction between cystinosin and a member of the sodium/hydrogen exchanger, NHE3, which plays a pivotal role in proximal tubular cell homeostasis. NHE3 is located at the apical membrane of kidney proximal tubules and contributes to the majority of renal sodium absorption (Nwia et al, 2022). In addition, NHE3 also indirectly contributes to bicarbonates reabsorption which is essential for maintaining acid base homeostasis and overall kidney function (Li et al, 2013). Furthermore, NHE3 undergoes dynamic cycling between the apical plasma membrane and the early endosomal compartment, participating in albumin endocytosis in proximal tubules (Gekle et al, 1999, 2004). The Na⁺/H⁺ exchange regulatory cofactor (NHERF) family are involved in NHE3 regulation, in multiple aspects of the regulated trafficking of NHE3 (Cha et al, 2010; Sarker et al, 2011; Murtazina et al, 2011, 2007). However, the mechanism by which NHERF proteins regulate the balance of NHE3 between the plasma membrane and endosomes has remained unclear, since the NHERFs are primarily localized to the plasma membrane, with no identified endosomal localization (Singh et al, 2015).

The reason underlying the pathogenesis of the renal Fanconi syndrome in cystinosis is still unresolved. Despite ubiquitous expression of cystinosin, its first manifestation is renal Fanconi syndrome before one year of age, revealing early severe dysfunction of the proximal tubular cells and reflecting their unique vulnerability in this disease. Decades of research and successive

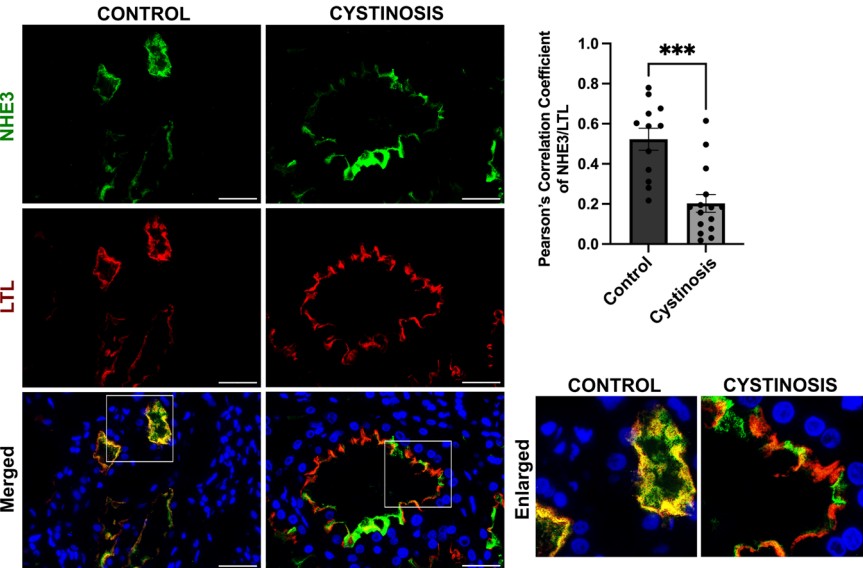

**Figure 6. NHE3 localization in human control donor and cystinosis patient kidney tissue sample.**

Representative immunofluorescence images of formalin fixed paraffin embedded (FFPE) human kidney sections from a control donor and a cystinosis patient. Sections were stained with anti-NHE3 (green), anti-Lotus tetragonolobus lectin (LTL-Rhodamine conjugated) (red) and DAPI (nuclei). Scale bars: 40 μm. The quantitative data of colocalization between NHE3 and LTL was determined by Pearson Correlation Coefficient. The data are presented as mean ± SEM from $n = 3$ biological repeats. Two-tailed Student's t-test was used; $P = 0.0001$, ***$P < 0.001$. Source data are available online for this figure.

hypotheses have attempted to explain the pathogenesis of the Fanconi syndrome in cystinosis from global metabolic or redox defects (Chol et al, 2004; Levtchenko et al, 2005; Mannucci et al, 2006), to impaired lysosomal trafficking and proteolysis (Raggi et al, 2014; Ivanova et al, 2015; Schulman et al, 1969) and to autophagy impairment (Andrzejewska et al, 2016; Festa et al, 2018; Berquez et al, 2023; Napolitano et al, 2015; Rega et al, 2022). These studies emphasize the importance of the presence of cystinosin per se for cellular homeostasis, that goes beyond its transport function. However, the underlying mechanism causing renal FS in cystinosis has yet to be explained. Cystinosin belongs to PQ-loop protein family which are predicted to function as cargo receptors in vesicle transport (Saudek, 2012), potentially linking cystinosin to broader lysosomal or endosomal trafficking processes beyond cystine transport. Our investigation of the interaction of cystinosin with NHE3 in the PTCs revealed that these two proteins colocalized within the endosomes, distinguishing this complex from the lysosomal cystine transporter function of cystinosin. Most importantly, in human *CTNS*-deficient PTCs, *Ctns*$^{-/-}$ mice and cystinosis patient kidney, NHE3 was mislocalized to the endoplasmic reticulum and NHE3 vesicular trafficking was impaired, providing the first evidence that cystinosin is involved in NHE3 trafficking. These novel findings are conserved in primitive models such as the *P. Pastoris* yeast model underscoring their importance. Indeed, we observed that the yeast orthologues of the NHE proteins and cystinosin, Nhx1 and Ers1, respectively, colocalized in the endocytic vesicles. Furthermore, mislocalization and deficient trafficking of Nhx1 were also observed in the Ers1-deficient lines. Ers1 has also been shown to move from phagosome to lysosome *in C.elegans* (Saudek, 2012; Yu et al, 2008). In *P. Pastoris*, Δers1S Δers1L double mutant, Nhx1 protruded beyond the late endosomes

and a large fraction localized into the vacuolar matrix. This suggests that the double deletions of Ers1 (S and L) affect late endosome/multivesicular body (MVB) morphology. Nhx1 is reported to be crucial for MVB vesicle formation in late endosomes (Mitsui et al, 2011) and MVB-vacuolar lysosome fusion, a key step in surface protein degradation during endocytosis (Karim and Brett, 2018). This correlates with the phenotypic trait we observed under double deletion of ERS1 (S and L), thus suggesting that Ers1 might function to retain Nhx1 in the endocytic pathway through an unknown retrograde/recycling mechanism. In contrast, deletion of Ers1 in *S. cerevisiae* did not lead to cystine accumulation (Simpkins et al, 2016), and we obtained similar results in *P. pastoris* strains lacking Ers1S, Ers1L, or both, even under nitrogen starvation conditions. These results suggest that cystinosin's role in vesicular trafficking may be its primary function rather than serving as a cystine transporter.

This study focused on the human NHE isoforms expressed at the plasma membrane, specifically examining their role in maintaining sodium homeostasis in the kidney such as NHE2 and NHE3. Intracellular isoforms, such as the vacuolar and endosomal members (e.g., NHE6 and NHE8), which are primarily involved in pH and ion balance, were not included in our investigation (Bobulescu et al, 2005). Surprisingly, while both cystinosin and cystinosinLKG interacted with NHE3, rescue of the expression of *CTNS*, but not *CTNS-LKG*, was able to restore normal trafficking of NHE3 in cystinosis patient PTCs and improve sodium and albumin uptake in CTNS-deficient PTC lines. The specific role of cystinosinLKG is unknown. This isoform was shown to decrease cystine accumulation and apoptosis in Ctns-deficient PTCs (Taranta et al, 2017). In tissues, *CTNS-LKG* mRNA was found to be expressed at a level included between 5–20% of the canonical

isoform in most tissues except in testis, in which *CTNS-LKG* mRNA level is similar to *CTNS* (Taranta et al, 2012). In this study, we showed that only cystinosin is involved in NHE3 trafficking, and thus, while both isoforms are involved in cystine transport (Taranta et al, 2008; Bellomo et al, 2016), they have distinct functions for cellular trafficking that could be specific to proteins or cell types. Further studies will be required to determine the specific roles of cystinosin versus cystinosinLKG in each cell type.

We confirmed the abnormal NHE3 localization in the PTCs in the murine *Ctns*⁻/⁻ kidneys and cystinosis patient kidney biopsies. Loss of apical multi-ligand endocytic receptors, megalin, and cubilin, has also been observed in *Ctns*⁻/⁻ mice (Gaide Chevronnay et al, 2014; Raggi et al, 2014), and was confirmed in our study by western in the *CTNS*⁻/⁻ HK2 cells upon albumin stimulation. Under normal physiological conditions, NHE3 specifically interacts with megalin in proximal tubules, participating in megalin/cubilin-mediated endocytosis (Gekle et al, 1999, 2004; Biemesderfer et al, 1999). Megalin and cubilin facilitate receptor-mediated endocytosis of low-molecular-weight proteins in the kidney and are expressed at the renal brush border (Christensen and Birn, 2002; Leheste et al, 1999; Sun et al, 2017). Therefore, abnormal NHE3 trafficking to the brush border in cystinosis models could also help explain the loss of megalin and cubilin, accounting for the overall water, small proteins and solute loss associated with the renal FS. In addition, several reports suggested absorptive functions for both NHE3 and NHE2 isoforms in intestinal and renal proximal tubule brush border, with NHE3 predominantly assuming the major role (Orlowski and Grinstein, 2004; Nwia et al, 2022; Hoogerwerf et al, 1996). We observed an increased expression of NHE2 in *CTNS*-deficient PTCs and in *Ctns*⁻/⁻ kidney, probably acting as a compensatory mechanism to the loss of NHE3 at the apical membrane. Increased NHE2 protein is observed in both CTNS-deficient PTCs and *Ctns*⁻/⁻ kidneys, but elevated NHE2 mRNA is detected only in CTNS-deficient PTCs, likely due to mRNA detection sensitivity between isolated cells and the more complex, heterogeneous whole kidney tissue. Nonetheless, the contribution of NHE2 to absorption, if any, is not significant enough to compensate for NHE3 loss (Ledoussal et al, 2001b, 2001a). Interestingly, in CTNS-deficient PTCs, NHE3 mRNA levels remain unchanged; however, the reduced protein levels suggest that mislocalized NHE3 may undergo enhanced degradation through the ER-associated degradation (ERAD) pathway, likely as a result of its retention in the endoplasmic reticulum.

The current therapy for cystinosis, cysteamine, which allows lysosomal cystine clearance, delays disease progression but fails to correct the Fanconi syndrome. Some recent studies indicate that starting cysteamine treatment immediately after birth, particularly in asymptomatic infants, may slow the progression of renal Fanconi syndrome. However, these observations are largely anecdotal and require further investigation to determine the optimal timing for treatment initiation, overall drug tolerance, compliance, and the long-term effectiveness of the therapy in mitigating the severity of Fanconi syndrome and kidney dysfunction (Hohenfellner et al, 2022, 2019). Our study supports that the renal FS in cystinosis is due to cystinosin-specific function in the PTCs, and not due to the toxic impact of the cystine storage. Therefore, the treatment of the FS in cystinosis requires the restoration of cystinosin in the PTCs. We previously demonstrated that stable engraftment of bone

marrow-derived cells from *Ctns*-expressing HSPC transplantation in *Ctns*⁻/⁻ mice resulted in a reduction in cystine content in tissues and long-term preservation of the kidney, and improvement of the renal FS (Syres et al, 2009; Yeagy et al, 2011). The primary mechanism underlying this therapeutic effect involved the formation of TNTs by HSPC-derived macrophages, facilitating the transfer of healthy lysosomes containing functional cystinosin to the diseased host cells, including the proximal tubular cells. Notably, these TNTs enable bidirectional transport, transferring cystinosin-containing lysosomes into affected cells while retrieving cystine-loaded lysosomes for degradation within the macrophages (Naphade et al, 2015; Goodman et al, 2019). This reciprocal exchange effectively reduces cystine accumulation, restores lysosomal function, and preserves proximal tubular architecture and kidney function over time (Syres et al, 2009; Yeagy et al, 2011). In this study, our results showed that NHE3 localization at the brush border was significantly improved in the *Ctns*⁻/⁻ mice transplanted with WT HSPCs, along with improved renal FS and function, further supporting restoration of cystinosin protein in the PTCs of the treated mice. The knowledge obtained from this study will also be pertinent to the ongoing stem cell gene therapy clinical trial for cystinosis as well as other rare kidney diseases (Khare and Cherqui, 2024).

Renal Fanconi syndrome is a multifactorial condition characterized by generalized dysfunction of the proximal tubules, leading to impaired reabsorption of various substances. While NHE3 plays a role in sodium and bicarbonate reabsorption, its dysfunction alone may not fully explain the spectrum of FS pathology. Indeed, in mice and patients with genetic loss-of-function mutations in *NHE3*, no Fanconi syndrome has been described (Schultheis et al, 1998; Downie et al, 2021) supporting the fact that cystinosin mediated FS is a multifactorial implication rather than a single gene effect. As such, novel roles of cystinosin in PTCs such as its involvement in autophagy (Andrzejewska et al, 2016; Luciani et al, 2018; Festa et al, 2018; Berquez et al, 2023; Napolitano et al, 2015), TFEB expression (Rega et al, 2016), and inflammation (Lobry et al, 2019) has been identified. In addition, loss of expression and mislocalization of proteins like Sodium-Glucose Cotransporter 2 (*SGLT2*), megalin (*LRP2*), and Type IIa Sodium-Dependent Phosphate Transporter (*NPT2*) have been reported, though the mechanisms have not been fully explored (Gaide Chevronnay et al, 2014). Interestingly, NHE3 interacts directly or indirectly with these proteins (Tanimura et al, 2011; Bachmann et al, 2004; Biemesderfer et al, 1999; Pessoa et al, 2014). Restoring cystinosin function may alleviate these effects by stabilizing NHE3 localization and function, potentially addressing one of the primary causes of FS. However, while the restoration of NHE3 expression following *CTNS* supplementation is an intriguing finding, it does not prove that NHE3 dysfunction is the sole factor contributing to the development of FS in cystinosis. The relationship may be indirect or part of a broader network of cellular changes.

In summary, cystinosin interacts with NHE3 and plays a crucial role in its expression, trafficking, and function. Given that NHE3 has an essential role in the PTCs, interacting directly and indirectly with several transporters, this novel finding provides a mechanism underlying the renal Fanconi syndrome in cystinosis. These findings may lead to novel therapeutic approaches for cystinosis and provide valuable insights into the current HSPC gene therapy clinical trial for cystinosis.

# Methods

### Reagents and tools table

| Reagent/resource | Reference or source | Identifier or catalog number |
|---|---|---|
| **Experimental models** | | |
| **Mice** | | |
| C57BL/6 *Ctns*$^{-/-}$ | Cherqui et al, 2002; Nevo et al, 2010 | N/A |
| C57BL/6-Tg (CAG-EGFP) 1Osb/J | The Jackson Laboratory | 003291 |
| **Mammalian cells** | | |
| Human N and CT PTCs | Dr. Corinne Antignac | N/A |
| HK-2 and *CTNS*$^{-/-}$ HK-2 cells | Dr. Sergio Catz (Zhang et al, 2019) | N/A |
| HEK293T | ATCC | CRL-3216 |
| **Recombinant DNA** | | |
| *Lentiviral Packaging Plasmids* (pMDLg/pRRE, pHCMV-g, pRSV-Rev) | Harrison et al, 2013 | N/A |
| *pCCL-CTNS-DsRed* | Harrison et al, 2013 | N/A |
| *pCCL-CTNS-LKG-DsRed* | Harrison et al, 2013 | N/A |
| *pCCL-NHE1-GFP* | This study | N/A |
| *pCCL-NHE2-GFP* | This study | N/A |
| *pCCL-NHE3-GFP* | This study | N/A |
| *Pichia pastoris* strains | This study (Table 1) | N/A |
| *Pichia pastoris* plasmids | This study (Table 2) | N/A |
| **Primers** | IDT | This study (Table 3) |
| NHE-3 shRNA (h) Lentiviral Particles | Santa Cruz Biotechnology | sc-36059-V |
| **Antibodies** | | |
| Anti-Mouse IgG HRP Conjugate | Promega | W402B |
| Anti-Rabbit IgG HRP Conjugate | Invitrogen | G21234 |
| Anti-Rabbit IgG Alexa Fluor Plus 647 | Invitrogen | A32733 |
| Anti-Rat IgG Alexa Fluor Plus 647 | Invitrogen | A48272 |
| Anti-Rabbit IgG Alexa Fluor 488 | Invitrogen | A11008 |
| Anti-Rat IgG Alexa Fluor Plus 488 | Invitrogen | A11006 |
| Anti-LAMP1 | Santa Cruz Biotechnology | sc-19,992 |
| Anti-EEA1 | BD Transduction Laboratories™ | 610457 |
| Anti-GM130 | BD Transduction Laboratories™ | 610822 |

| Reagent/resource | Reference or source | Identifier or catalog number |
|---|---|---|
| Anti-GFP | SICGEN | AB0020-200 |
| Anti-GFP | Abcam | Ab290 |
| Anti-SERCA2 ATP2A2 | Proteintech | 27311-1-AP |
| Rat anti-NHE3 | BiCell Scientific | 21003rs |
| Rabbit anti-NHE3 | BiCell Scientific | 21003 |
| Anti-NHE3 | Millipore | AB3085 |
| Anti-Megalin | Proteintech | 19700-1-AP |
| Anti-NHE2 antibody | Novus | NBP2-38236 |
| Anti-NHE1 antibody | Proteintech | 67363-1-1g |
| Anti-GAPDH antibody | Proteintech | HRP-60004 |
| Anti-RFP | Proteintech | 6g6 |
| Normal mouse IgG | Santa Cruz Biotechnology | Sc-2025 |
| Normal rabbit IgG | Cell Signaling Technology | 2729S |
| **Chemicals & other reagents** | | |
| Annexin-binding buffer 5X concentrate | Invitrogen | V13246 |
| Dynabeads Protein A | Invitrogen | 10001D |
| Cysteamine | Sigma | 30070 |
| EIPA (5-(*N*-ethyl-*N*-isopropyl)-amiloride | Sigma | A3085 |
| Polybrene | Santa Cruz Biotechnology | sc-134220 |
| Citrate Unmasking Solution (10X) | Cell Signaling Technology | 14746 |
| Pierce ECL Western Blotting Substrate | Thermo Scientific | 32209 |
| Pierce RIPA Buffer | Thermo Scientific | 89901 |
| GFP-Trap Magnetic Agarose | Chromotek | gtma-20 |
| Sodium Green™ Tetraacetate | Invitrogen | S6901 |
| Alexa Fluor 555-conjugated albumin | Invitrogen | A34786 |
| AnnexinV | Invitrogen | 331200 |
| 7-amino-actinomycin D (7-AAD) | Invitrogen | 00-6993-50 |
| Puromycin dihydrochloride | Santa cruz | sc-108071 |
| Kidney FFPE tissue (non-cystinosis healthy donor) | Zyagen | HP-901 |
| Miltenyi LS columns | Miltenyi Biotec | 130-042-401 |
| Anti-Sca-1 MicroBead Kit | Miltenyi Biotec | 130-123-124 |
| Lotus Tetragonolobus Lectin (LTL) Rhodamine conjugate | BiCell Scientific | 00010R |
| ER-Tracker Red Dye | Invitrogen | E34250 |
| iScript Reverse Transcription Supermix for RT-qPCR | BIORAD | 1708841 |
| iTaq Universal SYBR Green Supermix | BIORAD | 1725122 |
| 4-Sulfonato calix [8] arene, sodium salt | BIOTIUM | 70037 |

| Reagent/resource | Reference or source | Identifier or catalog number |
|---|---|---|
| **Assays** | | |
| QuantiChrom Creatinine Assay Kit | BioAssay Systems | DICT-500 |
| EnzyChrom Glucose Assay Kit II | BioAssay Systems | EGL2-100 |
| Sodium Assay Kit | Abcam | ab211096 |
| **Software** | | |
| ImageJ | NIH | https://imagej.nih.gov/ij/ |
| ImagePro Premier | Media Cybernetics | N/A |
| Graphpad Prism v.10 | Graphpad | N/A |
| Fiji Image Analysis | NIH | https://imagej.net/Fiji |
| AxioVision software V4.8.2.0 | Carl Zeiss Microscopy | |
| GelQuant.Net | Biochem Lab Solutions | http://biochemlabsolutions.com/GelQuantNET.html |
| BD software | BD Biosciences | N/A |
| NIS-Elements software | Nikon | N/A |
| Imaris (version 7.0) | Bitplane Scientific Software | N/A |

## Mice

All mice used in this study were of the C57BL/6 background. The C57BL/6 $Ctns^{-/-}$ transgenic mouse model was generously provided by Dr. Corinne Antignac (Inserm U983, Paris, France) (Cherqui et al, 2002; Nevo et al, 2010). Genotyping of the mice was conducted using standard PCR reactions with primers detailed in Table 3. For hematopoietic stem/progenitor cell transplantation experiments, GFP transgenic mice expressing enhanced GFP cDNA under the control of the chicken beta-actin promoter (C57Bl/6-Tg (CAG-EGFP) 1Osb/J; 003291, Jackson Laboratory) were utilized as donors. The mice were housed in a controlled environment with regulated temperature and humidity, maintained on a 12-h light/dark cycle, and provided free access to water and food. Both male and female mice were included in the experiments. Each experimental and control group comprised 6–10 mice, analyzed at approximately 8 months of age. All mice were bred at the University of California, San Diego (UCSD) vivarium, and all experimental procedures were conducted in accordance with protocol (S12288) approved by the UCSD Institutional Animal Care and Use Committee.

## Mammalian cells

Human N and CT PTCs were generously provided by Dr. Corinne Antignac. They are transformed PTCs isolated from the urine of a healthy donor (N), and from a cystinosis patient (CT) that has a 57 kb deletion in the heterozygous state and a G >T transition at the donor splice site of exon 7 (800 G >T) (Kalatzis et al, 2002). HK-2 (proximal tubule cells) and $CTNS^{-/-}$ HK-2 cells were provided by Dr. Sergio Catz (Zhang et al, 2019). $CTNS^{-/-}$ HK-2 cells had

elevated cystine levels (Zhang et al, 2019). The human cell line HEK293T (embryonic kidney), obtained from ATCC was used for producing lentiviral particles for LV-CTNS-DsRed; LV-CTNS-LKG-DsRed, LV-NHE1-GFP, LV-NHE2-GFP, and LV-NHE3-GFP. HEK293T and $CTNS^{-/-}$ HK-2 cells were used for immunoprecipitation studies. NHE3 knockdown cells were generated using NHE-3 shRNA (h) lentiviral particles (Santa cruz, sc-36059-V). Briefly, after one day of culturing the cells, polybrene (sc-134220) was added to the culture medium at a final concentration of 5 μg/ml, followed by the addition of viral particles at a multiplicity of infection (MOI) of 40 for 24 h. Two days after transduction, cells were split at a 1:5 ratio, and 24 h later, puromycin dihydrochloride (sc-108071) was applied at 3.5 μg/ml to select for stably transduced cells. NHE3 knockdown efficiency was confirmed by quantitative PCR using the primers listed in Table 3.

## Human kidney biopsy

Kidney biopsy from a patient affected with cystinosis was obtained and the studies have been approved by the institutional review board committee (IRB #141150) and have been performed in accordance with the ethical standards. Patient signed informed consent. The biopsy was provided as formalin-fixed paraffin-embedded (FFPE) blocks. Tissue from non-cystinosis healthy donor was obtained from Zyagen (HP-901). 5-μm sections were cut and stained by rabbit/rat anti-NHE3 (1:50; BiCell Scientific), LTL Rhodamine conjugate (1:100; BiCell Scientific), rabbit anti-SERCA2- ATPase (1:30; Proteintech) and DAPI stain.

## Lentiviral particle production

HEK293T cells line was cultured in DMEM supplemented with 10% heat inactivated FBS, with a prescribed dose of Penicillin/Streptomycin (Invitrogen, Life Technologies) added as recommended antibiotics. All cells were maintained at 37 °C in a humid incubator with 5% $CO_2$. Lentiviral particles were produced using the 3 packaging plasmids (pMDLg/pRRE, pHCMV-g, pRSV-Rev) (Harrison et al, 2013) along with the plasmid carrying CTNS-DsRed, CTNS-LKG-DsRed, and NHE1-GFP, NHE2-GFP, NHE3-GFP and were transfected using calcium phosphate protocol (Harrison et al, 2013). Vector particles were harvested in media and concentrated through ultracentrifugation at 25,000 rpm for 2 h at 4 °C. The titers of the concentrated virus, finally dissolved in Stemspan medium (StemCell Technologies, Vancouver, British Columbia, Canada), were determined by infecting HEK293T cells with serial dilutions of the virus preparations and evaluating them via flow cytometry and droplet digital (dd) PCR. Subsequently, the virus was introduced to HEK293T and maintained for 14 days to establish the stable cell lines. The GFP and DsRed positive cells were selected using fluorescence-activated cell sorting (FACS). NHE3 shRNA knockdown cells were prepared using NHE-3 shRNA (h) Lentiviral particle following manufacturer's protocol (Santa cruz, sc-36059-V).

## Yeast studies

### Yeast cells
The studies were carried out using methylotrophic yeast *Pichia pastoris*.

**Table 1.** *Pichia pastoris* strains.

| Description | Strain | Genotype | Reference |
|---|---|---|---|
| WT | PPY12 | *his4, arg4* | (Gould et al, 1992) |
| WT + Sec7-DsRed | PPY12 | PPY12 Sec7-DsRed::*SEC7*(*HIS4* selection) *arg4 his4* | (Soderholm et al, 2004) |
| Δers1S | Sjcf1545 | PPY12 Δers1S::*ZEOCIN arg4 his4* | This study |
| Δers1L | Sjcf1519 | PPY12 Δers1L::*KanMX arg4 his4* | This study |
| Δers1S Δers1L | Sjcf1546 | PPY12 Δers1S::*ZEOCIN* Δers1L::*KanMX arg4 his4* | This study |
| Δypt7 | Srrm197 | PPY12 Δypt7::*KanMX arg4 his4* | (Manjithaya et al, 2010) |
| Δvps1 | Sjcf1614 | PPY12 Δvps1::*ZEOCIN arg4 his4* | This study |
| Δvps15 | OP5 | GS200 Δvps15::*ScARG4 arg4 his4* | (Stasyk et al, 1999) |

**Table 2.** *Pichia pastoris* plasmids.

| Plasmid | Promoter | Fusion protein | Integration locus | Selectable marker |
|---|---|---|---|---|
| pJCF728 | *ERS1S* | Ers1S-GFP+ | *ARG4* | *ARG4* |
| pJCF155 | *NHX1* | Nhx1-GFP | *ARG4* | *ARG4* |
| pTA12 | *VPS8* | Vps8-2xmCherry | *VPS8* | *NAT* |
| pTA8 | *NHX1/ERS1S* | Nhx1-VC/Ers1S-VN | *ARG4* | *ARG4* |
| pTA3 | *NHX1* | Nhx1-VC | *HIS4* | *HIS4* |
| pTA2 | *ERS1S* | Ers1S VN | *HIS4* | *HIS4* |

Strains and plasmids used are shown in Tables 1 and 2, respectively. Medium used in this study: YPD (2% glucose, 2% bacto-peptone, 1% yeast extract), YNB (0.17% yeast nitrogen base without amino acids and 0.5% ammonium sulfate), YNB-N (0.17% yeast nitrogen base without amino acids and ammonium sulfate), CSM (complete synthetic medium of amino acids and supplements), glucose medium (2x YNB, 0.79 g/L CSM, 0.04 mg/L biotin, 2% dextrose), SD-N or starvation medium (1x YNB-N and 2% dextrose).

Extraction of intracellular cystine.    Samples (50 ml) of cultures were harvested by centrifugation at $2000 \times g$ and washed twice in 100 mM phosphate buffer (pH 7.5). The cells were suspended in 5% trichloroacetic acid (TCA) in water and broken with 0.5-mm glass beads. Cell debris was removed by centrifugation ($10,000 \times g$) at 4 °C, and the supernatant was stored at −20 °C for MS analysis of cystine content.

Fluorescence Microscopy.    Cells were grown in YPD at 30 °C until exponential phase (1–2 OD600/mL), washed twice with sterile water, and then transferred to glucose medium for 6 h. Mid-log cells were then pelleted, 1.5 μl of cells was mixed with 1% low melting point agarose and placed on a glass slide with a coverslip and imaged using 63× or 100× magnification on a Carl Zeiss Axioskop 2 MOT microscope (Carl Zeiss Microscopy, Gottingen, Germany). Images were taken on an AxioCam HRm digital camera (Carl Zeiss MicroImaging GmbH, Gottingen, Germany); no digital gain was used, exposure was adjusted as needed, except for BiFC which was kept constant during microscopy in different strains. Images were processed using AxioVision software V4.8.2.0 (Carl Zeiss Microscopy, White Plains, NY, USA). The images are representative results from experiments conducted at least in triplicate. Methodology to

determine late endosome (LE) or trans-Golgi network/early endosome (TGN/EE) localization in different background strains is as follows: Cells showing distinct LE labeling with Vps8-2xmCherry or TGN/EE labeling with Sec7-DsRed were first marked in the red channel in the AxioVision software. Then, these marks were analyzed for colocalization in the green channel with the GFP-tagged proteins or fluorescence obtained from Bimolecular Fluorescence Complementation (BiFC). FM4-64 staining was visualized after a short pulse (~3min) followed by a chase in the presence of the quencher (4-Sulfonato calix [8] arene, sodium salt) SCAS (BIOTIUM cat#70037).

## Immunoprecipitation (IP) assay

HEK293T and CTNS-deficient HK-2 stably transduced with various combinations of LV-CTNS-DsRed; LV-CTNS-LKG-DsRed, LV-NHE1-GFP, LV-NHE2-GFP, and LV-NHE3-GFP, were cultured and harvested from 15 cm dish plates. Whole cell extracts were prepared using IP Lysis Buffer (50 mM Tris-HCL pH 7.5, 150 mM NaCl, 1 mM EDTA, 1% Igepal CA-630, 10% Glycerol, 0.5 mM DTT) supplemented with freshly added PIC, PMSF & DTT. Supernatants were collected by centrifugation at $13,000 \times g$ for 10 min at 4 °C. Protein conc. was measured by Pierce BCA Protein Assay Kit. 10% Input was kept at −80 °C for later use. Prior to immunoprecipitation (IP), lysates underwent pre-clearing via incubation with Dynabeads at 4 °C for 30 min. GFP pull-down was executed using GFP-Trap Magnetic Beads (Chromotek-gtma), while DsRed pull-down utilized Protein A Dynabeads pre-coupled with RFP Antibody (Chromotek-6G6) at 4 °C for 2–4 h. The pre-coupled beads were then incubated with the 500ug-1mg protein lysate at 4 °C O/N. Next day, the tubes were replaced, and the beads were washed with IP Lysis Buffer followed by IP Wash Buffer (50 mM Tris-HCL pH 7.5, 200 mM NaCl, 1 mM EDTA, 1% Igepal CA-630, 10% Glycerol, 1 mM DTT) (with freshly added PIC, PMSF & DTT). Elution of proteins was achieved using 30 μl of 2X protein loading dye. Co-immunoprecipitated proteins were resolved by SDS-PAGE and probed for GFP (Abcam, ab290) and DsRed (Chromotek-6G6). Pulldown using normal mouse IgG (sc-2025) or normal rabbit IgG (CST-2729) was used as a negative control.

## Immunoblotting

Cells and murine kidney tissues were homogenized in Pierce RIPA buffer (Sigma, St Louis, MO) supplemented with Protease Inhibitor

**Table 3.  Primer sequences for qPCR.**

| Gene | Full name | Purpose | Direction (5'–3') | Sequence |
|---|---|---|---|---|
| Human primers | | | | |
| *GAPDH* | Glyceraldehyde-3-phosphate dehydrogenase | qPCR Housekeeping | Forward<br>Reverse | TCAAGGCTGAGAACGGGAAG<br>CGCCCCACTTGATTTTGGAG |
| *CTNS* | Cystinosin | mRNA Expression | Forward<br>Reverse | TCCTCCTGTCGTAAAGCTGGA<br>GCCGGTCTGATTGGAGTGAT |
| *NHE3* | Na + /H+ exchanger 3 | mRNA Expression | Forward<br>Reverse | GTCACCGTGGTTCTGTACAAT<br>CACCACGAAGAAGGACACTATG |
| *NHE2* | Na + /H+ exchanger 2 | mRNA Expression | Forward<br>Reverse | TCAAATCCGTCAGCGAACTTT<br>TTCTCGCAAACTGTGTCGCC |
| *NHE1* | Na + /H+ exchanger 1 | mRNA Expression | Forward<br>Reverse | CACTGTCATCTTCTTCACCGTC<br>ATCTCTTCGTTGATGGAGCG |
| Murine primers | | | | |
| *Gapdh* | glyceraldehyde-3-phosphate dehydrogenase | qPCR Housekeeping | Forward<br>Reverse | GCACAGTCAAGGCCGAGAAT<br>GCCTTCTCCATGGTGGTGAA |
| *Ctns* | Cystinosin | mRNA Expression | Forward<br>Reverse | CCACATGGCTCCAGTTCCTCT<br>CACACCGCCAATGCTCCAG |
| *Nhe3* | Na + /H+ exchanger 3 | mRNA Expression | Forward<br>Reverse | ATGCAGTGACTGTGGTCTTG<br>CACCACGAAGAAGGACACTATG |
| *Nhe2* | Na + /H+ exchanger 2 | mRNA Expression | Forward<br>Reverse | ATAAGGAAGCTCACGCCAG<br>CCGTCAGGTTGTGTCTGTTAT |
| *Ctns* WT | Cystinosin | Genotyping | Forward<br>Reverse | CTCCAGATGTTCCTCCAGTC<br>AGTCCGAACTTGGTTGGGT |
| *Ctns* mutant | Cystinosin | Genotyping | Forward<br>Reverse | GCAGGAATTCGATATCAAGC<br>AAAGTGGAGGTAGGAAAGAGG |

All primers were reconstituted at 100 µM and used at a working concentration of 5 µM.

**Table 4.  Donor-derived HSPC engraftment in *Ctns*$^{-/-}$ mice transplanted with GFP$^+$ HSPCs.**

| Mice (*Ctns*$^{-/-}$) | Gender | Engraftment of GFP$^+$ cells in blood (%) |
|---|---|---|
| 1. | Female | 76.5 |
| 2. | Male | 69.7 |
| 3. | Male | 73 |
| 4. | Female | 77.2 |
| 5. | Female | 6.9 |
| 6. | Male | 57.6 |
| 7. | Male | 57.3 |
| 8. | Female | 73.7 |
| 9. | Male | 70.7 |
| 10. | Male | 49.3 |

Cocktail (Sigma, St Louis, MO). Protein concentrations were determined using the Pierce BCA Protein Assay Kit and 30-50 µg of proteins were loaded onto SDS-PAGE gels for subsequent immunoblotting following standard protocols. The proteins transferred onto the PVDF membrane was incubated with the primary antibodies at a 1:1000 dilutions in 5% BSA in TBS-T, with overnight incubation at 4 °C for primary antibodies and 1-h incubation at room temperature for secondary antibodies. The primary antibodies used were rabbit anti-NHE3 (Millipore, AB3085), rabbit anti-GFP (Abcam, Ab290), rabbit anti-Megalin (Proteintech, 19700-1-AP), rabbit anti-NHE2 antibody (Novus,

NBP2-38236), mouse anti-NHE1 antibody (Proteintech, 67363-1-1g) and mouse anti-GAPDH antibody (Proteintech, HRP-60004) followed by goat anti-mouse or anti-rabbit horseradish peroxidase-conjugated secondary antibodies. Blots were developed using Enhanced Chemiluminescence (ECL) reagent according to the manufacturer's protocol (GE Healthcare, Pittsburgh, USA) and imaged using the Azure 600 imager (Azure Biosystems, Dublin, CA, USA). Quantification of bands was carried out using GelQuant.Net software.

## RNA extraction and real-time quantitative PCR (RT-qPCR)

Total RNA was extracted from both cells and mice kidney (after homogenization in Precellys 24) employing the RNeasy Mini Kit as per the manufacturer's protocol (Qiagen, Hilden, Germany, cat. 74104). Subsequently, 500 ng of RNA was transcribed into cDNA using iScript cDNA Synthesis Kit (Bio-Rad, Hercules, CA, cat. 1708840). For the RT-qPCR reaction setup, 5 µl of iTaq Universal SYBR Green Supermix (Bio-Rad, Hercules, CA, cat.1725121), 3 µl of 1:10 diluted cDNA (2.5 ng/µl), and 1 µl of forward and reverse primer (5 µM each) were combined. The reaction was performed on a CFX96 thermocycler (Bio-Rad) under the following conditions: 95 °C (30 s); 40 cycles of 95 °C (5 s) and 60 °C (30 s); then 65 °C (5 s); and 95 °C (5 s). Gene expression was quantified utilizing the ΔΔCt method relative to the wild-type (WT) and normalized to the endogenous control (GAPDH). All primer sequences are shown in Table 3.

## Determination of intracellular sodium

To measure intracellular sodium concentration [Na$^+$], we used the sodium ion fluorescence indicator Sodium Green™ Tetraacetate (Invitrogen, Catalog #S6901) (Amorino and Fox, 1995). We seeded $0.6 \times 10^6$ cells in 24-well plates overnight. The next day, the cells were washed twice with PBS and incubated with Sodium Green™ Tetraacetate at a final concentration of 2 µM in dimethyl sulfoxide for 8 min. After incubation, the cells were washed three times with PBS to remove excess dye and processed for flow cytometry (BD Accuri C6, BD Biosciences). AAD-7 was used to exclude dead cells. All flow cytometric analyses were performed using BD software. We also conducted this assay following cells treated with NHE3 inhibitor EIPA (5-(N-ethyl-N-isopropyl)-amiloride (Sigma) at a concentration of 100 µM for 4 h. DMSO was used as the vehicle control for EIPA treatment with a final concentration of 0.1%.

## Albumin uptake assay

To assess the cells' ability to uptake albumin, $0.6 \times 10^6$ cells were seeded in a 24-well plate overnight. The next day, the cells were washed twice with PBS and incubated with 50 µg/mL Alexa Fluor 555-conjugated albumin (Invitrogen, Cat #A34786) for approximately 16 h for complete intracellular uptake and processing of albumin. This was also done in time course for the following timepoints 0.5, 1, 3, 6, 12, and 16 h. After incubation, the cells were washed three times with PBS to remove excess dye and processed for flow cytometry (BD Accuri C6, BD Biosciences). AnnexinV (Invitrogen, Cat #331200) & 7-amino-actinomycin D (7-AAD) (Invitrogen, Cat # 00-6993-50) was used to exclude dead cells. All flow cytometric analyses were conducted using BD software. We also conducted this assay following cells treated with NHE3 inhibitor EIPA (5-(N-ethyl-N-isopropyl)-amiloride (Sigma) at a concentration of 100 µM for 4 h.

## Total internal reflection fluorescence microscopy (TIRFM) and data analysis

For live-cell TIRFM imaging, human PTCs were seeded in 8-well plates with coverglass bottoms (Lab-Tek borosilicate, Nunc, Thermo) in phenol red-free RPMI medium. Cells were placed on a pre-warmed microscope stage, and imaging was done using a 100× 1.45 numerical aperture (NA) TIRF objective on a custom-modified Nikon TE2000U microscope with TIRF illumination. Laser illumination (488 and 543 nm) was angled to create an evanescent field depth of <100 nm. Images were captured on a 14-bit cooled CCD camera (Hamamatsu) controlled by NIS-Elements software (Nikon) at 1-s intervals with 200–600 ms exposure times. Analysis was performed using ImageJ (version 1.43) and Imaris (version 7.0, Bitplane Scientific Software). Granule movement was tracked across all movie frames, including vesicles visible in the TIRFM zone for at least three frames. Images with mild fading were auto-thresholded in Imaris to ensure consistent tracking.

## Hematopoietic stem and progenitor cell (HSPC) isolation, transplantation, and engraftment

Bone marrow cells were extracted from the femurs of 6- to 8-week-old $Ctns^{-/-}$ or WT-GFP$^+$ transgenic mice. HSPCs were isolated through immunomagnetic separation utilizing an anti-Sca1 antibody linked to magnetic beads (Miltenyi Biotec). The $\sim 2 \times 10^6$ Sca1$^+$ HSPCs cells suspended in 100 µL of phosphate-buffered saline (PBS) were then directly injected via tail vein into previous day lethally irradiated (7 Gy; X-Rad 320, PXi) $Ctns^{-/-}$ mice. In the case of mice receiving WT GFP$^+$ HSPCs, engraftment of the transplanted cells was assessed in peripheral blood two months post-transplantation. Blood samples obtained from the tails were treated with red blood cell lysis buffer (eBioscience) and subsequently examined using flow cytometry (BD Accuri C6, BD Biosciences) to ascertain the proportion of GFP$^+$ cells. Blood engraftment % is presented in Table 4.

## Immunofluorescence, image acquisition and analysis

Murine kidney tissue sections were collected from euthanized mice, fixed using 10% neutral buffered formalin (NBF), and then embedded in paraffin wax. Standard methods were employed to section the tissue at a thickness of 5 µm. After deparaffinization, the sections were transferred to pre-warmed antigen retrieval solution at 95 °C for 30 min, followed by cooling at room temperature for 20 min. Subsequently, the sections were placed in blocking solution (0.25% Triton X-100 and 3% BSA in tris-buffered saline) and then incubated overnight at 4 °C with primary antibodies: rabbit/rat anti-NHE3 (1:50; BiCell Scientific), rabbit anti-SERCA2-ATPase (1:50; Proteintech) and rabbit anti-GFP (1:500; Abcam). The next day, appropriate Alexa Fluor-conjugated secondary antibodies (Invitrogen) were added for antigen visualization, along with LTL Rhodamine conjugate (1:100; BiCell Scientific) and DAPI stain. Images were captured using a Keyence BZ-X710 digital microscope. ImagePro Premier software (Media Cybernetics) was utilized for all quantification with thresholding, determining the proportion of expression and colocalization (Pearson's correlation coefficient) for NHE3 and LTL/SERCA2-ATPase in z-stacks (multiple images taken per section) or stitched images. Regarding Immunocyto-chemistry (ICC), HEK293T cells were seeded in Ibidi 35-mm round-bottom plates (81158) overnight. The following day, cells were fixed with 4% paraformaldehyde, stained with DAPI, and imaged using a Nikon Eclipse Ti2-E microscope. Single-slice confocal images were acquired at the focal plane showing optimal spatial overlap between channels. All images were captured using identical acquisition settings across samples within each experiment and analyzed using Fiji Imagej.

NHE3-GFP transduced PTCs were seeded at 70% confluence in a 96-well plate with glass-like polymer bottom black frame (P96-1.5P, Cellvis), then fixed with 4% paraformaldehyde (Electron Microscopy Sciences, 15,710) for 8 min and blocked with 1% BSA (Rockland, BSA-50) in PBS (Corning, 21–031-CV), in the presence of 0.01% saponin (Calbiochem, 558,255), for 1 h. Samples were labeled with the indicated primary antibodies overnight at 4 °C in the presence of 0.01% saponin and 1% BSA. Samples were washed 3 times and subsequently incubated with the appropriate combinations of Alexa Fluor (488 or 594)-conjugated anti-goat, or anti-mouse secondary antibodies (Thermo Fisher Scientific, A-32814, A-21203, respectively). Nuclei were stained with Hoechst 33342 (DAPI; Millipore-Sigma, D9542) and samples were preserved in Fluoromount-G reagent (AnaSpec, AS-83218) and kept at 4 °C until analyzed. Samples were analyzed using a Zeiss LSM 880 or Zeiss LSM 980 laser-scanning confocal microscope attached to a

Axio Observer Z1 microscope at 21 °C, using a 63x oil Plan Apo, 1.4-numerical aperture objective. Images were captured using identical acquisition settings across all samples within each experiment and were imported into ZEN Black software (Zeiss LSM software) for colocalization analysis and processed using ImageJ and Adobe Photoshop. The laser power and gain were maintained throughout the experiments to analyze wild-type and *Ctns*$^{-/-}$ cells comparatively.

For confocal images, all were acquired using the full dynamic intensity range (1–16383/1–65535) of the specified fluorophores. To assess colocalization, we acquired single slice confocal images at the plane of cells displaying accurate spatial overlap between channels. For quantitative colocalization analysis, regions of interest (ROI) were manually drawn around individual cells expressing both markers, excluding background areas. The colocalization analysis was performed on at least 15–20 cells per condition across three independent experiments. Threshold values were set using automatic thresholding algorithms in Fiji ImageJ to distinguish true signal from background noise in each channel independently. Background thresholds were established using secondary antibody controls and further refined using ImageJ's threshold tool to distinguish specific signal from background. These thresholds were typically set between 500 and 800 (in the 16-bit dynamic range of 0–65536). The following antibodies were used for immunofluorescence in this study: anti-LAMP1 (Santa Cruz Biotechnology, sc-19,992); anti-EEA1 (BD Transduction Laboratories™, 610457); anti-GM130 (BD Transduction Laboratories™, 610822) goat anti-GFP (SICGEN, AB0020-200). We employed Manders' overlap coefficients to quantify the degree of colocalization between NHE3 and the respective markers (EEA1, GM130, LAMP1 or ER-tracker). The weighted colocalization coefficient (Mander's overlap coefficient) was calculated as MOC = $\sum i(Ri \times Gi)/\sqrt{(\sum iRi^2 \times \sum iGi^2)}$, where Ri and Gi represent the intensity values of corresponding pixels in the red and green channels.

### Blood and urine analysis

Serum was collected via retro-orbital bleeding or cardiac puncture, and 24-h urine samples were obtained using metabolic cages. Serum creatinine, and urinary glucose, creatinine, and sodium (Na$^+$) levels were measured using colorimetric assay kits according to the manufacturers' instructions (BioAssay Systems, Hayward, CA: Serum and Urine Creatinine, DICT-500; Urine Glucose, EGL2-100) and Abcam (Cambridge, MA: Sodium Assay Kit, ab211096).

### Statistical analysis

The statistically significant differences were determined using GraphPad software Prism v.10.0. Data presented as mean ± SEM or mean ± SD, is mentioned in the figure legend. *p* values less than or equal to 0.05 were considered statistically significant. Comparisons between 2 groups were performed using an unpaired or paired 2-tailed Student's t test. One-way or two-way analysis of variance (ANOVA) was used to compare multiple groups. For the mice studies, all cohorts in the experiments were age-matched and sex-balanced. Experimenters were blinded to the genotype of the samples as much as possible. We did not exclude any animals from our experiments.

### Graphics

The schematic in Fig. 5A and the synopsis image graphics were created with BioRender.com.

## Data availability

This study includes no data deposited in external repositories.

The source data of this paper are collected in the following database record: biostudies:S-SCDT-10_1038-S44319-026-00736-1.

## Peer review information

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

## Acknowledgements

The study was supported by funds from the Cystinosis Research Foundation, the California Institute for Regenerative Medicine (CIRM, CLIN2-11478 and TRAN1-13983), the National Institute of Health (NIH) R01-NS108965, R01NS135162, R01-AG086443 to SC; R01DK110162, and P01HL152958 to SDCatz, the MPS Society, and the Friedreich's Ataxia Research Alliance (FARA). The UCSD Neuroscience Microscopy Shared Facility was funded by grant P30-NS047101. The Automated Zeiss LSM 980 Airyscan 2 Multiscale super-resolution Confocal microscope was funded by an NIH S10 grant S10OD030417 to SDCatz. We gratefully acknowledge Dr. Corinne Antignac for generously providing the mouse model of cystinosis and sharing the human N and CT PTCs and Dr. Benjamin Glick for providing the yeast strain PPY12+Sec7-DsRed. We acknowledge the cystinosis patient for donation of the kidney tissue. We acknowledge Jay Sharma, and the students Samantha Diaz, Pruthva Mania, Andrea Kelly, Tariq Aniff, Nia Asbill and Junmyung Lee for their help on the project.

## Author contributions

**Veenita Khare**: Conceptualization; Data curation; Software; Formal analysis; Validation; Investigation; Visualization; Methodology; Writing—original draft; Writing—review and editing. **Jean-Claude Farré**: Conceptualization; Data curation; Formal analysis; Validation; Methodology; Writing—original draft; Writing—review and editing. **Mouad Ait Kbaich**: Data curation; Formal analysis; Methodology. **Céline J Rocca**: Conceptualization; Data curation. **Cynthia Tang**: Data curation. **Xuan Ma**: Data curation. **Kavya Biederman**: Data curation. **Ioli Mathur**: Data curation. **Rafael A Badell-Grau**: Data curation. **Anusha Sivakumar**: Data curation. **Rola Chen**: Data curation. **Sergio D Catz**: Conceptualization; Formal analysis; Supervision; Funding acquisition. **Stephanie Cherqui**: Conceptualization; Resources; Formal analysis; Supervision; Funding acquisition; Validation; Investigation; Writing—original draft; Project administration; Writing—review and editing.

Source data underlying figure panels in this paper may have individual authorship assigned. Where available, figure panel/source data authorship is listed in the following database record: biostudies:S-SCDT-10_1038-S44319-026-00736-1.

## Disclosure and competing interests statement

Stephanie Cherqui is a cofounder, shareholder, and a member of both the Scientific Board and board of directors of Papillon Therapeutics Inc. Stephanie Cherqui is also the Chair of the Scientific Review Board and a member of Board of Trustees of the Cystinosis Research Foundation. This work is covered in the patent entitled "Methods of Treating Lysosomal Disorders" (#US-2024-0009247-A1). Anusha Sivakumar is a consultant for Papillon Therapeutics, Inc and receives income. The terms of this arrangement have been reviewed and approved by the University of California San Diego in accordance with its conflict-of-interest policies. The remaining authors declare that the research was conducted in the absence of any commercial or financial relationships that could be construed as a potential conflict of interest.

# Expanded View Figures

**Figure EV1. Cross-species sequence alignment, structural models, interactions/colocalization analysis of Nhx1 and Ers1 and cystine level measurement.**

(A) Clustal Omega multiple sequence alignment of human cystinosin (Uniprot accession number A0A0S2Z3K3), cystinosinLKG (A0A0S2Z3I9), *P. pastoris* Ers1S (C4R5N7), *P. pastoris* Ers1L (C4R120), and *S. cerevisiae* Ers1 (P1761). Black boxes indicate amino acid identity, gray boxes indicate amino acid similarity, with the red-boxed area highlighting the comparison of the seven-transmembrane domains of cystinosin across the different organisms. We used BoxShade for highlighting (https://junli. netlify.app/apps/boxshade/#forms::boxshade). The multiple sequence alignment was performed using Clustal Omega from EMBL (https://www.ebi.ac.uk/jdispatcher/ msa/clustalo). The threshold for comparison is set by default to greater than 50%. (B) Displays images of the AlphaFold-predicted structures of the various cystinosin isoforms in each organism described in (A). (C) BiFC assay performed in *P. pastoris* displaying an interaction between Ers1s and Nhx1, and their colocalization with the early endosome marker, Sec7-DsRed (TGN/EE) (white arrow). Negative controls, the single Venus moieties (Ers1S-VN and Nhx1-VC) do not fluoresce in the green or red channel (yellow arrows). Scale bars: 5 µm. (D, E) Violin plots of puncta per cell for Nhx1-GFP (D) and Vps8-2xmCherry (E) in WT, $\Delta vps1$, $\Delta vps15$, $\Delta ypt7$, $\Delta ers1S$ and $\Delta ers1L$. Each dot is one cell; violins show the distribution. Yellow triangles mark cells that display an Nhx1 vacuolar rim ("ring") phenotype (continuous or near-continuous Nhx1 signal outlining the vacuole membrane; scored as a binary per-cell attribute). Rim-positive cells still contribute puncta counts as usual; the ring itself is not counted as additional puncta. (F) Percentage of Nhx1 puncta that touch a Vps8 punctum in the same cell (Nhx1 on endosome). (G) Percentage of Vps8 puncta that touch an Nhx1 punctum (Vps8 with Nhx1). Touching/colocalization was defined operationally on 2D widefield images as overlap or edge-to-edge contact by ≥1 pixel between binary masks after channel alignment; no z-stacks were used. Values are per cell. For (D–G) Red squares with error bars = mean ± SEM. Sample sizes (cells): WT $n = 50$, $\Delta vps15$ $n = 50$, $\Delta vps1$ $n = 50$, $\Delta ypt7$ $n = 36$, $\Delta ers1S$ $n = 50$, $\Delta ers1L$ $n = 50$ (≥2 independent experiments). (H) Cystine content was measured by mass spectrometry in WT, $\Delta ypt7$ yeast strains and in Ers1S and L knockout strains as well as in the double knockout mutants (both Ers1S and L) in both regular glucose culture medium and under starvation conditions. $\Delta ypt7$ strain was used as a negative control, as its absence results in low amino acid content during nitrogen starvation due to impaired autophagosome-vacuole fusion. Two-way analysis of variance (ANOVA) was used with comparison done between different cell lines under the same condition. The data in (H) represents the mean ± SEM from $n = 3$ biological repeats.

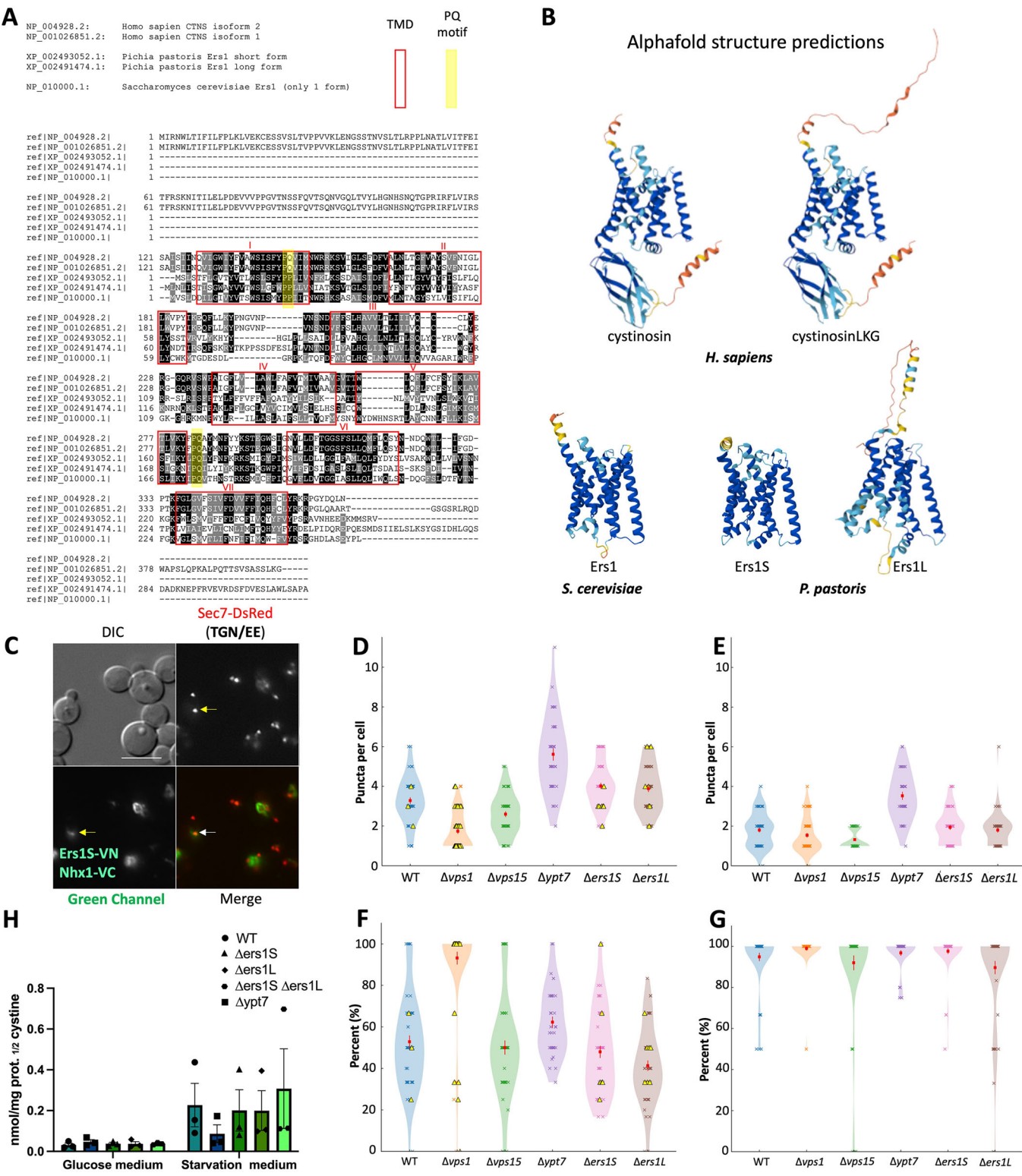

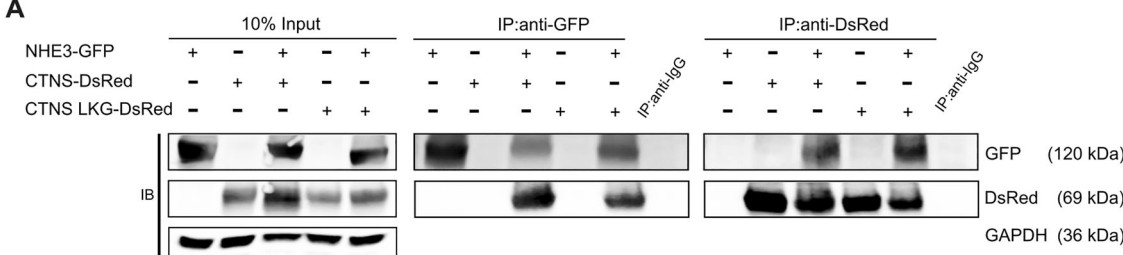

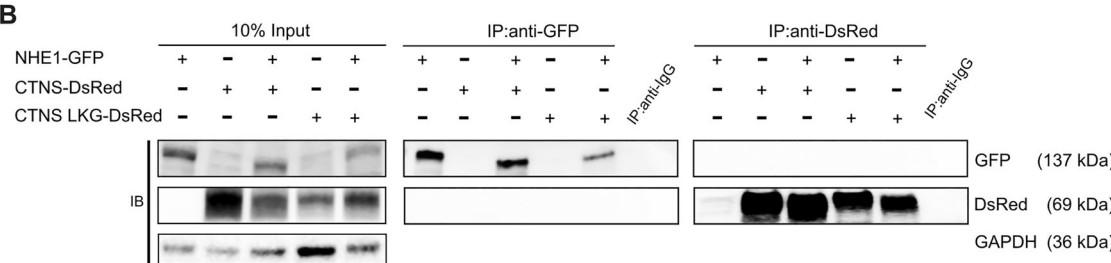

**Figure EV2. Cystinosin interacts with NHE3 but not with NHE1.**

Immunoprecipitation assay performed in (**A**) CTNS-deficient HK-2 and (**B**) HEK293T cells stably expressing NHE3 and NHE1, respectively, along with cystinosin and cystinosinLKG showing interaction with NHE3 (**A**) and no interaction with NHE1 (**B**) with both cystinosin and cystinosinLKG. Input lysates show proper expression of the proteins with glyceraldehyde-3-phosphate dehydrogenase (GAPDH) used as a loading control. Pull down with anti-IgG was used as a negative control.

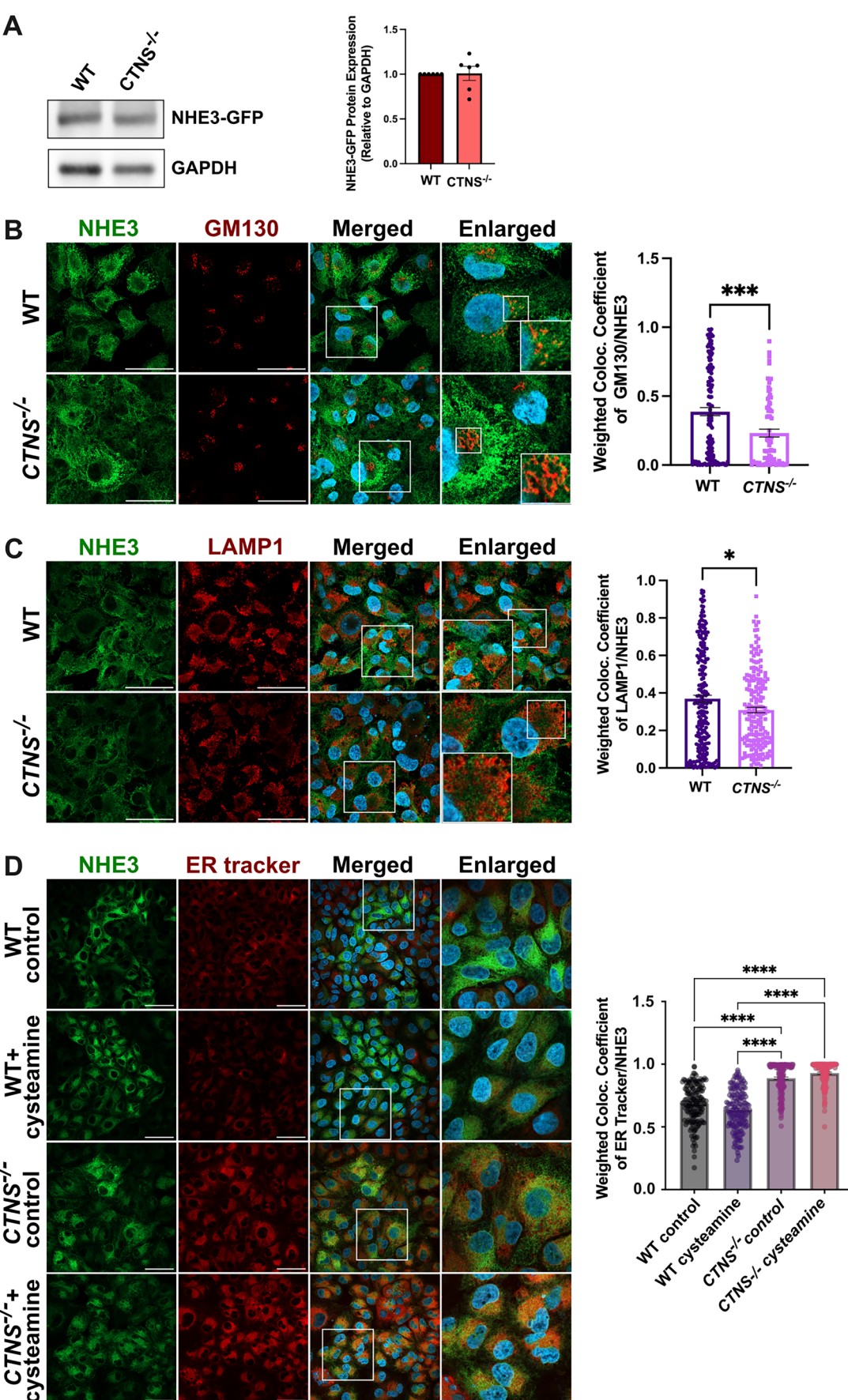

◀ **Figure EV3. Cystinosin deficiency impairs NHE3 trafficking in human WT and *CTNS*$^{-/-}$ HK2 proximal tubular cells.**

(A) Representative Western blot analysis of NHE3-GFP expression with GAPDH used as a loading control ($n = 6$ biological repeats). (B, C) Representative immunofluorescence images of NHE3-GFP (green) with markers of Golgi (GM130) (red) (B) and lysosomes (LAMP1) (red) (C) with their corresponding quantification showing defect in NHE3 subcellular localization in *CTNS*$^{-/-}$ HK2 cells. (D) Representative immunofluorescence images of NHE3-GFP (green) in WT and *CTNS*$^{-/-}$ HK2 cells, with or without cysteamine treatment along with ER tracker with their corresponding quantification. Scale bars: 50 µm (B–D). Bar graphs are presented as the mean ± SEM. For (B) $P = 0.0006$; (C) $P = 0.0124$. *$P < 0.05$; ***$P < 0.001$ using two-tailed Student's t test for Figures (B, C) and One-way ANOVA was done for (D); ****$P < 0.0001$. For (B–D), $n = 3$ biological repeats. Source data are available online for this figure.

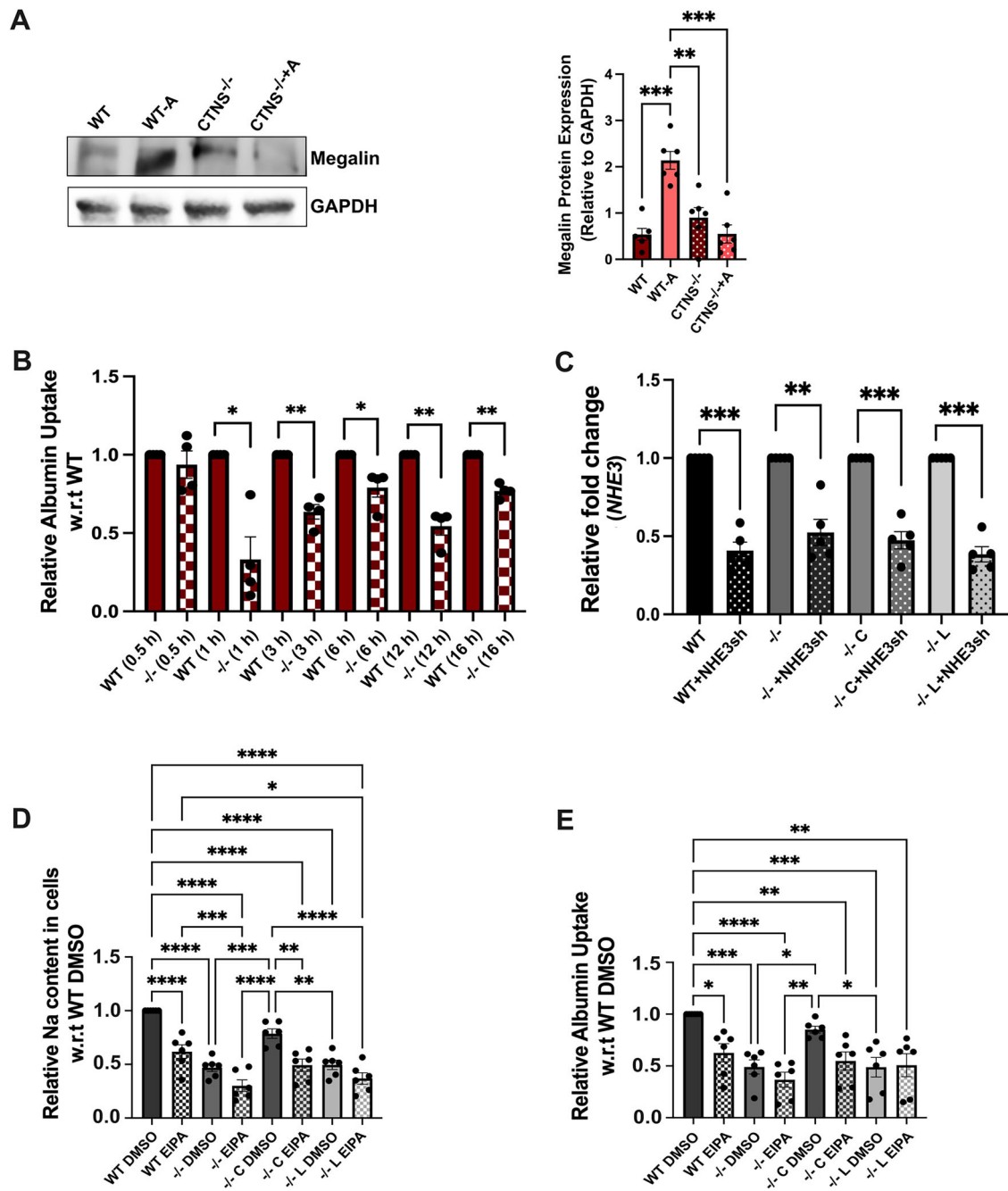

**Figure EV4. Cystinosin deficiency impairs NHE3 function in human WT and _CTNS⁻/⁻_ HK2 proximal tubular cells.**

(A) Representative Western blot analysis of Megalin expression, with or without Albumin555 treatment for 16 h; GAPDH is used as a loading control (n = 6). (B) Albumin uptake studies in WT and _CTNS⁻/⁻_ HK2 cells at different timepoints between 0.5 and 16 h measured using albumin555 by flow cytometry (n = 4). (C) _NHE3_ mRNA expression measured with quantitative PCR in cells treated with or without NHE3sh RNA confirming NHE3 knockdown (n = 5). (D, E) Na⁺ and albumin uptake in WT and _CTNS⁻/⁻_ HK2 cells (−/−), and in _CTNS⁻/⁻_ HK2 cells transduced with LV-CTNS (C) or LV-CTNS-LKG (L) treated with DMSO (vehicle), or the sodium transporter inhibitor (EIPA) (n = 6). For data in figure (A–E) each dot represents an independent biological replicate. Bar graphs are presented as the mean ± SEM. Two-tailed Student's t test was used for (B, C). One-way ANOVA was done for (A, D, E). For (A), WT vs. WT-A, P = 0.0005; WT-A vs. _CTNS⁻/⁻_, P = 0.0049; WT-A vs. _CTNS⁻/⁻_-A, P = 0.0005. For (B), WT vs. _CTNS⁻/⁻_, P = 0.0185, P = 0.0043, P = 0.0423, P = 0.0041, P = 0.0020, for 1, 3, 6, 12 and 16 h, respectively. For (C), P = 0.0004, P = 0.0048, P = 0.0007, P = 0.0002, for WT vs. WT+NHE3sh, −/− vs. −/− + NHE3sh, −/− C vs. −/− C + NHE3sh and −/− L vs. −/− L+ NHE3sh, respectively. For (D), P = 0.0008, P = 0.0025, P = 0.0021, P = 0.0007, P = 0.0135, for −/− DMSO vs. −/− C DMSO, −/− C DMSO vs. −/− L DMSO, −/− C DMSO vs. −/− C EIPA, WT EIPA vs. −/− EIPA and WT EIPA vs. −/− L EIPA, respectively. For (E), P = 0.0008, P = 0.0008, P = 0.0283, P = 0.0041, P = 0.0013, P = 0.0387, P = 0.0369 and P = 0.0017 for WT DMSO vs. −/− DMSO, WT DMSO vs. −/− L DMSO, WT DMSO vs. WT EIPA, WT DMSO vs. −/− C EIPA, WT DMSO vs. −/− L EIPA, −/− DMSO vs. −/− C DMSO, −/− C DMSO vs. −/− L DMSO and −/− C DMSO vs. −/− EIPA, respectively. *P < 0.05; **P < 0.01; ***P < 0.001; ****P < 0.0001.

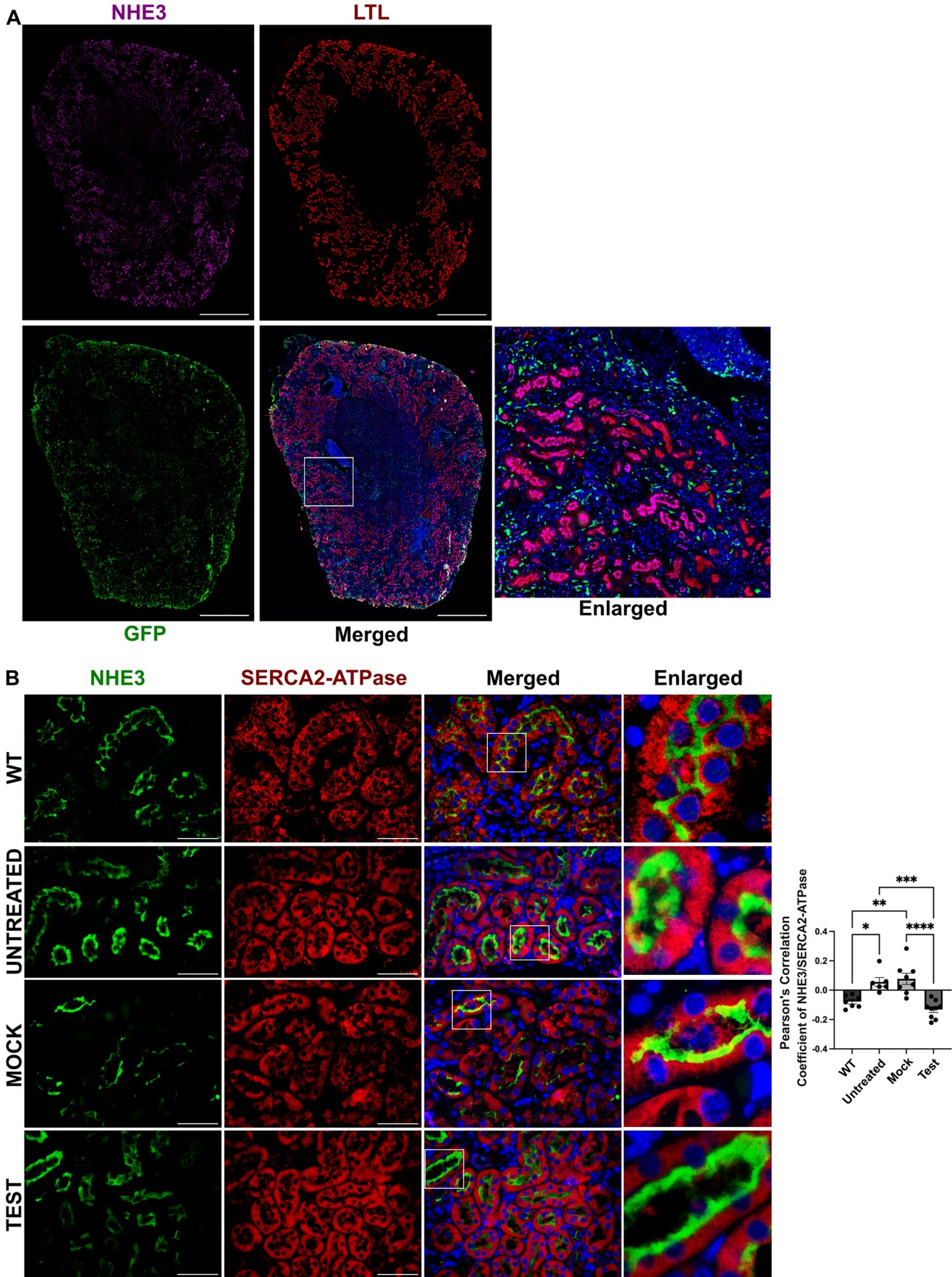

**Figure EV5. GFP⁺ WT HSPC distribution in the kidney following WT HSPC transplantation and colocalization analysis with an ER marker.**

(A) Overview of immunofluorescence images of entire kidney section from the Test mice stained with anti-GFP (green), anti-NHE3 (magenta) antibodies and anti-LTL (red) conjugate. This image shows the distribution of the GFP⁺ WT HSPC-derived cells as well as the colocalization of LTL and NHE3 within the kidney of *Ctns⁻/⁻* mice transplanted with WT HSPCs. (B) Representative immunofluorescence images of formalin fixed paraffin embedded (FFPE) kidney sections from WT, Ctns⁻/⁻, Mock and Test mice, stained with anti-NHE3 (green) and the ER marker anti-SERCA2 (red) antibodies, along with DAPI (nuclei; blue). Corresponding colocalization quantification between NHE3 and SERCA2 was performed using Pearson Correlation Coefficient. The data are presented as mean ± SEM (WT $n = 7$, Untreated $n = 6$, Mock $n = 8$, Test $n = 9$). Each dot represents an individual mouse. Scale bars: 4 mm (A); 50 µm (B). One-way analysis of variance (ANOVA) was used, WT vs. Untreated, $P = 0.0170$; WT vs. Mock, $P = 0.0025$; Untreated vs. Test, $P = 0.0003$. *$P < 0.05$; **$P < 0.01$; ***$P < 0.001$; ****$P < 0.0001$.

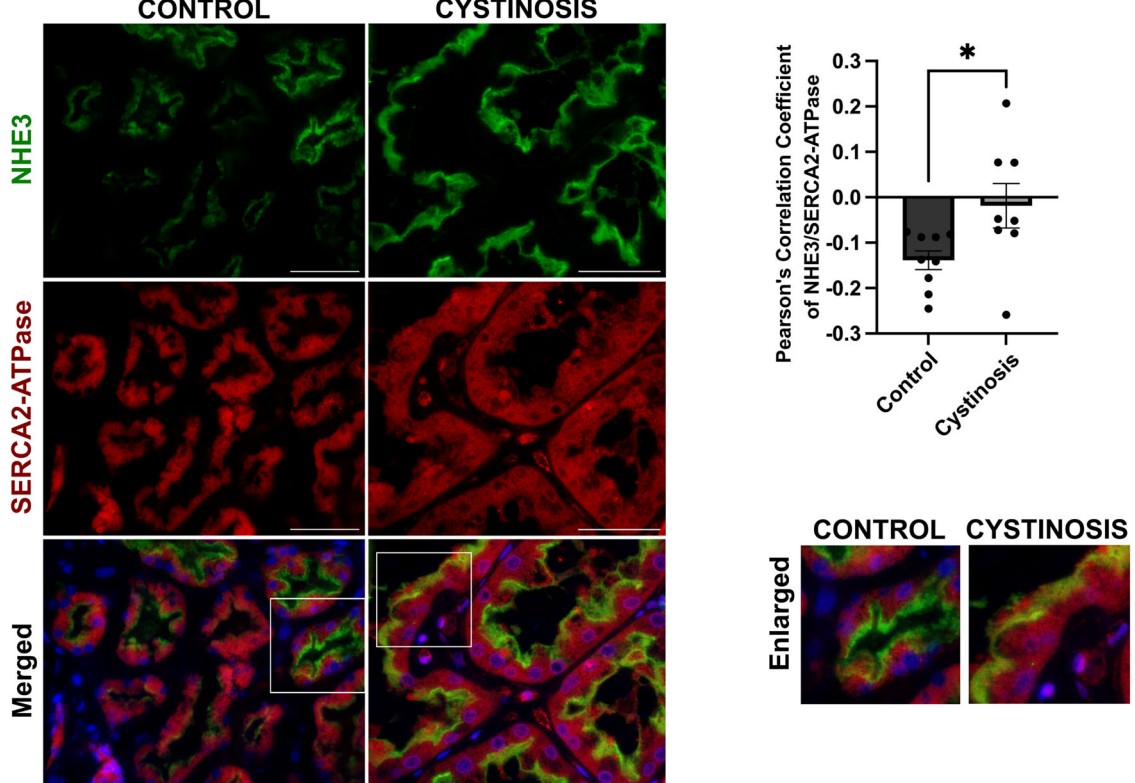

**Figure EV6.   NHE3 localization study with an ER marker in human control donor and cystinosis patient kidney tissue sample.**

Representative immunofluorescence images of formalin fixed paraffin embedded (FFPE) human kidney sections from a control donor and a cystinosis patient. Sections were stained with anti-NHE3 (green), anti-SERCA2 (red) antibodies and DAPI (nuclei). Scale bars: 50 μm. The quantitative data of colocalization between NHE3 and SERCA2 was determined by Pearson Correlation Coefficient. The data is presented as mean ± SEM from $n = 3$ biological repeats. Two-tailed Student's t-test was used; $P = 0.0495$; ($*P < 0.05$).

