## [Peer Review File · EMBO Reports]

Cystinosin regulates Na⁺/H⁺ Exchanger 3 trafficking and function in kidney proximal tubular cells

Veenita Khare, Jean-Claude Farre, Mouad Ait Kbaich, Céline J. Rocca, Cynthia Tang, Xuan Ma, Kavya Biederman, Ioli Mathur, Rafael A. Badell-Grau, Anusha Sivakumar, Rola Chen, Sergio D. Catz, and Stephanie Cherqui

Corresponding author(s): Stephanie Cherqui (scherqui@health.ucsd.edu)

Review Timeline:

Transfer Date:	4th Jun 25
Editorial Decision:	18th Jun 25
Revision Received:	31st Oct 25
Editorial Decision:	17th Dec 25
Revision Received:	8th Jan 26
Editorial Decision:	12th Feb 26
Revision Received:	12th Feb 26
Accepted:	13th Feb 26

Editor: Deniz Senyilmaz Tiebe / Martina Rembold

Transaction Report: This manuscript was transferred to

EMBO reports following peer review at Review Commons.

Review
COMMONS

Review #1

1. Evidence, reproducibility and clarity:

Evidence, reproducibility and clarity (Required)

The manuscript describes the dysregulated trafficking of NHE3 in proximal tubular cells of cystinosis patients and its relationship with cystinosis, the primary transporter affected in the disease. This abnormal trafficking primarily disrupts sodium and water reabsorption, contributing to proximal tubulopathy, a key feature of Fanconi Syndrome.

The manuscript is very well-written, with the authors using strong and appropriate methods to validate their findings.

The results support the idea that to effectively treat Fanconi Syndrome (FS), it is not enough to simply eliminate cystine from the lysosomes; rather, the cystinosis-specific functions in the proximal tubular cells (PTCs) need to be restored, which supports the ongoing clinical trial.

****Major concern:****

Even though it has been published previously, it would be crucial to include a more thorough characterization of the cystinosis phenotype and rescue in the mouse model, specifically focusing on cystine accumulation and key Fanconi Syndrome (FS) parameters. While the HSPC transplantation improved protein co-localization, no functional improvements were demonstrated. This lack of functional data leaves the relevance of the potential treatment largely unvalidated and raises concerns about the clinical applicability of the approach.

2. Significance:

Significance (Required)

- The authors employ various models of cystinosis, acknowledging the limitations of each and selecting the most suitable one for addressing each research question.

- In significance the authors state that this study offers new molecular insights in the endosomal trafficking of NHE3 and provides a new mechanism that may explain the underlying pathogenesis and early onset of the renal Fanconi syndrome in cystinosis. While the described mechanism contributes to the understanding of Fanconi syndrome, I believe it does not fully explain the condition, as other factors are likely involved as well. In the

discussion, this is also mentioned by the authors. The significance statement could be modified to make it more clear.

- The study identifies a novel function of cystinosin in regulating NHE protein trafficking, specifically linking NHE3 mislocalization to Fanconi Syndrome (FS). This suggests that restoring NHE3 localization may be essential for treating FS. However, while the study provides important molecular insights, it does not demonstrate functional rescue after HSPC treatment, leaving the effect of restoring expression and localization on FS improvement uncertain.

- This manuscript will interest researchers in nephrology, molecular biology, and translational medicine, particularly those studying Fanconi Syndrome, cystinosis, and endosomal trafficking. It appeals to both basic scientists investigating NHE protein regulation and cellular transport and clinicians exploring potential therapeutic strategies, including HSPC transplantation for renal disorders.

- My expertise lies in kidney models for investigating disease mechanisms and developing novel therapies, with over 10 years of experience studying cystinosis. I am familiar with the limitations of each model described in the manuscript, but I believe they were appropriately chosen for the different research questions. However, my main concern is the lack of data showing the effect of the treatment on the proposed Fanconi Syndrome (FS) phenotype in mice. Demonstrating this is essential, as rescuing FS would significantly strengthen the message and the impact of the proposed treatment.

3. How much time do you estimate the authors will need to complete the suggested revisions:

Estimated time to Complete Revisions (Required)

(Decision Recommendation)

Between 1 and 3 months

4. Review Commons values the work of reviewers and encourages them to get credit for their work. Select 'Yes' below to register your reviewing activity at Web of Science Reviewer Recognition Service (formerly Publons); note that the content of your review will not be visible on Web of Science.

Yes

Review #2

1. Evidence, reproducibility and clarity:

Evidence, reproducibility and clarity (Required)

****Evidence, reproducibility, and clarity****

In this manuscript, Khare and colleagues present novel insights into the potential pathways and mechanisms underlying the pathogenesis of cystinosis, a prototypical lysosomal storage disorder caused by the loss of the cystine transporter cystinosin (CTNS). CTNS deficiency leads to early proximal tubule (PT) dysfunction and tubulopathy, which progresses to chronic kidney disease and multisystem complications later in life. Using genetically modified organisms, patient-derived cells, and kidney tissue from individuals with cystinosis, the authors demonstrate that CTNS, beyond its established role in lysosomal cystine transport, also appears to regulate the trafficking and compartmentalization of specific members of the sodium-hydrogen exchanger (Na^+/H^+) family (NHEs). Notably, they show that the absence of CTNS results in the mislocalization and impaired vesicular trafficking of NHE3, contributing to the urinary loss of sodium and other essential nutrients. This unexpected interaction seems to be evolutionarily conserved, as supported by compelling evidence from yeast model studies. Furthermore, proof-of-concept experiments suggest that the observed phenotypic changes are dependent on CTNS, as demonstrated by rescue strategies involving transplantation of wild-type hematopoietic stem and progenitor cells (HSPCs), which reintroduce functional CTNS and restore PT homeostasis. Despite these novel findings, the study is limited by a lack of molecular and structural detail on the mechanisms underlying CTNS-NHE3 interactions and their role in organelle homeostasis and trafficking-related pathways. As such, the conclusions remain largely observational and descriptive, though they are well supported by the presented data.

****Major concerns:****

1. Using *in vivo* imaging and BiFC assay technologies, the authors demonstrate that Ers1 (a functional homolog of CTNS) interacts with Nhx1 (the yeast ortholog of mammalian NHE) in endosomes, suggesting a role for Ers1 in facilitating the intracellular trafficking and compartmentalization of Nhx1. However, the mechanistic details of how Ers1 regulates

Nhx1 trafficking remain insufficiently addressed. Do single or double *ers1* mutants impair Nhx1 localization by interfering with endolysosomal maturation or vesicular dynamics? Are additional nutrient transporters, beyond Nhx1, similarly mislocalized or functionally compromised in the *ers1* mutant background? Moreover, can VPS1 overexpression rescue the trafficking defects and restore proper endosomal localization of Nhx1 in *ers1* mutants, thereby providing functional evidence for this regulatory pathway?

2. Given the absence of kidney proximal tubule (PT)-like structures in yeast, the authors could have strengthened their conclusions by incorporating the zebrafish model to validate the evolutionary conservation and biological relevance of the Ers1-Nhx1 interaction in the context of renal physiology. The zebrafish pronephros closely recapitulates key features of the mammalian kidney, including epithelial polarity, tight junction complexes, a well-developed endolysosomal network, mitochondria-driven energy metabolism, and the expression of endocytic receptors and nutrient transporters such as CTNS and NHE3. Incorporating this model would have provided a more physiologically relevant system to probe the functional consequences of CTNS-NHE3 interactions *in vivo*, particularly in the setting of early cystinosis-related PT dysfunction.

3. Extending their findings to a mammalian cell culture system (HEK293), the authors show that CTNS appears to interact with NHE3 and NHE2, but not with NHE1, suggesting a degree of specificity in CTNS-NHE interactions. However, the study does not address whether these interactions occur within the endolysosomal system, nor does it examine the subcellular localization of NHE proteins in the absence of CTNS. Additional experiments are needed to determine whether CTNS deficiency leads to NHE mislocalization or altered trafficking patterns, which would further support the proposed role of CTNS in regulating ion transporter compartmentalization and function.

4. In Fig. 3, the authors' reconciliation of their findings is unclear. Specifically, NHE3 appears to be trapped within the ER in CTNS-deficient HK2 cells. What are the mechanistic factors driving this unexpected ion transporter mislocalization in mutant cells? Are these changes a direct consequence of CTNS loss, occurring independently of cystine storage (e.g., after cysteamine treatment)? The authors should transfect CTNS-deficient HK2 cells with tagged WT-CTNS and perform co-immunoprecipitation (co-IP) to confirm the interaction between CTNS and NHE3. They should also identify potential interaction partners using mass spectrometry or Western blotting, with a focus on proteins involved in trafficking, ER stress response, or endocytosis.

5. Does NHE3 accumulate within the ER, as observed in HK2 cells, in kidney tubular cells

derived from cystinosis patients, in the proximal tubular (PT) segments of CtnsKO mice, and kidney samples from human cystinosis patients? Understanding whether this mislocalization of NHE3 occurs in these clinically relevant models is crucial, as it could provide insights into the pathological mechanisms underlying cystinosis and its impact on kidney function. If similar mislocalization is observed in patient-derived cells and animal models, it would strengthen the link between CTNS deficiency and disrupted ion transporter trafficking, which may contribute to the renal dysfunction seen in cystinosis.

6. Additionally, how do the authors explain the rescue of NHE3 mislocalization and functional compartmentalization upon transplantation with wild-type (WT) hematopoietic stem and progenitor cells (HSPCs)? Investigating the effect of HSPC transplantation is important to determine whether restoring functional CTNS expression can correct the trafficking defects in NHE3 and whether this has therapeutic potential for cystinosis patients.

7. Finally, what about the levels of sodium in the urine of transplanted KO mice? Monitoring sodium excretion in these mice after transplantation can provide functional evidence of transporter recovery. This would help determine if the restoration of NHE3 localization leads to an improvement in sodium reabsorption and overall kidney function, offering further insight into the therapeutic benefits of HSPC-based interventions for cystinosis.

****Minor changes:****

1. The introduction and discussion should be streamlined and narrowed down, focusing on the key points that directly address the rationale, core findings and significance of the study.

2. Quantification of the colocalized signals in Fig.1a-d.

3. Functional validation of KO strategies in Figs. 2 and 3, i.e., cystine measurements, expression levels of endocytic receptors, endocytic trafficking and ligand uptake.

4. Some graphs (Fig.3e-h) are not so easy to read and follow. In addition, crucial information regarding the reproducibility and number of biologically independent experiments/samples, and multiple comparisons test used during the statistical analysis.

5. Applying a one-way ANOVA with a small sample size (n=3) is generally not recommended. With such a small sample size, the power of the ANOVA test to detect

meaningful differences between groups is very low. A low power increases the likelihood of Type II errors, where true differences between groups are not detected.

One-way ANOVA relies on several assumptions:

a) Normality: The data should be approximately normally distributed within each group. Small sample sizes make it difficult to reliably assess the normality of the data.

b) Homogeneity of Variance: The variance within each group should be approximately equal. With small samples, this assumption may be harder to check and violate. It would be better to consider other (non-parametric) approaches or increase the sample size if feasible.

2. Significance:

Significance (Required)

Although a detailed mechanistic understanding remains to be fully elucidated, this study significantly advances our understanding of the biological roles of lysosomal transporters- specifically CTNS- in maintaining tissue homeostasis and contributing to disease pathology. Importantly, it highlights a novel functional link between CTNS and the regulation of nutrient transporter trafficking, suggesting new therapeutic avenues. These findings have broad relevance for the basic research community, particularly in the fields of lysosome biology and transport physiology, and they lay a critical foundation for the development of targeted therapies aimed at restoring transporter function in cystinosis and potentially other lysosomal storage disorders.

3. How much time do you estimate the authors will need to complete the suggested revisions:

Estimated time to Complete Revisions (Required)

(Decision Recommendation)

More than 6 months

4. Review Commons values the work of reviewers and encourages them to get credit for their work. Select 'Yes' below to register your reviewing activity at Web of Science Reviewer Recognition Service (formerly Publons); note that the content of your review will not be visible on Web of Science.

Yes

Review #3

1. Evidence, reproducibility and clarity:

Evidence, reproducibility and clarity (Required)

The manuscript titled "Cystinosis is involved in Na⁺/H⁺ Exchanger 3 trafficking in the proximal tubular cells: new insights into the renal Fanconi syndrome in cystinosis" by Veenita Khare and colleagues explores a previously unknown function of cystinosis in regulating sodium/hydrogen (Na⁺/H⁺) exchanger proteins within endosomes. Their findings indicate that cystinosis influences the localization, movement, and sodium absorption of NHE3 in proximal tubular cells (PTCs). Remarkably, reintroducing CTNS into CTNS-deficient PTCs corrected these abnormalities, whereas CTNS-LKG, an isoform found in lysosomes and the plasma membrane, did not. Furthermore, NHE3 mislocalization was confirmed in *Ctns*^{-/-} mice as well as in kidney tissue from cystinosis patients. Notably, transplantation of wild-type hematopoietic stem and progenitor cells into *Ctns*^{-/-} mice restored NHE3 expression at the brush border membrane. While this research sheds light on a novel role of cystinosis in NHE3 trafficking and potentially expands the known cellular dysfunctions associated with CTNS loss of function, several major concerns must be addressed before this manuscript can be considered for publication.

****Major concerns:****

- The authors demonstrate that exogenously expressed cystinosis interacts with exogenously expressed NHE3 and NHE2, but not with NHE1. However, the Western blot presented in Figure S2 is unconvincing. The membranes from IP:GFP blotted with anti-DsRed antibody and from IP:DsRed blotted with anti-GFP antibody are completely white, with no visible background. The authors should provide a longer exposure of the membrane to determine whether background signals or non-specific bands appear.
- The authors should discuss the potential transcriptional effects leading to NHE2 overexpression in *CTNS*^{-/-} cells and clarify why NHE3 protein levels are reduced, despite unchanged mRNA expression.

- The data in Figures 3C, 3D, and S3, along with the corresponding quantitative analyses, are unconvincing due to the broad and diffuse NHE3 expression observed in both WT and CTNS^{-/-} cells. The authors should evaluate NHE3 expression levels before assessing colocalization with organelle markers and consider performing a cycloheximide treatment to inhibit de novo protein synthesis before analyzing NHE3 localization. Additionally, the images provided do not accurately represent the quantitative analysis, as they display extreme conditions (high or zero colocalization), which do not effectively support the intended conclusions.
- Figures 3F, H: The effect of albumin on megalin/cubilin expression should be assessed.
- Figures 3F, H: A time-course analysis of albumin endocytosis would be more appropriate for assessing the internalization rate and lysosomal trafficking rather than the current 16-hour endpoint assessment. At this late time point, intracellular albumin levels result from a combination of internalization, lysosomal processing, and exocytosis (as TFEB activation may enhance lysosomal exocytosis).
- Figures 3G, H: EIPA is used at a very high concentration, which may lead to non-specific effects. A more targeted approach, such as siRNA-mediated NHE3 downregulation, could provide more reliable results.
- Figures 3E, G: NHE3 levels at the plasma membrane should be assessed using biochemical approaches (e.g., membrane protein biotinylation) or imaging techniques such as confocal or TIRF microscopy.
- Figure 4: The claims made are not clearly supported by the provided data. Instead of pseudo-TIRF imaging, laser scanning confocal live imaging may be more suitable for assessing vesicle tracking and movement, unless vesicle fusion or endocytosis from the plasma membrane is the primary focus. Additionally, analysis should be conducted on cells expressing similar NHE3 levels to ensure consistency.
- Figure 5C: The discrepancy between NHE2 mRNA and protein levels in WT vs. CTNS^{-/-} mice is not explained or discussed.
- Figure 5D: The provided images are highly variable and not convincing. The magnifications used for the images correlate with the quantitative analysis, but other regions of the section do not align with the inset, leading to inconsistencies. The restored

CTNS expression by HSCGT should be assessed in tubules with correct NHE3 localization (TEST samples).

2. Significance:

Significance (Required)

This study presents a novel function of cystinosin in regulating the trafficking of Na⁺/H⁺ exchanger 3 (NHE3) in proximal tubular cells (PTCs). By demonstrating how cystinosin deficiency leads to NHE3 mislocalization and how hematopoietic stem cell transplantation can restore its proper expression, the authors provide new insights into renal Fanconi syndrome in cystinosis.

Strengths

The study identifies a previously unrecognized role of cystinosin in NHE3 endosomal trafficking, adding a new mechanistic layer to our understanding of cystinosis-related renal dysfunction.

The use of multiple experimental models, including *Ctns*^{-/-} mice, cystinosis patient kidney samples, and various in vitro approaches, strengthens the validity of the findings.

The therapeutic implications of hematopoietic stem cell transplantation in rescuing NHE3 expression provide a potential translational perspective that may be of interest for future clinical research.

Limitations and Areas for Improvement

Some data presentations, particularly Western blot images and immunofluorescence quantifications, require further clarification to support the claims made in the manuscript.

The discrepancy between NHE2 mRNA and protein levels in *Ctns*^{-/-} mice remains unexplained and should be addressed.

The choice of experimental conditions (e.g., EIPA concentration, endpoint albumin endocytosis assessment) may lead to non-specific effects and should be refined.

The study would benefit from additional mechanistic insights into the transcriptional regulation of NHE3/NHE2 in cystinosin-deficient cells.

Advances in the Field

This study extends the current understanding of cystinosis beyond its classical role in lysosomal cystine transport. To our knowledge, this is the first report demonstrating that cystinosis directly affects NHE3 trafficking in PTCs, a finding with significant implications for renal physiology in cystinosis. The insights provided here are both mechanistic (identifying a new trafficking function of cystinosis) and potentially translational (suggesting stem cell-based approaches as a therapeutic avenue).

Audience and Impact

This work will primarily interest specialized audiences in nephrology, cell biology (lysosomal disease and trafficking), and rare genetic disorders. More specifically:

Basic researchers studying renal physiology, lysosomal function, and protein trafficking will find the mechanistic insights valuable.

Translational/clinical researchers investigating cystinosis and renal Fanconi syndrome may be particularly interested in the therapeutic implications of hematopoietic stem cell transplantation.

Broader nephrology community may also benefit from understanding how lysosomal dysfunction impacts sodium homeostasis.

While the study is largely mechanistic, the translational component may expand its relevance beyond the specific field of cystinosis research.

My expertise lies in cell biology, renal physiology, lysosomal trafficking, and rare genetic disorders. I am comfortable evaluating the mechanistic aspects of the study, including protein trafficking and renal tubular dysfunction. However, I defer to experts in hematopoietic stem cell transplantation and nephrology-specific clinical applications for further assessment of the translational potential of these findings.

3. How much time do you estimate the authors will need to complete the suggested revisions:

Estimated time to Complete Revisions (Required)

(Decision Recommendation)

More than 6 months

Yes

Revision Plan

Manuscript number: RC-2025-02907

Corresponding author(s): Stephanie, Cherqui

[The “revision plan” should delineate the revisions that authors intend to carry out in response to the points raised by the referees. It also provides the authors with the opportunity to explain their view of the paper and of the referee reports.]

The document is important for the editors of affiliate journals when they make a first decision on the transferred manuscript. It will also be useful to readers of the reprint and help them to obtain a balanced view of the paper.

*If you wish to submit a full revision, please use our "Full Revision" template. **It is important to use the appropriate template to clearly inform the editors of your intentions.**]*

1. General Statements [optional]

This section is optional. Insert here any general statements you wish to make about the goal of the study or about the reviews.

We sincerely thank the reviewers for their thoughtful and constructive comments. We have put together a plan of action to address all the points raised, as detailed in the following sections.

2. Description of the planned revisions

Insert here a point-by-point reply that explains what revisions, additional experimentations and analyses are planned to address the points raised by the referees.

Reviewer #1 (Evidence, reproducibility and clarity (Required)):

The manuscript describes the dysregulated trafficking of NHE3 in proximal tubular cells of cystinosis patients and its relationship with cystinosin, the primary transporter affected in the disease. This abnormal trafficking primarily disrupts sodium and water reabsorption, contributing to proximal tubulopathy, a key feature of Fanconi Syndrome.

The manuscript is very well-written, with the authors using strong and appropriate methods to validate their findings.

The results support the idea that to effectively treat Fanconi Syndrome (FS), it is not enough to simply eliminate cystine from the lysosomes; rather, the cystinosin-specific functions in the proximal tubular cells (PTCs) need to be restored, which supports the ongoing clinical trial.

Major concern:

Revision Plan

1. Even though it has been published previously, it would be crucial to include a more thorough characterization of the cystinosis phenotype and rescue in the mouse model, specifically focusing on cystine accumulation and key Fanconi Syndrome (FS) parameters. While the HSPC transplantation improved protein co-localization, no functional improvements were demonstrated. This lack of functional data leaves the relevance of the potential treatment largely unvalidated and raises concerns about the clinical applicability of the approach.

Response: We thank the reviewer for the constructive comments. We will evaluate kidney function and key Fanconi Syndrome (FS) parameters in the urine and serum samples of the mice that we have kept frozen.

Reviewer #1 (Significance (Required)):

2. The study identifies a novel function of cystinosin in regulating NHE protein trafficking, specifically linking NHE3 mis localization to Fanconi Syndrome (FS). This suggests that restoring NHE3 localization may be essential for treating FS. However, while the study provides important molecular insights, it does not demonstrate functional rescue after HSPC treatment, leaving the effect of restoring expression and localization on FS improvement uncertain.

Response: We will evaluate kidney function and key Fanconi Syndrome (FS) parameters in the urine and serum samples of the mice. This analysis will further help in reassessing the impact of HSPC transplantation on functional improvement.

3. This manuscript will interest researchers in nephrology, molecular biology, and translational medicine, particularly those studying Fanconi Syndrome, cystinosis, and endosomal trafficking. It appeals to both basic scientists investigating NHE protein regulation and cellular transport and clinicians exploring potential therapeutic strategies, including HSPC transplantation for renal disorders.

My expertise lies in kidney models for investigating disease mechanisms and developing novel therapies, with over 10 years of experience studying cystinosis. I am familiar with the limitations of each model described in the manuscript, but I believe they were appropriately chosen for the different research questions. However, my main concern is the lack of data showing the effect of the treatment on the proposed Fanconi Syndrome (FS) phenotype in mice. Demonstrating this is essential, as rescuing FS would significantly strengthen the message and the impact of the proposed treatment.

Response: As stipulated above, this will be done as the samples are readily available and just needs to be processed.

Reviewer #2 (Evidence, reproducibility and clarity (Required)):

In this manuscript, Khare and colleagues present novel insights into the potential pathways and mechanisms underlying the pathogenesis of cystinosis, a prototypical lysosomal storage disorder

caused by the loss of the cystine transporter cystinosin (CTNS). CTNS deficiency leads to early proximal tubule (PT) dysfunction and tubulopathy, which progresses to chronic kidney disease and multisystem complications later in life.

Using genetically modified organisms, patient-derived cells, and kidney tissue from individuals with cystinosis, the authors demonstrate that CTNS, beyond its established role in lysosomal cystine transport, also appears to regulate the trafficking and compartmentalization of specific members of the sodium-hydrogen exchanger (Na^+/H^+) family (NHEs). Notably, they show that the absence of CTNS results in the mis localization and impaired vesicular trafficking of NHE3, contributing to the urinary loss of sodium and other essential nutrients. This unexpected interaction seems to be evolutionarily conserved, as supported by compelling evidence from yeast model studies.

Furthermore, proof-of-concept experiments suggest that the observed phenotypic changes are dependent on CTNS, as demonstrated by rescue strategies involving transplantation of wild-type hematopoietic stem and progenitor cells (HSPCs), which reintroduce functional CTNS and restore PT homeostasis.

Despite these novel findings, the study is limited by a lack of molecular and structural detail on the mechanisms underlying CTNS-NHE3 interactions and their role in organelle homeostasis and trafficking-related pathways. As such, the conclusions remain largely observational and descriptive, though they are well supported by the presented data.

Major concerns:

7. In Fig. 3, the authors' reconciliation of their findings is unclear. Specifically, NHE3 appears to be trapped within the ER in CTNS-deficient HK2 cells.

- a. What are the mechanistic factors driving this unexpected ion transporter mis localization in mutant cells?
- b. Are these changes a direct consequence of CTNS loss, occurring independently of cystine storage (e.g., after cysteamine treatment)?
- c. The authors should transfect CTNS-deficient HK2 cells with tagged WT-CTNS and perform co-immunoprecipitation (co-IP) to confirm the interaction between CTNS and NHE3.
- d. They should also identify potential interaction partners using mass spectrometry or Western blotting, with a focus on proteins involved in trafficking, ER stress response, or endocytosis.

Response:

- a. Please refer to section "Description of analyses that author prefer not to carry out".
- b. We believe that NHE3 mis localization and altered trafficking are indeed a direct consequence of CTNS loss as this can be rescued by adding the CTNS gene back to the cells. To prove that cystine storage does not have any impact in this phenomenon, we will treat the HK2 and CTNS-KO HK2 cells with cysteamine as suggested by the reviewer.
- c. We have already demonstrated the interaction between tagged NHE3 and tagged CTNS by co-IP in the embryonic kidney cells, HEK293. If the reviewers find it appropriate, we could extend this analysis by transfecting CTNS-deficient HK-2 cells with tagged wild-type CTNS and performing co-IP to further validate the interaction between tagged-CTNS and tagged-NHE3, but

Revision Plan

we do not expect anything different. This interaction has also been demonstrated by unbiased screening in yeast and confirmed by BiFC analysis in yeast.

d. Please refer to section “Description of analyses that author prefer not to carry out”.

8. Does NHE3 accumulate within the ER, as observed in HK2 cells, in kidney tubular cells derived from cystinosis patients, in the proximal tubular (PT) segments of *Ctns*KO mice, and kidney samples from human cystinosis patients? Understanding whether this mislocalization of NHE3 occurs in these clinically relevant models is crucial, as it could provide insights into the pathological mechanisms underlying cystinosis and its impact on kidney function. If similar mislocalization is observed in patient-derived cells and animal models, it would strengthen the link between CTNS deficiency and disrupted ion transporter trafficking, which may contribute to the renal dysfunction seen in cystinosis.

Response: We propose to perform immunofluorescence in murine and human kidney to verify the subcellular localization of NHE3 in CTNS-KO proximal tubular cells using co-staining with endoplasmic reticulum (ER) markers.

10. Finally, what about the levels of sodium in the urine of transplanted KO mice? Monitoring sodium excretion in these mice after transplantation can provide functional evidence of transporter recovery. This would help determine if the restoration of NHE3 localization leads to an improvement in sodium reabsorption and overall kidney function, offering further insight into the therapeutic benefits of HSPC-based interventions for cystinosis.

Response: We thank the reviewer for the valuable suggestion and propose to assess urinary sodium levels in mice following transplantation compared to WT and *Ctns*^{-/-} controls.

Minor

2. Quantification of the colocalized signals in Fig.1a-d.

Response- We will add the quantification data in Fig.1a-d.

Reviewer #2 (Significance (Required)):

Although a detailed mechanistic understanding remains to be fully elucidated, this study significantly advances our understanding of the biological roles of lysosomal transporters- specifically CTNS-in maintaining tissue homeostasis and contributing to disease pathology. Importantly, it highlights a novel functional link between CTNS and the regulation of nutrient transporter trafficking, suggesting new therapeutic avenues. These findings have broad relevance for the basic research community, particularly in the fields of lysosome biology and transport physiology, and they lay a critical foundation for the development of targeted therapies aimed at restoring transporter function in cystinosis and potentially other lysosomal storage disorders.

Response: We sincerely thank the reviewer for the positive and encouraging feedback.

Revision Plan

Reviewer #3 (Evidence, reproducibility and clarity (Required)):

The manuscript titled "Cystinosin is involved in Na⁺/H⁺ Exchanger 3 trafficking in the proximal tubular cells: new insights into the renal Fanconi syndrome in cystinosis" by Veenita Khare and colleagues explores a previously unknown function of cystinosin in regulating sodium/hydrogen (Na⁺/H⁺) exchanger proteins within endosomes. Their findings indicate that cystinosin influences the localization, movement, and sodium absorption of NHE3 in proximal tubular cells (PTCs). Remarkably, reintroducing CTNS into CTNS-deficient PTCs corrected these abnormalities, whereas CTNS-LKG, an isoform found in lysosomes and the plasma membrane, did not. Furthermore, NHE3 mis localization was confirmed in Ctns^{-/-} mice as well as in kidney tissue from cystinosis patients. Notably, transplantation of wild-type hematopoietic stem and progenitor cells into Ctns^{-/-} mice restored NHE3 expression at the brush border membrane. While this research sheds light on a novel role of cystinosin in NHE3 trafficking and potentially expands the known cellular dysfunctions associated with CTNS loss of function, several major concerns must be addressed before this manuscript can be considered for publication.

Major concerns:

1. The authors demonstrate that exogenously expressed cystinosin interacts with exogenously expressed NHE3 and NHE2, but not with NHE1. However, the Western blot presented in Figure S2 is unconvincing. The membranes from IP:GFP blotted with anti-DsRed antibody and from IP:DsRed blotted with anti-GFP antibody are completely white, with no visible background. The authors should provide a longer exposure of the membrane to determine whether background signals or non-specific bands appear.

Response: Please find the longer exposure blot images provided below for your review. On the right hand side, please see the raw blots and on the left hand side the blots probed with the respective antibodies.

IP:GFP

IP:GFP blotted with anti-GFP antibody

Revision Plan

**IP:GFP
blotted with anti-DsRed antibody**

IP:DsRed

**IP:DsRed
blotted with anti-DsRed
antibody**

**IP:DsRed
blotted with anti-GFP
antibody**

3. The data in Figures 3C, 3D, and S3, along with the corresponding quantitative analyses, are unconvincing due to the broad and diffuse NHE3 expression observed in both WT and CTNS^{-/-} cells. The authors should evaluate NHE3 expression levels before assessing colocalization with organelle markers and consider performing a cycloheximide treatment to inhibit de novo protein synthesis before analyzing NHE3 localization. Additionally, the images provided do not accurately represent the quantitative analysis, as they display extreme conditions (high or zero colocalization), which do not effectively support the intended conclusions.

Revision Plan

Response: We have not observed differences in NHE3 expression levels in these cells. We will include the analysis of NHE3 expression level by Western blot, as suggested by the reviewer. We will choose different representative images as suggested.

6. Figures 3G, H: EIPA is used at a very high concentration, which may lead to non-specific effects. A more targeted approach, such as siRNA-mediated NHE3 downregulation, could provide more reliable results.

Response: We will perform NHE3 knockdown using siRNA-mediated approach to evaluate its effect on intracellular sodium (Na^+) content and albumin uptake as suggested by the Reviewer.

7. Figures 3E, G: NHE3 levels at the plasma membrane should be assessed using biochemical approaches (e.g., membrane protein biotinylation) or imaging techniques such as confocal or TIRF microscopy.

Response: We will assess endogenous NHE3 levels at the plasma membrane using TIRF microscopy after calibration at 100-130 nm depth.

8. Figure 4: The claims made are not clearly supported by the provided data. Instead of pseudo-TIRF imaging, laser scanning confocal live imaging may be more suitable for assessing vesicle tracking and movement, unless vesicle fusion or endocytosis from the plasma membrane is the primary focus. Additionally, analysis should be conducted on cells expressing similar NHE3 levels to ensure consistency.

Response: Pseudo-TIRFM (oblique illumination), which illuminates approximately **400 nm** into the sample, is well-suited for our study as the cells used are relatively flat and approximately 400 nm in depth. Given this geometry, pseudo-TIRFM provides high-contrast imaging of vesicle dynamics in the cell body near the basal membrane and performs better than confocal microscopy for our purposes as its gentle illumination avoids photobleaching. Furthermore, we have ensured that analyses were performed on cells expressing comparable levels of NHE3 to maintain consistency across experimental conditions. We will choose different representative images and, as stated above, will include the analysis of NHE3 expression levels.

10. Figure 5D: The provided images are highly variable and not convincing. The magnifications used for the images correlate with the quantitative analysis, but other regions of the section do not align with the inset, leading to inconsistencies. The restored CTNS expression by HSCGT should be assessed in tubules with correct NHE3 localization (TEST samples).

Response: Quantification was performed using images from entire kidney sections acquired at lower magnification and stitched together to ensure comprehensive analysis as shown in Supplemental Figure 4. For illustrative purposes, we used higher magnification images to highlight specific features. We can provide improved representative images to enhance clarity.

Reviewer #3 (Significance (Required)):

Revision Plan

This study presents a novel function of cystinosin in regulating the trafficking of Na⁺/H⁺ exchanger 3 (NHE3) in proximal tubular cells (PTCs). By demonstrating how cystinosin deficiency leads to NHE3 mis localization and how hematopoietic stem cell transplantation can restore its proper expression, the authors provide new insights into renal Fanconi syndrome in cystinosis.

Strengths

The study identifies a previously unrecognized role of cystinosin in NHE3 endosomal trafficking, adding a new mechanistic layer to our understanding of cystinosis-related renal dysfunction.

The use of multiple experimental models, including Ctns^{-/-} mice, cystinosis patient kidney samples, and various in vitro approaches, strengthens the validity of the findings.

The therapeutic implications of hematopoietic stem cell transplantation in rescuing NHE3 expression provide a potential translational perspective that may be of interest for future clinical research.

Response: We sincerely thank the reviewer for the comments.

Limitations and Areas for Improvement

1. Some data presentations, particularly Western blot images and immunofluorescence quantifications, require further clarification to support the claims made in the manuscript.

Response: We have addressed this in the previous Response # Major concerns: 1,10 under “Description of the planned revision section”.

2. The discrepancy between NHE2 mRNA and protein levels in Ctns^{-/-} mice remains unexplained and should be addressed.

Response: We have addressed this in the Response # Major concerns: 9 under “already been incorporated in the transferred manuscript section”.

3. The choice of experimental conditions (e.g., EIPA concentration, endpoint albumin endocytosis assessment) may lead to non-specific effects and should be refined.

Response: We will perform NHE3 knockdown using siRNA-mediated to evaluate the specific impact of decreased NHE3 on intracellular sodium (Na⁺) content and albumin uptake.

Advance in the Field

This study extends the current understanding of cystinosin beyond its classical role in lysosomal cystine transport. To our knowledge, this is the first report demonstrating that cystinosin directly affects NHE3 trafficking in PTCs, a finding with significant implications for renal physiology in cystinosis. The insights provided here are both mechanistic (identifying a new trafficking function of cystinosin) and potentially translational (suggesting stem cell-based approaches as a therapeutic avenue).

Audience and Impact

Revision Plan

This work will primarily interest specialized audiences in nephrology, cell biology (lysosomal disease and trafficking), and rare genetic disorders. More specifically: Basic researchers studying renal physiology, lysosomal function, and protein trafficking will find the mechanistic insights valuable.

Translational/clinical researchers investigating cystinosis and renal Fanconi syndrome may be particularly interested in the therapeutic implications of hematopoietic stem cell transplantation. Broader nephrology community may also benefit from understanding how lysosomal dysfunction impacts sodium homeostasis.

While the study is largely mechanistic, the translational component may expand its relevance beyond the specific field of cystinosis research.

My expertise lies in cell biology, renal physiology, lysosomal trafficking, and rare genetic disorders. I am comfortable evaluating the mechanistic aspects of the study, including protein trafficking and renal tubular dysfunction. However, I defer to experts in hematopoietic stem cell transplantation and nephrology-specific clinical applications for further assessment of the translational potential of these findings.

Response: We thank the Reviewer for these positive comments, and we have tried addressing most of the limitations raised by the Reviewer.

2. Description of the revisions that have already been incorporated in the transferred manuscript

Please insert a point-by-point reply describing the revisions that were already carried out and included in the transferred manuscript. If no revisions have been carried out yet, please leave this section empty.

Reviewer #1 (Significance (Required)):

1. The authors employ various models of cystinosis, acknowledging the limitations of each and selecting the most suitable one for addressing each research question.

In significance the authors state that this study offers new molecular insights in the endosomal trafficking of NHE3 and provides a new mechanism that may explain the underlying pathogenesis and early onset of the renal Fanconi syndrome in cystinosis. While the described mechanism contributes to the understanding of Fanconi syndrome, I believe it does not fully explain the condition, as other factors are likely involved as well. In the discussion, this is also mentioned by the authors. The significance statement could be modified to make it more clear.

Response: We appreciate the reviewer's comment. We have modified the significance statement as suggested.

Reviewer #2 (Evidence, reproducibility and clarity (Required)):

Major concerns:

Revision Plan

2. Do single or double *ers1* mutants impair Nhx1 localization by interfering with endolysosomal maturation or vesicular dynamics?

Response: We believe this point is already addressed in the current manuscript. As described in the Results section: “Interestingly, two major differences were observed for Nhx1 localizations in the $\Delta ers1S \Delta ers1L$ double mutant, first Nhx1 protruded beyond Vps8 (yellow arrows; Fig. 1C) and second a large fraction of Nhx1 localized into the vacuolar matrix (yellow circle; Fig. 1C).” These observations suggest that the absence of both Ers1 proteins may lead to altered MVB morphology. The mis localization of Nhx1 to the vacuolar lumen further implies that, in the absence of Ers1, Nhx1 might be mistargeted and degraded by vacuolar proteases. While these findings point to a defect at the level of endosomal compartment identity or membrane sorting, there is no evidence of a broad disruption in endolysosomal maturation or trafficking. Vacuolar morphology remains unaffected in the mutants, and no general defects in vesicular dynamics have been observed.

4. Moreover, can VPS1 overexpression rescue the trafficking defects and restore proper endosomal localization of Nhx1 in *ers1* mutants, thereby providing functional evidence for this regulatory pathway?

Response: Vps1 is essential for endosomal membrane fission, but its activity depends on precise recruitment, GTP binding, and interaction with cofactors. Simply overexpressing VPS1 is not sufficient to drive function, as its activation is regulated through spatial localization and GTPase cycling rather than protein abundance. Supporting this, two independent overexpression screens found no phenotypic consequences of VPS1 overexpression: *Sopko et al.* (2006) reported no vacuolar morphology defects in a global screen of 5,500 yeast ORFs,¹ and *Arlt et al.* (2011) similarly observed no endocytic trafficking phenotype or vacuolar alterations upon VPS1 overexpression.² Additionally, *Burston et al.* (2009) did not identify VPS1 as a modulator in their targeted trafficking screen.³ These findings argue that Vps1 levels are not limiting in wild-type cells, and that its recruitment and activation are the key regulatory steps. Thus, overexpressing VPS1 is unlikely to rescue Nhx1 mis localization in *ers1* mutants, where defects likely stem from upstream disruptions in membrane identity or protein recruitment.

6. Extending their findings to a mammalian cell culture system (HEK293), the authors show that CTNS appears to interact with NHE3 and NHE2, but not with NHE1, suggesting a degree of specificity in CTNS-NHE interactions. However, the study does not address whether these interactions occur within the endolysosomal system, nor does it examine the subcellular localization of NHE proteins in the absence of CTNS. Additional experiments are needed to determine whether CTNS deficiency leads to NHE mis localization or altered trafficking patterns, which would further support the proposed role of CTNS in regulating ion transporter compartmentalization and function.

Response: We respectfully believe this has been addressed in the present manuscript. Indeed, we showed the mis-localization of NHE3 in HK-2 cells under both WT and CTNS KO conditions. Then, we showed in proximal tubular cells (PTCs) that the trafficking of NHE3 is altered in cystinosis patient PTCs by TIRFM. In addition, we showed in both PTCs and HK2 cells a co-

Revision Plan

localization in the endosomes in the healthy cells supporting that their interaction occurs in this cellular compartment. This was also shown in the yeast model.

9. Additionally, how do the authors explain the rescue of NHE3 mis localization and functional compartmentalization upon transplantation with wild-type (WT) hematopoietic stem and progenitor cells (HSPCs)? Investigating the effect of HSPC transplantation is important to determine whether restoring functional CTNS expression can correct the trafficking defects in NHE3 and whether this has therapeutic potential for cystinosis patients.

Response: In our earlier work,⁴ we showed that cystinosin, a lysosomal transmembrane protein, is delivered by a mechanism known as vesicular cross-correction. This process is initiated following the differentiation of hematopoietic stem and progenitor cells (HSPCs) into macrophages, which mediate paracrine effects through exosome release and the formation of long membrane projections such as tunneling nanotubes (TNTs). In our previous work, we characterized this cross-correction mechanism, showing that HSPC-derived macrophages not only generate TNTs but also extend nanotubular structures resembling invadopodia that traverse the dense tubular basement membrane to deliver cystinosin directly into diseased proximal tubular cells in *Ctns*^{-/-} mice. Notably, these TNTs enable bidirectional transport—transferring cystinosin-loaded lysosomes into affected cells while retrieving cystine-laden lysosomes for degradation within the macrophages. This reciprocal exchange effectively reduces cystine accumulation, restores lysosomal function, and preserves proximal tubular architecture and kidney function over time. To our knowledge, this is the first demonstration of genetic correction in a lysosomal storage disorder via bidirectional vesicular exchange through TNTs, underscoring the broader therapeutic potential of HSPC-based strategies for diseases caused by defective vesicular proteins. We added this explanation in the Discussion of the revised manuscript.

Reviewer #2 (Evidence, reproducibility and clarity (Required)):

Minor changes:

1. The introduction and discussion should be streamlined and narrowed down, focusing on the key points that directly address the rationale, core findings and significance of the study.

Response: We revised the Introduction and Discussion sections to better highlight the relevance and significance of the study.

3. Functional validation of KO strategies in Figs. 2 and 3, i.e., cystine measurements, expression levels of endocytic receptors, endocytic trafficking and ligand uptake.

Response- The knockout (KO) cells were used in Figure 3, not Figure 2. These cells were generated in the laboratory of our collaborator, Dr. Sergio D. Catz. His team confirmed elevated cystine levels under the *Ctns*-KO condition - while cystine is barely detectable in wild-type cells, it is significantly increased in *CTNS*-KO cells.⁵ They also demonstrated altered expression and localization of key proteins: the chaperone-mediated autophagy (CMA) receptor LAMP2A and the plasma membrane localization of megalin, are both impaired in *CTNS*-KO proximal tubular cells (PTCs).

Revision Plan

4. Some graphs (Fig.3e-h) are not so easy to read and follow. In addition, crucial information regarding the reproducibility and number of biologically independent experiments/samples, and multiple comparisons test used during the statistical analysis.

Response: We revised the graphs in Figures 3e–h to improve clarity and readability. Additionally, we have included details on the number of biologically independent samples and specified the multiple comparisons test used in the statistical analysis.

5. Applying a one-way ANOVA with a small sample size ($n=3$) is generally not recommended. With such a small sample size, the power of the ANOVA test to detect meaningful differences between groups is very low. A low power increases the likelihood of Type II errors, where true differences between groups are not detected.

One-way ANOVA relies on several assumptions:

- i. Normality: The data should be approximately normally distributed within each group. Small sample sizes make it difficult to reliably assess the normality of the data.
- ii. Homogeneity of Variance: The variance within each group should be approximately equal. With small samples, this assumption may be harder to check and violate. It would be better to consider other (non-parametric) approaches or increase the sample size if feasible.

Response: We will increase the sample size as suggested by the reviewer.

Reviewer #3 (Evidence, reproducibility and clarity (Required)):

Major concerns:

2. The authors should discuss the potential transcriptional effects leading to NHE2 overexpression in *CTNS*^{-/-} cells and clarify why NHE3 protein levels are reduced, despite unchanged mRNA expression.

Response: The overexpression of NHE2 in *CTNS*^{-/-} cells may result from compensatory transcriptional responses triggered by the decreased of NHE3 protein at the membrane of the PTCs. Indeed, although NHE3 mRNA levels remain unchanged, the observed reduction in protein levels may be due to the increased degradation of the mis-localized NHE3 protein in *CTNS*-KO cells when trapped in the ER through a process known as ER-associated degradation (ERAD). We will add this potential mechanism in the Discussion of the revised manuscript.

9. Figure 5C: The discrepancy between NHE2 mRNA and protein levels in WT vs. *CTNS*^{-/-} mice is not explained or discussed.

Response: In both *CTNS*-KO HK2 cells and *Ctns*^{-/-} mice, increased expression of NHE2 protein is observed, however, as noted by the reviewer, increased NHE2 mRNA expression is seen only in the *CTNS*-KO HK2 cells. This could be due to the sensitivity of the mRNA detection in cells as opposed to the entire kidney due to tissue complexity. We will add this hypothesis in the Discussion of the revised manuscript.

3. Description of analyses that authors prefer not to carry out

Please include a point-by-point response explaining why some of the requested data or additional analyses might not be necessary or cannot be provided within the scope of a revision. This can be due to time or resource limitations or in case of disagreement about the necessity of such additional data given the scope of the study. Please leave empty if not applicable.

Reviewer #2 (Evidence, reproducibility and clarity (Required)):

Major concerns:

1. Using in vivo imaging and BiFC assay technologies, the authors demonstrate that Ers1 (a functional homolog of CTNS) interacts with Nhx1 (the yeast ortholog of mammalian NHE) in endosomes, suggesting a role for Ers1 in facilitating the intracellular trafficking and compartmentalization of Nhx1. However, the mechanistic details of how Ers1 regulates Nhx1 trafficking remain insufficiently addressed.

Response: We agree that understanding the precise mechanism by which Ers1 influences Nhx1 trafficking would be valuable. However, addressing this question falls outside the scope of the current study. Here, *P. pastoris* served mainly as a genetically accessible system to establish proof-of-principle evidence for an interaction between Ers1 and the yeast homolog of NHE, Nhx1. Our broader interest is in mammalian cells, where CTNS likely affects the trafficking of a wider range of cargos—many of which have no equivalents in yeast. Detailed mechanistic studies in that context will be the focus of future research.

3. Are additional nutrient transporters, beyond Nhx1, similarly mis localized or functionally compromised in the *ers1* mutant background?

Response: We thank the reviewer for this insightful question. Whether other nutrient transporters are mis localized or functionally impaired in the *ers1* mutant background is indeed a relevant point and is currently under investigation, particularly in mammalian cells where our primary interest lies. However, a broader analysis of additional transporters extends beyond the scope of this manuscript.

5. Given the absence of kidney proximal tubule (PT)-like structures in yeast, the authors could have strengthened their conclusions by incorporating the zebrafish model to validate the evolutionary conservation and biological relevance of the Ers1-Nhx1 interaction in the context of renal physiology. The zebrafish pronephros closely recapitulates key features of the mammalian kidney, including epithelial polarity, tight junction complexes, a well-developed endolysosomal network, mitochondria-driven energy metabolism, and the expression of endocytic receptors and nutrient transporters such as CTNS and NHE3. Incorporating this model would have provided a more physiologically relevant system to probe the functional consequences of CTNS-NHE3 interactions in vivo, particularly in the setting of early cystinosis-related PT dysfunction.

Revision Plan

Response: We are using human proximal tubular cells (PTCs) and a mouse model to demonstrate evolutionary conservation; we respectfully believe that these models are more directly relevant to our study than zebrafish.

7. In Fig. 3, the authors' reconciliation of their findings is unclear. Specifically, NHE3 appears to be trapped within the ER in CTNS-deficient HK2 cells.

a. What are the mechanistic factors driving this unexpected ion transporter mis localization in mutant cells?

d. They should also identify potential interaction partners using mass spectrometry or Western blotting, with a focus on proteins involved in trafficking, ER stress response, or endocytosis.

Response: a & d. While the mechanistic basis underlying the unexpected mis localization of ion transporter, NHE3 in CTNS-deficient cells is indeed intriguing, a detailed investigation into these pathways—such as identifying interaction partners via mass spectrometry or Western blotting focused on trafficking, ER stress, or endocytosis—is beyond the scope of the current study.

Reviewer #3 (Evidence, reproducibility and clarity (Required)):

Major concerns:

4. Figures 3F, H: The effect of albumin on megalin/cubilin expression should be assessed.

Response: Studies have shown that albumin stimulation enhances the expression of megalin and cubilin.^{6,7} If the reviewers deem it necessary, we can perform Western blot analyses to re-evaluate the effect of albumin on megalin and cubilin expression in our system.

5. Figures 3F, H: A time-course analysis of albumin endocytosis would be more appropriate for assessing the internalization rate and lysosomal trafficking rather than the current 16-hour endpoint assessment. At this late time point, intracellular albumin levels result from a combination of internalization, lysosomal processing, and exocytosis (as TFEB activation may enhance lysosomal exocytosis).

Response: Previous studies have demonstrated that albumin stimulation enhances the expression of megalin and cubilin, with expression increasing at 12 hours and peaking around 24 hours following treatment with 20 mg/mL BSA.^{6,7} Our objective was to stimulate these receptors to investigate the role of NHE3 in albumin-mediated endocytosis. Given this well-established expression timeline, we believe that an additional time-course experiment may not substantially enhance the current findings.

Reviewer #3 (Significance (Required)):

4. The study would benefit from additional mechanistic insights into the transcriptional regulation of NHE3/NHE2 in cystinosis-deficient cells.

Response: We appreciate the reviewer's insightful suggestion. While we agree that a deeper investigation into the transcriptional regulation of NHE3/NHE2 in cystinosis-deficient cells could yield valuable mechanistic insights, we respectfully consider this beyond the scope of the current study.

References

1. Sopko, R. *et al.* Mapping Pathways and Phenotypes by Systematic Gene Overexpression. *Molecular Cell* **21**, 319–330 (2006).
2. Arlt, H., Perz, A. & Ungermann, C. An Overexpression Screen in *Saccharomyces cerevisiae* Identifies Novel Genes that Affect Endocytic Protein Trafficking. *Traffic* **12**, 1592–1603 (2011).
3. Burston, H. E. *et al.* Regulators of yeast endocytosis identified by systematic quantitative analysis. *Journal of Cell Biology* **185**, 1097–1110 (2009).
4. Naphade, S. *et al.* Brief Reports: Lysosomal Cross-Correction by Hematopoietic Stem Cell-Derived Macrophages Via Tunneling Nanotubes. *Stem Cells* **33**, 301–309 (2015).
5. Zhang, J. *et al.* Chaperone-Mediated Autophagy Upregulation Rescues Megalin Expression and Localization in Cystinotic Proximal Tubule Cells. *Front Endocrinol (Lausanne)* **10**, 21 (2019).
6. Yang, J. *et al.* Expression of renal cubilin and its potential role in tubulointerstitial inflammation induced by albumin overload. *Front. Med. China* **2**, 25–34 (2008).
7. Liu, D. *et al.* Megalin/Cubulin-Lysosome-mediated Albumin Reabsorption Is Involved in the Tubular Cell Activation of NLRP3 Inflammasome and Tubulointerstitial Inflammation. *Journal of Biological Chemistry* **290**, 18018–18028 (2015).

Dear Dr. Cherqui,

Thank you for submitting your manuscript to EMBO Reports, which was previously reviewed at Review Commons.

Referees express interest in the proposed role of cystinosin in regulation of NHE3 and its subcellular localization. However, they also raise concerns that need to be addressed to consider publication in EMBO Reports.

Having looked at all documents, we would like to invite you to submit a revised manuscript as in your revision plan. It is not required to address concerns related to in-depth mechanism (referee#1, points 7a and 7d; referee #2, points 1, 3; referee #3, point 4) and employing a zebrafish model (referee #2, point 5). However, please note that we require strong referee endorsement on the rest of the points for publication here.

Please revise your manuscript with the understanding that the referee concerns (as in their reports) must be fully addressed and their suggestions taken on board. Please address all referee concerns in a complete point-by-point response. Acceptance of the manuscript will depend on a positive outcome of a second round of review. It is EMBO reports policy to allow a single round of major experimental revision only and acceptance or rejection of the manuscript will therefore depend on the completeness of your responses included in the next, final version of the manuscript.

We realize that it is difficult to revise to a specific deadline. In the interest of protecting the conceptual advance provided by the work, we recommend a revision within 3 months. Please discuss the revision progress ahead of this time with me if you require more time to complete the revisions, or if you have questions or comments regarding the revision (also by video chat).

1. A data availability section providing access to data deposited in public databases is missing (where applicable).
2. Your manuscript contains statistics and error bars based on $n=2$. Please use scatter plots in these cases.

You can submit the revision either as a Scientific Report or as a Research Article. For Scientific Reports, the revised manuscript can contain up to 5 main figures and 5 Expanded View figures, and it should not exceed 27000 characters. If the revision leads to a manuscript with more than 5 main figures it will be published as a Research Article. In this case the Results and Discussion section should be separate. If a Scientific Report is submitted, these sections have to be combined. This will help to shorten the manuscript text by eliminating some redundancy that is inevitable when discussing the same experiments twice. In either case, all materials and methods should be included in the main manuscript file.

3) We replaced Supplementary Information with Expanded View (EV) Figures and Tables that are collapsible/expandable online. A maximum of 5 EV Figures can be typeset. EV Figures should be cited as 'Figure EV1, Figure EV2' etc... in the text and their respective legends should be included in the main text after the legends of regular figures.

4) a .docx formatted letter INCLUDING the reviewers' reports and your detailed point-by-point responses to their comments. As part of the EMBO publication's Transparent Editorial Process, EMBO reports publishes online a Review Process File (RPF) to accompany accepted manuscripts. This File will be published in conjunction with your paper and will include the referee reports, your point-by-point response and all pertinent correspondence relating to the manuscript.

<https://www.embopress.org/page/journal/14693178/authorguide#transparentprocess>

5) a complete author checklist, which you can download from our author guidelines <https://www.embopress.org/page/journal/14693178/authorguide>. Please insert information in the checklist that is also reflected in the manuscript. The completed author checklist will also be part of the RPF.

6) Please note that all corresponding authors are required to supply an ORCID ID for their name upon submission of a revised manuscript (. Please find instructions on how to link your ORCID ID to your account in our manuscript tracking system in our Author guidelines

Additional information on source data and instruction on how to label the files are available:

<https://www.embopress.org/page/journal/14693178/authorguide#sourcedata>

9) Our journal encourages inclusion of *data citations in the reference list* to directly cite datasets that were re-used and obtained from public databases. Data citations in the article text are distinct from normal bibliographical citations and should directly link to the database records from which the data can be accessed. In the main text, data citations are formatted as follows: "Data ref: Smith et al, 2001" or "Data ref: NCBI Sequence Read Archive PRJNA342805, 2017". In the Reference list, data citations must be labeled with "[DATASET]". A data reference must provide the database name, accession number/identifiers and a resolvable link to the landing page from which the data can be accessed at the end of the reference. Further instructions are available at <http://www.embopress.org/page/journal/14693178/authorguide#referencesformat>

- the name of the statistical test used to generate error bars and P values,
- the number (n) of independent experiments (please specify technical or biological replicates) underlying each data point,
- the nature of the bars and error bars (s.d., s.e.m.),
- If the data are obtained from n Program fragment delivered error ``Can't locate object method "less" via package "than" (perhaps you forgot to load "than"?) at //ejpvfs23/sites23b/embo/www/letters/embo_decision_revise_and_review.txt line 56.' 2, use scatter blots showing the individual data points.

12) Please also note our reference format:

13) All Materials and Methods need to be described in the main text using our 'Structured Methods' format, which is required for all research articles. According to this format, the Methods section includes a Reagents and Tools Table (listing key reagents, experimental models, software and relevant equipment and including their sources and relevant identifiers) followed by a Methods and Protocols section describing the methods using a step-by-step protocol format. The aim is to facilitate adoption of the methodologies across labs. More information on how to adhere to this format as well as a downloadable template (.docx) for the Reagents and Tools Table can be found in our author guidelines:

I look forward to seeing a revised version of your manuscript when it is ready. Please let me know if you have questions or comments regarding the revision.

Kind regards,

Deniz Senyilmaz Tiebe

Deniz Senyilmaz Tiebe, PhD
Senior Scientific Editor
EMBO Reports

Response to Reviewers:**Manuscript number:** EMBOR-2025-62055V1-T [RC-2025-02907]**Corresponding author:** Stephanie, Cherqui

Reviewer #1 (Evidence, reproducibility and clarity (Required)):

The manuscript describes the dysregulated trafficking of NHE3 in proximal tubular cells of cystinosis patients and its relationship with cystinosin, the primary transporter affected in the disease. This abnormal trafficking primarily disrupts sodium and water reabsorption, contributing to proximal tubulopathy, a key feature of Fanconi Syndrome.

The manuscript is very well-written, with the authors using strong and appropriate methods to validate their findings.

The results support the idea that to effectively treat Fanconi Syndrome (FS), it is not enough to simply eliminate cystine from the lysosomes; rather, the cystinosin-specific functions in the proximal tubular cells (PTCs) need to be restored, which supports the ongoing clinical trial.

Major concern: Even though it has been published previously, it would be crucial to include a more thorough characterization of the cystinosis phenotype and rescue in the mouse model, specifically focusing on cystine accumulation and key Fanconi Syndrome (FS) parameters. While the HSPC transplantation improved protein co-localization, no functional improvements were demonstrated. This lack of functional data leaves the relevance of the potential treatment largely unvalidated and raises concerns about the clinical applicability of the approach.

Response: We thank the reviewer for the constructive comments. As requested, we evaluated kidney function by measuring serum creatinine levels and key Fanconi Syndrome (FS) parameters using previously frozen serum and 24h-urine samples from most of the study mice. The results show that untreated and mock-treated *Ctns*^{-/-} mice displayed significantly elevated serum creatinine and FS features, including polyuria, glucosuria, and sodium wasting compared to WT mice. These abnormalities were corrected in *Ctns*^{-/-} mice transplanted with WT HSPCs. The corresponding data are now included in Figs. 5E–H of the revised manuscript.

Reviewer #1 (Significance (Required)):

The authors employ various models of cystinosis, acknowledging the limitations of each and selecting the most suitable one for addressing each research question. In significance the authors state that this study offers new molecular insights in the endosomal trafficking of NHE3 and provides a new mechanism that may explain the underlying pathogenesis and early onset of the renal Fanconi syndrome in cystinosis. While the described mechanism contributes to the understanding of Fanconi syndrome, I believe it does not fully explain the condition, as other factors are likely involved as well. In the discussion, this is also mentioned by the authors. The significance statement could be modified to make it more clear.

Response: We appreciate the reviewer's comment. We modified the significance statement as suggested, now presented as the main findings of the study.

The study identifies a novel function of cystinosin in regulating NHE protein trafficking, specifically linking NHE3 mislocalization to Fanconi Syndrome (FS). This suggests that restoring NHE3 localization may be essential for treating FS. However, while the study provides important molecular insights, it does not demonstrate functional rescue after HSPC treatment, leaving the effect of restoring expression and localization on FS improvement uncertain.

Response: Functional rescue of the renal Fanconi Syndrome (FS) by WT HSPC transplantation has now been demonstrated by showing rescue of polyuria, glucosuria and sodium loss using previously frozen 24h-urine samples from most of the study mice. Kidney function improvement in the WT HSPC-transplanted was also shown by measuring serum creatinine in WT HSPC-transplanted mice compared to untreated and mock-treated *Ctns*^{-/-} mice in the new Fig. 5E. These data are depicted in Figs. 5F-H in the revised manuscript.

This manuscript will interest researchers in nephrology, molecular biology, and translational medicine, particularly those studying Fanconi Syndrome, cystinosis, and endosomal trafficking. It appeals to both basic scientists investigating NHE protein regulation and cellular transport and clinicians exploring potential therapeutic strategies, including HSPC transplantation for renal disorders.

My expertise lies in kidney models for investigating disease mechanisms and developing novel therapies, with over 10 years of experience studying cystinosis. I am familiar with the limitations of each model described in the manuscript, but I believe they were appropriately chosen for the different research questions. However, my main concern is the lack of data showing the effect of the treatment on the proposed Fanconi Syndrome (FS) phenotype in mice. Demonstrating this is essential, as rescuing FS would significantly strengthen the message and the impact of the proposed treatment.

Response: This has now been completed and the data displayed in the new Figs. 5E-H.

Reviewer #2 (Evidence, reproducibility and clarity (Required)):

Evidence, reproducibility, and clarity

In this manuscript, Khare and colleagues present novel insights into the potential pathways and mechanisms underlying the pathogenesis of cystinosis, a prototypical lysosomal storage disorder caused by the loss of the cystine transporter cystinosin (CTNS). CTNS deficiency leads to early proximal tubule (PT) dysfunction and tubulopathy, which progresses to chronic kidney disease and multisystem complications later in life.

Using genetically modified organisms, patient-derived cells, and kidney tissue from individuals with cystinosis, the authors demonstrate that CTNS, beyond its established role in lysosomal cystine transport, also appears to regulate the trafficking and compartmentalization of specific members of the sodium-hydrogen exchanger (Na⁺/H⁺) family (NHEs). Notably, they show that the

absence of CTNS results in the mislocalization and impaired vesicular trafficking of NHE3, contributing to the urinary loss of sodium and other essential nutrients. This unexpected interaction seems to be evolutionarily conserved, as supported by compelling evidence from yeast model studies.

Furthermore, proof-of-concept experiments suggest that the observed phenotypic changes are dependent on CTNS, as demonstrated by rescue strategies involving transplantation of wild-type hematopoietic stem and progenitor cells (HSPCs), which reintroduce functional CTNS and restore PT homeostasis.

Despite these novel findings, the study is limited by a lack of molecular and structural detail on the mechanisms underlying CTNS-NHE3 interactions and their role in organelle homeostasis and trafficking-related pathways. As such, the conclusions remain largely observational and descriptive, though they are well supported by the presented data.

Major concerns:

1. Using in vivo imaging and BiFC assay technologies, the authors demonstrate that Ers1 (a functional homolog of CTNS) interacts with Nhx1 (the yeast ortholog of mammalian NHE) in endosomes, suggesting a role for Ers1 in facilitating the intracellular trafficking and compartmentalization of Nhx1. However, the mechanistic details of how Ers1 regulates Nhx1 trafficking remain insufficiently addressed.

Response: We thank the reviewer for their thoughtful and constructive comments. We agree that understanding the precise mechanism by which Ers1 influences Nhx1 trafficking would be valuable. However, addressing this question falls outside the scope of the current study. Here, *P. pastoris* served mainly as a genetically accessible system to establish proof-of-principle evidence for an interaction between Ers1 and the yeast homolog of NHE, Nhx1. Our broader interest is in mammalian cells, where CTNS likely affects the trafficking of a wider range of cargos—many of which have no equivalents in yeast. Detailed mechanistic studies in that context will be the focus of future research. - This response has been approved by Dr. Deniz Senyilmaz Tiebe.

Do single or double *ers1* mutants impair Nhx1 localization by interfering with endolysosomal maturation or vesicular dynamics?

Response: We believe this point is already addressed in the current manuscript. As described in the Results section: “Interestingly, two major differences were observed for Nhx1 localizations in the $\Delta ers1S \Delta ers1L$ double mutant, first Nhx1 protruded beyond Vps8 (yellow arrows; Fig. 1C) and second a large fraction of Nhx1 localized into the vacuolar matrix (yellow circle; Fig. 1C).” These observations suggest that the absence of both Ers1 proteins may lead to altered MVB morphology. The mis localization of Nhx1 to the vacuolar lumen further implies that, in the absence of Ers1, Nhx1 might be mistargeted and degraded by vacuolar proteases. While these findings point to a defect at the level of endosomal compartment identity or membrane sorting, there is no evidence of a broad disruption in endolysosomal maturation or trafficking. Vacuolar morphology remains unaffected in the mutants, and no general defects in vesicular dynamics have been observed.

Are additional nutrient transporters, beyond Nhx1, similarly mislocalized or functionally compromised in the *ers1* mutant background?

Response: We thank the reviewer for this insightful question. Whether other nutrient transporters are mislocalized or functionally impaired in the *ers1* mutant background is indeed a relevant point and is currently under investigation, particularly in mammalian cells where our primary interest lies. However, a broader analysis of additional transporters extends beyond the scope of this manuscript. - This response has been approved by Dr. Deniz Senyilmaz Tiebe.

Moreover, can VPS1 overexpression rescue the trafficking defects and restore proper endosomal localization of Nhx1 in *ers1* mutants, thereby providing functional evidence for this regulatory pathway?

Response: Vps1 is essential for endosomal membrane fission, but its activity depends on precise recruitment, GTP binding, and interaction with cofactors. Simply overexpressing VPS1 is not sufficient to drive function, as its activation is regulated through spatial localization and GTPase cycling rather than protein abundance. Supporting this, two independent overexpression screens found no phenotypic consequences of VPS1 overexpression: *Sopko et al.* reported no vacuolar morphology defects in a global screen of 5,500 yeast ORFs, (*Sopko et al.*, 2006) and *Arlt et al.* similarly observed no endocytic trafficking phenotype or vacuolar alterations upon VPS1 overexpression. (*Arlt et al.*, 2011) Additionally, *Burston et al.* did not identify VPS1 as a modulator in their targeted trafficking screen. (*Burston et al.*, 2009) These findings argue that Vps1 levels are not limiting in wild-type cells, and that its recruitment and activation are the key regulatory steps. Thus, overexpressing VPS1 is unlikely to rescue Nhx1 mis localization in *ers1* mutants, where defects likely stem from upstream disruptions in membrane identity or protein recruitment.

2. Given the absence of kidney proximal tubule (PT)-like structures in yeast, the authors could have strengthened their conclusions by incorporating the zebrafish model to validate the evolutionary conservation and biological relevance of the *Ers1-Nhx1* interaction in the context of renal physiology. The zebrafish pronephros closely recapitulates key features of the mammalian kidney, including epithelial polarity, tight junction complexes, a well-developed endolysosomal network, mitochondria-driven energy metabolism, and the expression of endocytic receptors and nutrient transporters such as CTNS and NHE3. Incorporating this model would have provided a more physiologically relevant system to probe the functional consequences of CTNS-NHE3 interactions *in vivo*, particularly in the setting of early cystinosis-related PT dysfunction.

Response: In addition to the yeast model, we are using human proximal tubular cells (PTCs) and a mouse model of cystinosis to demonstrate evolutionary conservation and biological relevance of the *Ers1-Nhx1* interaction in this manuscript. We respectfully believe that these models are more directly relevant to our study than zebrafish. - This response has been approved by Dr. Deniz Senyilmaz Tiebe.

3. Extending their findings to a mammalian cell culture system (HEK293), the authors show that CTNS appears to interact with NHE3 and NHE2, but not with NHE1, suggesting a degree of specificity in CTNS-NHE interactions. However, the study does not address whether these interactions occur within the endolysosomal system, nor does it examine the subcellular

localization of NHE proteins in the absence of CTNS. Additional experiments are needed to determine whether CTNS deficiency leads to NHE mislocalization or altered trafficking patterns, which would further support the proposed role of CTNS in regulating ion transporter compartmentalization and function.

Response: We respectfully believe this has been addressed in the present manuscript. Indeed, we showed the mislocalization of NHE3 in HK-2 cells under both WT and CTNS KO conditions. Then, we showed in proximal tubular cells (PTCs) that the trafficking of NHE3 is altered in cystinosis patient PTCs by TIRFM. In addition, we showed in both PTCs and HK2 cells a co-localization in the endosomes in the healthy cells supporting that their interaction occurs in this cellular compartment. This was also shown in the yeast model.

4. In Fig. 3, the authors' reconciliation of their findings is unclear. Specifically, NHE3 appears to be trapped within the ER in CTNS-deficient HK2 cells.

What are the mechanistic factors driving this unexpected ion transporter mislocalization in mutant cells?

Response: While the mechanistic basis underlying the unexpected mislocalization of ion transporter, NHE3 in CTNS-deficient cells is indeed intriguing, a detailed investigation into these pathways is beyond the scope of the current study. - This response has been approved by Dr. Deniz Senyilmaz Tiebe.

Are these changes a direct consequence of CTNS loss, occurring independently of cystine storage (e.g., after cysteamine treatment)?

Response: We believe that NHE3 mislocalization and altered trafficking are a direct consequence of CTNS loss, as reintroduction of the *CTNS* gene into the cells rescues these defects. To prove that cystine storage does not have any impact in this phenomenon, we treated the HK2 and CTNS-KO HK2 cells with cysteamine as suggested by the reviewer and the data showed that cysteamine treatment does not rescue NHE3 subcellular localization in the CTNS-deficient cells. These data have been included in the new Expanded View Figure EV3D in the revised manuscript.

The authors should transfect CTNS-deficient HK2 cells with tagged WT-CTNS and perform co-immunoprecipitation (co-IP) to confirm the interaction between CTNS and NHE3.

Response: We generated stable CTNS-deficient HK-2 cells using lentiviral vectors (LV) expressing NHE3 fused to GFP (LV-NHE3-GFP) and CTNS or CTNS-LKG fused to DsRed (LV-CTNS-DsRed; LV-CTNS-LKG-DsRed). Immunoprecipitation (IP) assays with GFP or DsRed tags from whole-cell lysates revealed reciprocal co-precipitation of NHE3-GFP with both cystinosin-DsRed and cystinosinLKG-DsRed. These data are included in the new Expanded View Fig. EV2A in the revised manuscript.

They should also identify potential interaction partners using mass spectrometry or Western blotting, with a focus on proteins involved in trafficking, ER stress response, or endocytosis.

Response: While the mechanistic basis underlying the unexpected mislocalization of ion transporter, NHE3 in CTNS-deficient cells is indeed intriguing, a detailed investigation into these pathways—such as identifying interaction partners via mass spectrometry or Western blotting focused on trafficking, ER stress, or endocytosis—is beyond the scope of the current study. - This response has been approved by Dr. Deniz Senyilmaz Tiebe.

5. Does NHE3 accumulate within the ER, as observed in HK2 cells, in kidney tubular cells derived from cystinosis patients, in the proximal tubular (PT) segments of *Ctns*KO mice, and kidney samples from human cystinosis patients? Understanding whether this mislocalization of NHE3 occurs in these clinically relevant models is crucial, as it could provide insights into the pathological mechanisms underlying cystinosis and its impact on kidney function. If similar mislocalization is observed in patient-derived cells and animal models, it would strengthen the link between CTNS deficiency and disrupted ion transporter trafficking, which may contribute to the renal dysfunction seen in cystinosis.

Response: We performed immunofluorescence analysis in murine and human kidney to assess the subcellular localization of NHE3 in CTNS-deficient proximal tubular cells using co-staining with endoplasmic reticulum (ER) marker, SERCA2-ATPase. These data are included in the new Expanded View Figs. EV5B and EV6 in the revised manuscript. Interestingly, NHE3 and the ER marker, SERCA2-ATPase, exhibited a negative correlation in WT kidney, consistent with proper trafficking (Fig. EV5B). In contrast, untreated and Mock-transplanted *Ctns*^{-/-} kidneys showed a shift toward a positive correlation, indicating abnormal retention of NHE3 within the ER (Fig. EV5B). Remarkably, HSPCs transplantation restored normal NHE3 trafficking and localization, reducing its ER entrapment, as demonstrated by Pearson's correlation analysis. (Fig. EV5B). Similarly, in the human normal (control) kidney tissue, NHE3 and the ER marker SERCA2-ATPase were negatively correlated, whereas positive correlation was observed in the cystinosis patient kidney (Fig. EV6).

6. Additionally, how do the authors explain the rescue of NHE3 mislocalization and functional compartmentalization upon transplantation with wild-type (WT) hematopoietic stem and progenitor cells (HSPCs)? Investigating the effect of HSPC transplantation is important to determine whether restoring functional CTNS expression can correct the trafficking defects in NHE3 and whether this has therapeutic potential for cystinosis patients.

Response: In our earlier work, (Naphade *et al*, 2015) we showed that cystinosin, a lysosomal transmembrane protein, is delivered by a mechanism known as vesicular cross-correction. This process is initiated following the differentiation of hematopoietic stem and progenitor cells (HSPCs) into macrophages, which mediate paracrine effects through exosome release and the formation of long membrane projections such as tunneling nanotubes (TNTs). In our previous work, we characterized this cross-correction mechanism, showing that HSPC-derived macrophages not only generate TNTs but also extend nanotubular structures resembling invadopodia that traverse the dense tubular basement membrane to deliver cystinosin directly into diseased proximal tubular cells in *Ctns*^{-/-} mice. Notably, these TNTs enable bidirectional transport—transferring cystinosin-loaded lysosomes into affected cells while retrieving cystine-laden lysosomes for degradation within the macrophages. This reciprocal exchange effectively reduces cystine accumulation, restores lysosomal function, and preserves proximal tubular

architecture and kidney function over time. To our knowledge, this is the first demonstration of genetic correction in a lysosomal storage disorder via bidirectional vesicular exchange through TNTs, underscoring the broader therapeutic potential of HSPC-based strategies for diseases caused by defective vesicular proteins. We added this explanation in the Discussion of the revised manuscript.

7. Finally, what about the levels of sodium in the urine of transplanted KO mice? Monitoring sodium excretion in these mice after transplantation can provide functional evidence of transporter recovery. This would help determine if the restoration of NHE3 localization leads to an improvement in sodium reabsorption and overall kidney function, offering further insight into the therapeutic benefits of HSPC-based interventions for cystinosis.

Response: We assessed urinary sodium levels in mice following transplantation compared to WT and *Ctns*^{-/-} controls. The data demonstrated significant increase of sodium in untreated and mock-treated *Ctns*^{-/-} mice compared to WT mice, and this defect was rescued in the WT HSPCs-transplanted *Ctns*^{-/-} mice; the graph is included in the new Fig. 5H in the revised manuscript.

Minor changes:

1. The introduction and discussion should be streamlined and narrowed down, focusing on the key points that directly address the rationale, core findings and significance of the study.

Response: We revised the Introduction and Discussion sections to better highlight the relevance and significance of the study.

2. Quantification of the colocalized signals in Fig. 1a-d.

Response: The quantification of colocalized and non-colocalized signals with Vps8 corresponding to Figs. 1A, C, D has been included in Figs. 1E, 1F and Expanded View Figures EV1D–G in the revised manuscript. In addition, the previous images have been replaced with new images to provide improved visual clarity and more effective data presentation.

3. Functional validation of KO strategies in Figs. 2 and 3, i.e., cystine measurements, expression levels of endocytic receptors, endocytic trafficking and ligand uptake.

Response- The knockout (KO) cells were used in Fig. 3, not Fig. 2. These cells were generated in the laboratory of our collaborator, Dr. Sergio D. Catz. His team confirmed elevated cystine levels under the *Ctns*-KO condition - while cystine is barely detectable in wild-type cells, it is significantly increased in *CTNS*-KO cells (Zhang *et al*, 2019a). We highlighted this important feature of the cell model in the Methods section-mammalian cells of the revised manuscript.

4. Some graphs (Fig.3e-h) are not so easy to read and follow. In addition, crucial information regarding the reproducibility and number of biologically independent experiments/samples, and multiple comparisons test used during the statistical analysis.

Response: We revised the graphs in Figs. 3E-H to improve clarity and readability. Additionally, we have included details on the number of biologically independent samples and specified the multiple comparisons test used in the statistical analysis.

5. Applying a one-way ANOVA with a small sample size ($n=3$) is generally not recommended. With such a small sample size, the power of the ANOVA test to detect meaningful differences between groups is very low. A low power increases the likelihood of Type II errors, where true differences between groups are not detected.

One-way ANOVA relies on several assumptions:

i. Normality: The data should be approximately normally distributed within each group. Small sample sizes make it difficult to reliably assess the normality of the data.

ii. Homogeneity of Variance: The variance within each group should be approximately equal. With small samples, this assumption may be harder to check and violate. It would be better to consider other (non-parametric) approaches or increase the sample size if feasible.

Response: We agree with the reviewer that the sample size of $n = 3$ in Fig. 3F was too small to support reliable ANOVA statistics. Accordingly, we increased the sample size to $n = 6$ to strengthen the statistical analysis. The updated data have been incorporated into Fig. 3F in the revised manuscript.

Reviewer #2 (Significance (Required)):

Although a detailed mechanistic understanding remains to be fully elucidated, this study significantly advances our understanding of the biological roles of lysosomal transporters—specifically CTNS—in maintaining tissue homeostasis and contributing to disease pathology. Importantly, it highlights a novel functional link between CTNS and the regulation of nutrient transporter trafficking, suggesting new therapeutic avenues. These findings have broad relevance for the basic research community, particularly in the fields of lysosome biology and transport physiology, and they lay a critical foundation for the development of targeted therapies aimed at restoring transporter function in cystinosis and potentially other lysosomal storage disorders.

Response: We sincerely thank the reviewer for the positive and encouraging feedback.

Reviewer #3 (Evidence, reproducibility and clarity (Required)):

The manuscript titled "Cystinosis is involved in Na^+/H^+ Exchanger 3 trafficking in the proximal tubular cells: new insights into the renal Fanconi syndrome in cystinosis" by Veenita Khare and colleagues explores a previously unknown function of cystinosis in regulating sodium/hydrogen (Na^+/H^+) exchanger proteins within endosomes. Their findings indicate that cystinosis influences the localization, movement, and sodium absorption of NHE3 in proximal tubular cells (PTCs). Remarkably, reintroducing CTNS into CTNS-deficient PTCs corrected these abnormalities, whereas CTNS-LKG, an isoform found in lysosomes and the plasma membrane, did not. Furthermore, NHE3 mislocalization was confirmed in *Ctns*^{-/-} mice as well as in kidney tissue from

cystinosis patients. Notably, transplantation of wild-type hematopoietic stem and progenitor cells into *Ctns*^{-/-} mice restored NHE3 expression at the brush border membrane. While this research sheds light on a novel role of cystinosin in NHE3 trafficking and potentially expands the known cellular dysfunctions associated with CTNS loss of function, several major concerns must be addressed before this manuscript can be considered for publication.

Major concerns:

- The authors demonstrate that exogenously expressed cystinosin interacts with exogenously expressed NHE3 and NHE2, but not with NHE1. However, the Western blot presented in Figure S2 (now Fig EV2B) is unconvincing. The membranes from IP:GFP blotted with anti-DsRed antibody and from IP:DsRed blotted with anti-GFP antibody are completely white, with no visible background. The authors should provide a longer exposure of the membrane to determine whether background signals or non-specific bands appear.

Response: We thank the reviewer for their thoughtful and constructive comments. Please find the longer exposure blot images provided below for your review. On the right-hand side, please see the raw blots and on the left-hand side the blots probed with the respective antibodies.

IP:GFP

**IP:GFP
blotted with anti-GFP antibody**

**IP:GFP
blotted with anti-DsRed
antibody**

IP:DsRed

IP:DsRed
blotted with anti-DsRed
antibody

IP:DsRed
blotted with anti-GFP
antibody

- The authors should discuss the potential transcriptional effects leading to NHE2 overexpression in *CTNS*^{-/-} cells and clarify why NHE3 protein levels are reduced, despite unchanged mRNA expression.

Response: The overexpression of NHE2 in *CTNS*^{-/-} cells may result from compensatory transcriptional responses triggered by the decreased of NHE3 protein at the membrane of the PTCs. Indeed, although NHE3 mRNA levels remain unchanged, the observed reduction in protein levels may be due to the increased degradation of the mislocalized NHE3 protein in *CTNS*-KO cells when trapped in the ER through a process known as ER-associated degradation (ERAD). We added this potential mechanism in the Discussion of the revised manuscript.

- The data in Figures 3C, 3D, and S3, along with the corresponding quantitative analyses, are unconvincing due to the broad and diffuse NHE3 expression observed in both WT and *CTNS*^{-/-} cells. The authors should evaluate NHE3 expression levels before assessing colocalization with organelle markers and consider performing a cycloheximide treatment to inhibit de novo protein synthesis before analyzing NHE3 localization. Additionally, the images provided do not accurately represent the quantitative analysis, as they display extreme conditions (high or zero colocalization), which do not effectively support the intended conclusions.

Response: We did not observe differences in NHE3 expression levels in these cells. As suggested, we included Western blot analysis of NHE3-GFP expression in Expanded View Fig EV3A in the revised manuscript and provided different representative images for Figs. 3C, 3D, and Expanded

View Figs. EV3B & EV3C. In addition, we have included enlarged inset views (provided twice for emphasis) in all the above figures to improve visualization and clarity.

- Figures 3F, H: The effect of albumin on megalin/cubilin expression should be assessed.

Response: As suggested, we included Western data to evaluate the effect of albumin on megalin expression in our system. We observed that 16-hour albumin increased megalin expression in the WT cells, but not in the CTNS^{-/-} HK2 cells, consistent with the reported defects in megalin trafficking and expression in cystinosis cells (Zhang *et al*, 2019b; Raggi *et al*, 2014; Gaide Chevronnay *et al*, 2014). This data is presented in Fig. EV4A in the revised manuscript. Since NHE3 specifically interacts with megalin in proximal tubules (Biemesderfer *et al*, 1999), these findings correlate with the abnormal NHE3 trafficking to the brush border in cystinosis models, which may also contribute to the loss of megalin and impaired albumin uptake. This hypothesis was stated in the Discussion and further emphasized in the revised manuscript.

- Figures 3F, H: A time-course analysis of albumin endocytosis would be more appropriate for assessing the internalization rate and lysosomal trafficking rather than the current 16-hour endpoint assessment. At this late time point, intracellular albumin levels result from a combination of internalization, lysosomal processing, and exocytosis (as TFEB activation may enhance lysosomal exocytosis).

Response: As suggested by the reviewer, we performed a time-course analysis of albumin endocytosis, which is included in Expanded View Fig. EV4B of the revised manuscript. The results show that Ctns^{-/-} cells exhibit reduced albumin uptake as early as 1-hour post-stimulation, and this difference persists up to 16 hours compared with WT controls. These findings indicate that the defect in albumin internalization is evident from the early stages of endocytosis and is maintained over time. This temporal pattern supports the role of cystinosin in regulating NHE3-mediated albumin endocytosis and trafficking.

- Figures 3G, H: EIPA is used at a very high concentration, which may lead to non-specific effects. A more targeted approach, such as siRNA-mediated NHE3 downregulation, could provide more reliable results.

Response: We performed NHE3 knockdown using an shRNA-mediated approach and evaluated its effect on intracellular sodium (Na⁺) content and albumin uptake as suggested by the Reviewer. The NHE3 knockdown was verified by qPCR and the data displayed in Expanded View Fig. EV4C. Similar results were observed with NHE3 shRNA as previously seen with EIPA, showing a significant decrease in Na⁺ and albumin uptake in WT HK2 cells, but not in CTNS-deficient HK2 cells, which already exhibited markedly reduced uptake compared to WT. These data have been included in Figs. 3G,H in the revised manuscript; the graphs with EIPA have been moved to the Expanded View Fig. EV4D,E.

- Figures 3E, G: NHE3 levels at the plasma membrane should be assessed using biochemical approaches (e.g., membrane protein biotinylation) or imaging techniques such as confocal or TIRF microscopy.

Response: Obtaining high-resolution images suitable for quantifying NHE3 at the plasma membrane has been challenging. Nevertheless, we quantified NHE3 localization at the brush border of proximal tubules *in vivo* in both CTNS-deficient and control mouse and human kidneys. The results show that NHE3 is properly localized at the plasma membrane in healthy control mice and humans, whereas it is mislocalized to the ER in *Ctns*^{-/-} mice and cystinosis patients. These findings are presented in Figs. 5, 6, and EV5, EV6 of the revised manuscript.

- Figure 4: The claims made are not clearly supported by the provided data. Instead of pseudo-TIRF imaging, laser scanning confocal live imaging may be more suitable for assessing vesicle tracking and movement, unless vesicle fusion or endocytosis from the plasma membrane is the primary focus. Additionally, analysis should be conducted on cells expressing similar NHE3 levels to ensure consistency.

Response: Pseudo-TIRFM (oblique illumination), which illuminates approximately 400 nm into the sample, is well-suited for our study as the cells used are relatively flat and approximately 400 nm in depth. Given this geometry, pseudo-TIRFM provides high-contrast imaging of vesicle dynamics in the cell body near the basal membrane and performs better than confocal microscopy for our purposes as its gentle illumination avoids photobleaching. Furthermore, we have ensured that analyses were performed on cells expressing comparable levels of NHE3 to maintain consistency across experimental conditions. Also, as suggested, we have provided improved, color-enhanced representative images illustrating NHE3 trafficking in normal PTCs, cystinosis patient-derived PTCs (CT PTCs), and CT PTCs transduced with LV-CTNS-DsRed or LV-CTNS-LKG-DsRed for the revised Fig. 4. We would also like to mention that the still images presented are representative; the accompanying videos (Supplementary Movies 1–4) provide a more detailed visualization and deeper insight into these trafficking dynamics. We specified this point in the revised manuscript.

- Figure 5C: The discrepancy between NHE2 mRNA and protein levels in WT vs. CTNS^{-/-} mice is not explained or discussed.

Response: In both CTNS-KO HK2 cells and *Ctns*^{-/-} mice, NHE2 protein expression is increased. However, as the reviewer noted, elevated NHE2 mRNA expression is observed only in the CTNS-KO HK2 cells. This discrepancy may reflect differences in mRNA detection sensitivity between isolated cells and whole kidney tissue, the latter being more complex and heterogeneous. We added this hypothesis in the Discussion of the revised manuscript.

- Figure 5D: The provided images are highly variable and not convincing. The magnifications used for the images correlate with the quantitative analysis, but other regions of the section do not align with the inset, leading to inconsistencies. The restored CTNS expression by HSCGT should be assessed in tubules with correct NHE3 localization (TEST samples).

Response: Quantification was performed using images from entire kidney sections acquired at lower magnification and stitched together to ensure comprehensive analysis as shown in Expanded View Fig. EV5A. For illustrative purposes, we used higher magnification images to highlight specific features. We have provided improved representative images to enhance clarity in Fig. 5D in the revised manuscript.

Reviewer #3 (Significance (Required)):

This study presents a novel function of cystinosin in regulating the trafficking of Na⁺/H⁺ exchanger 3 (NHE3) in proximal tubular cells (PTCs). By demonstrating how cystinosin deficiency leads to NHE3 mislocalization and how hematopoietic stem cell transplantation can restore its proper expression, the authors provide new insights into renal Fanconi syndrome in cystinosis.

Strengths

The study identifies a previously unrecognized role of cystinosin in NHE3 endosomal trafficking, adding a new mechanistic layer to our understanding of cystinosis-related renal dysfunction. The use of multiple experimental models, including *Ctns*^{-/-} mice, cystinosis patient kidney samples, and various in vitro approaches, strengthens the validity of the findings. The therapeutic implications of hematopoietic stem cell transplantation in rescuing NHE3 expression provide a potential translational perspective that may be of interest for future clinical research.

Response: We sincerely thank the reviewer for the comments.

Limitations and Areas for Improvement

Some data presentations, particularly Western blot images and immunofluorescence quantifications, require further clarification to support the claims made in the manuscript.

Response: We provided raw blots of longer exposure for Expanded View Fig. EV2B in page no-9-10 of this document. Also, we provided better representative images for Figs. 3C,D, 5D and Expanded View Figs. EV3B-C, and missing quantifications especially for Fig. 1.

The discrepancy between NHE2 mRNA and protein levels in *Ctns*^{-/-} mice remains unexplained and should be addressed.

Response: In both CTNS-KO HK2 cells and *Ctns*^{-/-} mice, NHE2 protein expression is increased. However, as the reviewer noted, elevated NHE2 mRNA expression is observed only in the CTNS-KO HK2 cells. This discrepancy may reflect differences in mRNA detection sensitivity between isolated cells and whole kidney tissue, the latter being more complex and heterogeneous. We added this hypothesis in the Discussion of the revised manuscript.

The choice of experimental conditions (e.g., EIPA concentration, endpoint albumin endocytosis assessment) may lead to non-specific effects and should be refined.

Response: As specific testing, we performed shRNA-mediated NHE3 knockdown to assess its effect on intracellular Na⁺ levels and albumin uptake. Similar results were observed with NHE3 shRNA as previously seen with EIPA, showing a significant decrease in Na⁺ and albumin uptake in WT HK2 cells, but not in CTNS-deficient HK2 cells, which already exhibited markedly reduced

uptake compared to WT. These data are presented in Figs. 3G,H and the data confirming the knockdown of NHE3 is presented in Expanded View Fig. EV4C. We also included a time-course analysis of albumin endocytosis showing reduced uptake in *Ctns*^{-/-} cells beginning at 1 hour, with the difference persisting through 16 hours. These data are displayed in Expanded View Fig. EV4B in the revised manuscript.

The study would benefit from additional mechanistic insights into the transcriptional regulation of NHE3/NHE2 in cystinosis-deficient cells.

Response: We appreciate the reviewer's insightful suggestion. While we agree that a deeper investigation into the transcriptional regulation of NHE3/NHE2 in cystinosis-deficient cells could yield valuable mechanistic insights, we respectfully consider this beyond the scope of the current study. - This response has been approved by Dr. Senyilmaz Tiebe.

Advance in the Field

This study extends the current understanding of cystinosis beyond its classical role in lysosomal cystine transport. To our knowledge, this is the first report demonstrating that cystinosis directly affects NHE3 trafficking in PTCs, a finding with significant implications for renal physiology in cystinosis. The insights provided here are both mechanistic (identifying a new trafficking function of cystinosis) and potentially translational (suggesting stem cell-based approaches as a therapeutic avenue).

Audience and Impact

This work will primarily interest specialized audiences in nephrology, cell biology (lysosomal disease and trafficking), and rare genetic disorders. More specifically:
Basic researchers studying renal physiology, lysosomal function, and protein trafficking will find the mechanistic insights valuable.
Translational/clinical researchers investigating cystinosis and renal Fanconi syndrome may be particularly interested in the therapeutic implications of hematopoietic stem cell transplantation.
Broader nephrology community may also benefit from understanding how lysosomal dysfunction impacts sodium homeostasis.
While the study is largely mechanistic, the translational component may expand its relevance beyond the specific field of cystinosis research.

My expertise lies in cell biology, renal physiology, lysosomal trafficking, and rare genetic disorders. I am comfortable evaluating the mechanistic aspects of the study, including protein trafficking and renal tubular dysfunction. However, I defer to experts in hematopoietic stem cell transplantation and nephrology-specific clinical applications for further assessment of the translational potential of these findings.

Response: We thank the Reviewer for these positive comments.

References

- Arlt H, Perz A & Ungermann C (2011) An Overexpression Screen in *Saccharomyces cerevisiae* Identifies Novel Genes that Affect Endocytic Protein Trafficking. *Traffic* 12: 1592–1603
- Biemesderfer D, Nagy T, DeGray B & Aronson PS (1999) Specific Association of Megalin and the Na⁺/H⁺ Exchanger Isoform NHE3 in the Proximal Tubule. *Journal of Biological Chemistry* 274: 17518–17524
- Burston HE, Maldonado-Báez L, Davey M, Montpetit B, Schluter C, Wendland B & Conibear E (2009) Regulators of yeast endocytosis identified by systematic quantitative analysis. *Journal of Cell Biology* 185: 1097–1110
- Gaide Chevronnay HP, Janssens V, Van Der Smissen P, N’Kuli F, Nevo N, Guiot Y, Levtchenko E, Marbaix E, Pierreux CE, Cherqui S, *et al* (2014) Time Course of Pathogenic and Adaptation Mechanisms in Cystinotic Mouse Kidneys. *Journal of the American Society of Nephrology* 25: 1256–1269
- Liu D, Wen Y, Tang T-T, Lv L-L, Tang R-N, Liu H, Ma K-L, Crowley SD & Liu B-C (2015) Megalin/Cubulin-Lysosome-mediated Albumin Reabsorption Is Involved in the Tubular Cell Activation of NLRP3 Inflammasome and Tubulointerstitial Inflammation. *Journal of Biological Chemistry* 290: 18018–18028
- Naphade S, Sharma J, Gaide Chevronnay HP, Shook MA, Yeagy BA, Rocca CJ, Ur SN, Lau AJ, Courtoy PJ & Cherqui S (2015) Brief Reports: Lysosomal Cross-Correction by Hematopoietic Stem Cell-Derived Macrophages Via Tunneling Nanotubes. *Stem Cells* 33: 301–309
- Raggi C, Luciani A, Nevo N, Antignac C, Terry S & Devuyst O (2014) Dedifferentiation and aberrations of the endolysosomal compartment characterize the early stage of nephropathic cystinosis. *Human Molecular Genetics* 23: 2266–2278
- Sopko R, Huang D, Preston N, Chua G, Papp B, Kafadar K, Snyder M, Oliver SG, Cyert M, Hughes TR, *et al* (2006) Mapping Pathways and Phenotypes by Systematic Gene Overexpression. *Molecular Cell* 21: 319–330
- Yang J, He Y, Shen H, Ding H, Li K & Wang H (2008) Expression of renal cubilin and its potential role in tubulointerstitial inflammation induced by albumin overload. *Front Med China* 2: 25–34
- Zhang J, He J, Johnson JL, Rahman F, Gavathiotis E, Cuervo AM & Catz SD (2019) Chaperone-Mediated Autophagy Upregulation Rescues Megalin Expression and Localization in Cystinotic Proximal Tubule Cells. *Front Endocrinol (Lausanne)* 10: 21

Dear Dr. Cherqui

Thank you for the submission of your revised manuscript to EMBO reports. We have now received the full set of referee reports that is copied below.

As you will see, all referees are very positive about the study and request only minor changes to improve a subset of confocal imaging data.

From the editorial side, there are also a few things that we need before we can proceed with the official acceptance of your study.

- Regarding the Author Contributions, we now use CRediT to specify the contributions of each author in the journal submission system. Therefore, please remove the Author Contributions from the manuscript file and make sure that the author contributions in our online manuscript tracking system are correct and up-to-date. The information you specified in the system will be automatically retrieved and typeset into the article. You can enter additional information in the free text box provided, if you wish. See also our guide to authors <https://www.embopress.org/page/journal/14693178/authorguide#authorshipguidelines>.
- The information on funding in the manuscript text and in the online manuscript tracking system must be congruent. In this regard we noted that the following items are missing in the system: NS108965, R01NS135162, R01-AG086443, R01DK110162, P01HL152958; the MPS Society, and the Friedreich's Ataxia Research Alliance (FARA), P30-NS047101, S10OD030417.
- The in-text callout to Supplementary movies 1-4 needs to be updated to Movies EV1-EV4.
- Each movie needs to be provided with its legend (zipped up together) so that we have 4 zip folders; the correct nomenclature should be Movie EV1-Movie EV4 ("Supplementary movies 1-4" is not correct).
- Please remove the Reagents and Tools table from the manuscript and upload it as a separate file.
- Please remove the paragraph "Main findings of the study" from the manuscript. The text could be edited and used for the synopsis text/bullet points (see below).
- We received bounced email alerts for the co-author Rola Chen - xic089@health.ucsd.edu. Please either remove the author from the author list in the system and then add her back using the correct/new email or send us the correct/new email and we will update the account accordingly.
- Please address the following points in the figure legends and please highlight the changes:
 - Please provide the exact p values in the legends of figures 1e, f; 3a-h; 4b-d; 5b-h; 6; EV 3b-d; EV 4a-e; EV 5b; EV 6. (Exact p-values are not required for small values <0.0001).
 - Please indicate the statistical test used for data analysis in the legends of figures 1e, f
 - Please provide information related to n in the legends of figures 4b; 5b-h EV 1d, f, g, h; EV 3a-d; EV 4a-e; EV 5b; EV 6
 - Please define the error bars in the legends of figures EV 1f, g, h
 - Please define the scale bar for figures EV 1c
 - Please note that the yellow and white arrows are not defined in the legend of figure 1a; EV 1c. This needs to be rectified.
- The scale bars in Figure 2B, 3C, 5D, Fig 6, Fig EV3B, EV3C, EV5A, EV5B are difficult to see.
- Please define the scale bar size exclusively in the figure legends and not in the image.
- In figure 1 the spaces are missing between the number and "min", e.g., 45min.
- Mouse work: please provide the reference number for approval in addition to the authority granting approval in the relevant methods section.
- The Data availability section should only refer to data deposited in public repositories. If your study has not produced novel datasets, please mention this fact in the Data Availability Section, e.g., by stating ""This study includes no data deposited in external repositories". Please remove the current statement: "All the data that support the findings of the study are available from the corresponding author upon reasonable request."
- As a standard, we modify the title and abstract of manuscripts to make them more accessible for our general readership. Please see and review my suggestions, which I pasted below my signature.

- Finally, EMBO Reports papers are accompanied online by

A) a short (1-2 sentences) summary of the findings and their significance,

B) 2-3 bullet points highlighting key results and

C) a schematic summary figure that provides a sketch of the major findings (not a data image).

Please provide the summary figure as a separate file in PNG or JPG format at a size of 550x300-600 pixels (width x height).

Please note that the size is rather small and that text needs to be readable at the final size. Please send us this information along with the revised manuscript.

With kind regards,

Martina Rembold, PhD

Senior Editor

EMBO reports

=====

Suggested title and abstract:

Cystinosin promotes Na⁺/H⁺ Exchanger 3 trafficking and function in kidney proximal tubular cells

Cystinosis is a systemic lysosomal storage disease resulting from mutations in the CTNS gene encoding the lysosomal cystine transporter cystinosin, leading to cystine accumulation in all organs. Despite cystinosin's ubiquitous expression, renal Fanconi syndrome (FS) is the first clinical manifestation of cystinosis, which is not prevented by cystine reduction therapy with cysteamine. Here, we report a novel interaction of cystinosin and sodium/hydrogen (Na⁺/H⁺) exchanger proteins in the endosomes of yeast and mammalian cells. NHE3 is a major absorptive sodium transporter at the apical membrane of proximal tubular cells (PTCs). Cystinosin is required for the correct subcellular localization and trafficking of NHE3 and for sodium uptake. Introducing CTNS successfully rescues these defects in CTNS- deficient PTCs, whereas CTNS-LKG, encoding the lysosomal and plasma membrane isoform of cystinosin, did not. NHE3 mislocalization was confirmed in Ctns^{-/-} mice and cystinosis patient kidney. Transplantation of wild-type hematopoietic stem and progenitor cells in Ctns^{-/-} mice restores NHE3 expression at the brush border membrane and improves FS-related phenotypes. This study uncovers an evolutionary conserved novel role of cystinosin in NHE3 trafficking, offering insights into FS pathogenesis and potential new therapeutic avenues.

=====

Referee #1:

While some mechanistic details remain unresolved, the authors have satisfactorily addressed all remaining issues.

Referee #2:

In this revised version of the manuscript, "Cystinosin is involved in Na⁺/H⁺ Exchanger 3 trafficking in the proximal tubular cells: new insights into the renal Fanconi syndrome in cystinosis", the authors have substantially improved the text and experimental framing and have provided reasonable responses and additional data addressing most of my initial concerns (e.g. interaction assays, transcriptional interpretation, and several mechanistic points). The overall narrative is now clearer and more coherent. Some issues with the imaging data still require attention, particularly in Figures 2B, 3C, EV3B, EV3C, and EV5B. The confocal panels would benefit from enhanced resolution, signal-to-noise ratio, and contrast, as it is currently difficult for the reader to appreciate the subcellular localization differences that underpin the main conclusions. Given that NHE3 trafficking is central to the study, the figures should more directly convey the reported effects. Improving image quality would also help clarify the relationship between the examples shown and the quantitative colocalization analysis. At present, values such as the Pearson coefficients suggest clear differences between conditions, whereas the corresponding images show rather diffuse staining in both control and CTNS-deficient models. This discrepancy makes it challenging for the reader to intuitively connect the visual data with the numerical results. Addressing these points would substantially improve the readability and interpretability of the imaging data and make the quantitative analyses more convincing. The rest of the experimental framework appears largely sound, and aligning the microscopy panels with the quantified results would strengthen the overall message of the manuscript. The central conceptual advance of this work remains compelling. The authors propose and support a model in which cystinosin regulates NHE3 trafficking in proximal tubular cells, thereby contributing to the pathophysiology of renal Fanconi syndrome in

cystinosis. This extends the role of cystinosin beyond lysosomal cystine transport and links lysosomal dysfunction to sodium handling at the brush border. The main limitation at this stage concerns the presentation of the imaging data supporting the trafficking and colocalization claims. Once the confocal figures are refined so that the visual evidence clearly reflects the quantitative analyses, the manuscript will more effectively communicate its strengths to the nephrology and cell biology communities.

In summary, I am supportive of the overall concept and believe the study has the potential to make a meaningful contribution to the field. However, in order to be qualitatively adequate for publication, I recommend final improvement of the the confocal imaging panels (Fig. 2B, 3C, EV3B, EV3C, EV5B) to ensure that image quality and interpretability are fully aligned with the quantitative analysis and the strength of the conclusions.

Referee #3:

The manuscript investigates dysregulated trafficking of NHE3 in proximal tubular cells from patients with cystinosis and explores its relationship with cystinosin, the lysosomal cystine transporter primarily affected in this disease. The authors propose that this aberrant trafficking impairs sodium and water reabsorption, thereby contributing to the proximal tubulopathy characteristic of Fanconi syndrome.

Overall, the manuscript is clearly written, and the authors employ appropriate and robust experimental approaches to support their conclusions.

The data reinforce the notion that effective treatment of Fanconi syndrome in cystinosis requires more than simply depleting lysosomal cystine. Instead, cystinosin-specific functions in proximal tubular cells must be restored, which provides mechanistic support for the rationale underlying the ongoing clinical trial.

The authors have adequately addressed my previous concerns regarding the functional relevance of their findings. The additional data on key Fanconi syndrome parameters in the mouse model, together with the demonstration of functional improvement following HSPC transplantation, substantially strengthen the translational value of the study and support the clinical applicability of the proposed therapeutic strategy.

Only a few typos should be corrected.

Referee #1:

While some mechanistic details remain unresolved, the authors have satisfactorily addressed all remaining issues.

Referee #2:

In this revised version of the manuscript, "Cystinosin is involved in Na⁺/H⁺ Exchanger 3 trafficking in the proximal tubular cells: new insights into the renal Fanconi syndrome in cystinosis", the authors have substantially improved the text and experimental framing and have provided reasonable responses and additional data addressing most of my initial concerns (e.g. interaction assays, transcriptional interpretation, and several mechanistic points). The overall narrative is now clearer and more coherent. Some issues with the imaging data still require attention, particularly in Figures 2B, 3C, EV3B, EV3C, and EV5B. The confocal panels would benefit from enhanced resolution, signal-to-noise ratio, and contrast, as it is currently difficult for the reader to appreciate the subcellular localization differences that underpin the main conclusions. Given that NHE3 trafficking is central to the study, the figures should more directly convey the reported effects.

Improving image quality would also help clarify the relationship between the examples shown and the quantitative colocalization analysis. At present, values such as the Pearson coefficients suggest clear differences between conditions, whereas the corresponding images show rather diffuse staining in both control and CTNS-deficient models. This discrepancy makes it challenging for the reader to intuitively connect the visual data with the numerical results.

Addressing these points would substantially improve the readability and interpretability of the imaging data and make the quantitative analyses more convincing. The rest of the experimental framework appears largely sound, and aligning the microscopy panels with the quantified results would strengthen the overall message of the manuscript.

The central conceptual advance of this work remains compelling. The authors propose and support a model in which cystinosin regulates NHE3 trafficking in proximal tubular cells, thereby contributing to the pathophysiology of renal Fanconi syndrome in cystinosis. This extends the role of cystinosin beyond lysosomal cystine transport and links lysosomal dysfunction to sodium handling at the brush border. The main limitation at this stage concerns the presentation of the imaging data supporting the

trafficking and colocalization claims. Once the confocal figures are refined so that the visual evidence clearly reflects the quantitative analyses, the manuscript will more effectively communicate its strengths to the nephrology and cell biology communities.

In summary, I am supportive of the overall concept and believe the study has the potential to make a meaningful contribution to the field. However, in order to be qualitatively adequate for publication, I recommend final improvement of the the confocal imaging panels (Fig. 2B, 3C, EV3B, EV3C, EV5B) to ensure that image quality and interpretability are fully aligned with the quantitative analysis and the strength of the conclusions.

Referee #3:

The manuscript investigates dysregulated trafficking of NHE3 in proximal tubular cells from patients with cystinosis and explores its relationship with cystinosin, the lysosomal cystine transporter primarily affected in this disease. The authors propose that this aberrant trafficking impairs sodium and water reabsorption, thereby contributing to the proximal tubulopathy characteristic of Fanconi syndrome.

Overall, the manuscript is clearly written, and the authors employ appropriate and robust experimental approaches to support their conclusions.

The data reinforce the notion that effective treatment of Fanconi syndrome in cystinosis requires more than simply depleting lysosomal cystine. Instead, cystinosin-specific functions in proximal tubular cells must be restored, which provides mechanistic support for the rationale underlying the ongoing clinical trial.

The authors have adequately addressed my previous concerns regarding the functional relevance of their findings. The additional data on key Fanconi syndrome parameters in the mouse model, together with the demonstration of functional improvement following HSPC transplantation, substantially strengthen the translational value of the study and support the clinical applicability of the proposed therapeutic strategy.

Only a few typos should be corrected.

Rev_Com_number: RC-2025-02907

New_manu_number: EMBOR-2025-62055V2

Corr_author: Cherqui

Title: Cystinosin is involved in Na⁺/H⁺ Exchanger 3 trafficking in the proximal tubular cells

Response to Reviewers:**Manuscript number:** EMBOR-2025-62055V2 [RC-2025-02907] [REV]**Corresponding author:** Stephanie, Cherqui

Editor Comments:

All referees are very positive about the study and request only minor changes to improve a subset of confocal imaging data.

Response: We sincerely thank the editor and the reviewers for the positive feedback.

Editorial Comments:

From the editorial side, there are also a few things that we need before we can proceed with the official acceptance of your study.

- Regarding the Author Contributions, we now use CRediT to specify the contributions of each author in the journal submission system. Therefore, please remove the Author Contributions from the manuscript file and make sure that the author contributions in our online manuscript tracking system are correct and up-to-date. The information you specified in the system will be automatically retrieved and typeset into the article. You can enter additional information in the free text box provided, if you wish. See also our guide to authors <https://www.embopress.org/page/journal/14693178/authorguide#authorshipguidelines>.

Response: The Author Contributions section has been removed from the manuscript file; the author contributions provided in the online manuscript tracking system are correct and up to date.

- The information on funding in the manuscript text and in the online manuscript tracking system must be congruent. In this regard we noted that the following items are missing in the system: NS108965, R01NS135162, R01AG086443, R01DK110162, P01HL152958; the MPS Society, and the Friedreich's Ataxia Research Alliance (FARA), P30-NS047101, S10OD030417.

Response: The funding information in the manuscript and the online manuscript tracking system is now congruent.

- The in-text callout to Supplementary movies 1-4 needs to be updated to Movies EV1-EV4.

Response: The in-text callout to Supplementary movies 1-4 has been updated to Movies EV1-EV4.

- Each movie needs to be provided with its legend (zipped up together) so that we have 4 zip folders; the correct nomenclature should be Movie EV1-Movie EV4 ("Supplementary movies 1-4" is not correct).

Response: Each movie has been provided along with its legend (zipped together), and four zip folders are included with the correct nomenclature (Movie EV1–Movie EV4).

- Please remove the Reagents and Tools table from the manuscript and upload it as a separate file.

Response: The Reagents and Tools table from the manuscript has been removed and uploaded as a separate file.

- Please remove the paragraph "Main findings of the study" from the manuscript. The text could be edited and used for the synopsis text/bullet points (see below).

Response: The paragraph "Main findings of the study" has been removed from the manuscript and we have added the following:

- A) a short (2 sentences) summary of the findings and their significance
- B) 3 bullet points highlighting key results

- We received bounced email alerts for the co-author Rola Chen - xic089@health.ucsd.edu. Please either remove the author from the author list in the system and then add her back using the correct/new email or send us the correct/new email and we will update the account accordingly.

Response: We have added the correct email ID for the author Rola Chen (xin.rola@outlook.com).

- Please address the following points in the figure legends and please highlight the changes:

Response: We have addressed all the points below by providing exact p values, specifying n numbers, identifying the statistical tests, error bars, and scale bars, and defining the arrows in all the relevant figure legends. All changes have been highlighted in red.

- Please provide the exact p values in the legends of figures 1e, f; 3a-h; 4b-d; 5b-h; 6; EV 3b-d; EV 4a-e; EV 5b; EV 6. (Exact p-values are not required for small values <0.0001).

Response: The exact p values have now been provided in the figure legends for Figs. 1E, F; 3A–H; 4B–D; 5B–H; 6; EV3B–D; EV4A–E; EV5B; and EV6 in the updated manuscript.

- Please indicate the statistical test used for data analysis in the legends of figures 1e, f

Response: The statistical test used for data analysis of figures 1E, F has been added to the figure legends in the updated manuscript.

- Please provide information related to n in the legends of figures 4b; 5b-h EV1d, f, g, h; EV3a-d; EV4a-e; EV5b; EV6

Response: The information related to n in the legends of figures 4B; 5B-H, EV1D, F, G, H; EV 3A-D; EV4A-E; EV5B; EV6 have been added to the figure legends in the updated manuscript.

- Please define the error bars in the legends of figures EV 1f, g, h

Response: The information related to the error bars in the legends of figures EV1F, G, H has been added to the figure legends in the updated manuscript.

- Please define the scale bar for figures EV 1c.

Response: We have defined the scale bar in the legend of the figures EV1C in the updated manuscript.

- Please note that the yellow and white arrows are not defined in the legend of figure 1a; EV 1c. This needs to be rectified.

Response: We have defined the yellow and white arrows in the legend of figures 1A and EV1C in the updated manuscript.

- The scale bars in Figure 2B, 3C, 5D, Fig 6, Fig EV3B, EV3C, EV5A, EV5B are difficult to see.

Response: The scale bars in Figs. 2B, 3C, 5D, Fig. 6, Fig. EV3B, EV3C, EV5A, EV5B has been enlarged for better visibility in the updated figures.

- Please define the scale bar size exclusively in the figure legends and not in the image.

Response: The scale bar size has now been defined exclusively in the figure legends and not in the image.

- In figure 1 the spaces are missing between the number and "min", e.g., 45min.

Response: The figure 1 has been corrected with spaces included between the number and min. The new figure has been provided.

- Mouse work: please provide the reference number for approval in addition to the authority granting approval in the relevant methods section.

Response: The reference number has been provided in the updated manuscript.

- The Data availability section should only refer to data deposited in public repositories. If your study has not produced novel datasets, please mention this fact in the Data Availability Section, e.g., by stating ""This study includes no data deposited in external repositories". Please remove the current statement: "All the data that support the findings of the study are available from the corresponding author upon reasonable request."

Response: We have included the statement “This study includes no data deposited in external repositories” in the data availability section of the revised manuscript.

- As a standard, we modify the title and abstract of manuscripts to make them more accessible for our general readership. Please see and review my suggestions, which I pasted below my signature.

Response: We thank the editor for the suggested modifications to the title and Abstract. We agree with the changes made to the Abstract. We have slightly modified the title, and the title is highlighted in red in the updated manuscript.

- Finally, EMBO Reports papers are accompanied online by
A) a short (1-2 sentences) summary of the findings and their significance,
B) 2-3 bullet points highlighting key results and
C) a schematic summary figure that provides a sketch of the major findings (not a data image). Please provide the summary figure as a separate file in PNG or JPG format at a size of 550x300-600 pixels (width x height). Please note that the size is rather small and that text needs to be readable at the final size. Please send us this information along with the revised manuscript.

Response: We have added the following in the updated manuscript:

- A) a short (1-2 sentences) summary of the findings and their significance
- B) 2-3 bullet points highlighting key results
- C) a schematic summary figure that provides a sketch of the major findings

Suggested title and abstract:

Cystinosin promotes Na⁺/H⁺ Exchanger 3 trafficking and function in kidney proximal tubular cells

Cystinosis is a systemic lysosomal storage disease resulting from mutations in the CTNS gene encoding the lysosomal cystine transporter cystinosin, leading to cystine accumulation in all organs. Despite cystinosin's ubiquitous expression, renal Fanconi syndrome (FS) is the first clinical manifestation of cystinosis, which is not prevented by cystine reduction therapy with cysteamine. Here, we report a novel interaction of cystinosin and sodium/hydrogen (Na⁺/H⁺) exchanger proteins in the endosomes of yeast and mammalian cells. NHE3 is a major absorptive sodium transporter at the apical membrane of proximal tubular cells (PTCs). Cystinosin is required for the correct subcellular localization and trafficking of NHE3 and for sodium uptake. Introducing CTNS successfully rescues these defects in CTNS- deficient PTCs, whereas CTNS-LKG, encoding the lysosomal and plasma membrane isoform of cystinosin, did not. NHE3 mislocalization was confirmed in Ctns^{-/-} mice and cystinosis patient kidney. Transplantation of wild-type hematopoietic stem and progenitor cells in Ctns^{-/-} mice restores NHE3 expression at the brush border membrane and improves FS-related phenotypes. This study uncovers an evolutionary conserved novel role of cystinosin in NHE3 trafficking, offering insights into FS

pathogenesis and potential new therapeutic avenues.

Response: We thank the editor for the suggested modifications to the title and Abstract. We agree with the changes made to the Abstract. We have slightly modified the title, and the title is highlighted in red in the updated manuscript.

=====

Referee #1:

While some mechanistic details remain unresolved, the authors have satisfactorily addressed all remaining issues.

Response: We are grateful to the reviewer for the constructive feedback.

Referee #2:

In this revised version of the manuscript, "Cystinosin is involved in Na⁺/H⁺ Exchanger 3 trafficking in the proximal tubular cells: new insights into the renal Fanconi syndrome in cystinosis", the authors have substantially improved the text and experimental framing and have provided reasonable responses and additional data addressing most of my initial concerns (e.g. interaction assays, transcriptional interpretation, and several mechanistic points). The overall narrative is now clearer and more coherent.

Response: We are grateful to the reviewer for the constructive feedback.

Some issues with the imaging data still require attention, particularly in Figures 2B, 3C, EV3B, EV3C, and EV5B. The confocal panels would benefit from enhanced resolution, signal-to-noise ratio, and contrast, as it is currently difficult for the reader to appreciate the subcellular localization differences that underpin the main conclusions. Given that NHE3 trafficking is central to the study, the figures should more directly convey the reported effects.

Improving image quality would also help clarify the relationship between the examples shown and the quantitative colocalization analysis. At present, values such as the Pearson coefficients suggest clear differences between conditions, whereas the corresponding images show rather diffuse staining in both control and CTNS-deficient models. This discrepancy makes it challenging for the reader to intuitively connect the visual data with the numerical results. Addressing these points would substantially improve the readability and interpretability of the imaging data and make the quantitative analyses more convincing. The rest of the experimental framework appears largely sound, and aligning the microscopy panels with the quantified results would strengthen the overall message of the manuscript.

The central conceptual advance of this work remains compelling. The authors propose and support a model in which cystinosin regulates NHE3 trafficking in proximal tubular cells, thereby contributing to the pathophysiology of renal Fanconi syndrome in cystinosis. This extends the role of cystinosin beyond lysosomal cystine transport and links lysosomal dysfunction to sodium handling at the brush border. The main limitation at this stage concerns

the presentation of the imaging data supporting the trafficking and colocalization claims. Once the confocal figures are refined so that the visual evidence clearly reflects the quantitative analyses, the manuscript will more effectively communicate its strengths to the nephrology and cell biology communities.

In summary, I am supportive of the overall concept and believe the study has the potential to make a meaningful contribution to the field. However, in order to be qualitatively adequate for publication, I recommend final improvement of the the confocal imaging panels (Fig. 2B, 3C, EV3B, EV3C, EV5B) to ensure that image quality and interpretability are fully aligned with the quantitative analysis and the strength of the conclusions.

Response: We appreciate the reviewer's comment. We have provided improved images for Figures 2B, 3C, EV3B, EV3C, and EV5B to enhance clarity in the updated figures.

Referee #3:

The manuscript investigates dysregulated trafficking of NHE3 in proximal tubular cells from patients with cystinosis and explores its relationship with cystinosin, the lysosomal cystine transporter primarily affected in this disease. The authors propose that this aberrant trafficking impairs sodium and water reabsorption, thereby contributing to the proximal tubulopathy characteristic of Fanconi syndrome.

Overall, the manuscript is clearly written, and the authors employ appropriate and robust experimental approaches to support their conclusions.

The data reinforce the notion that effective treatment of Fanconi syndrome in cystinosis requires more than simply depleting lysosomal cystine. Instead, cystinosin-specific functions in proximal tubular cells must be restored, which provides mechanistic support for the rationale underlying the ongoing clinical trial.

The authors have adequately addressed my previous concerns regarding the functional relevance of their findings. The additional data on key Fanconi syndrome parameters in the mouse model, together with the demonstration of functional improvement following HSPC transplantation, substantially strengthen the translational value of the study and support the clinical applicability of the proposed therapeutic strategy.

Only a few typos should be corrected.

Response: We thank the reviewer for the helpful comments. Typographical errors have been corrected to the best of our knowledge.

Dear Stephanie,

Thank you for the discussion and clarification of the imaging data shown in Figures 2B, 3C, and EV3. As noted by referee #1 and also by myself, the co-localization of NHE3 with CTNS or EEA1 is not intuitively visible in the images shown. You referred to published data showing a similar NHE3 localization as observed by you and clarified the imaging and quantification procedures. As discussed please add further clarification in the methods section and provide additional representative images in the source data. These additional images can be added to the same source data folders, e.g., Figure 2, panel B but should be clearly labeled as additional replicates. Please also provide a brief explanation as a point-by-point response.

Please note that our editorial assistants introduced a few changes to the current manuscript Word file in the system. They removed the synopsis text from the manuscript, the caption for the synopsis figure and the movie legends that were still part of the manuscript. Therefore, please use this file to introduce the discussed changes. Thank you very much.

I look forward to receiving your revised documents.

Kind regards,

Martina

Rev_Com_number: RC-2025-02907

New_manu_number: EMBOR-2025-62055V3

Corr_author: Cherqui

Title: Cystinosin regulates Na⁺/H⁺ Exchanger 3 trafficking and function in kidney proximal tubular cells

Response to Editor:**Manuscript number:** EMBOR-2025-62055V4 [RC-2025-02907] [REV]**Corresponding author:** Stephanie, Cherqui

Editor Comments:

Thank you for the discussion and clarification of the imaging data shown in Figures 2B, 3C, and EV3. As noted by referee #1 and also by myself, the co-localization of NHE3 with CTNS or EEA1 is not intuitively visible in the images shown. You referred to published data showing a similar NHE3 localization as observed by you and clarified the imaging and quantification procedures. As discussed please add further clarification in the methods section and provide additional representative images in the source data. These additional images can be added to the same source data folders, e.g., Figure 2, panel B but should be clearly labeled as additional replicates.

Response:

Thank you very much for your careful evaluation of the revised manuscript. In Figures 2B, 3C and EV3, we agree that visual inspection alone may not fully capture partial or transient co-localization events, particularly for proteins that dynamically traffic through endosomal compartments. To assess colocalization, we acquired single slice confocal images at the plane of cells displaying accurate spatial overlap between channels. Importantly, we have complemented the qualitative images with quantitative co-localization analysis (Manders' coefficients), which demonstrates a significant degree of EEA1–NHE3 overlap. We have elaborated the Methods section to include details and provided additional images in the Source Data file.

Dr. Stephanie Cherqui
University of California, San Diego
Department of Pediatrics
9500 Gilman Drive #0734
Israni Biomedical Research Facility II
La Jolla, CA 92093-0734
United States

Dear Stephanie,

Thank you for implementing the final revisions. I am very pleased to accept your manuscript for publication in the next available issue of EMBO reports. Thank you for your contribution to our journal.

You may qualify for financial assistance for your publication charges - either via a Springer Nature fully open access agreement or an EMBO initiative. Check your eligibility: <https://link.springer.com/journal/44319/how-to-publish-with-us>

Kind regards,

Martina

>>> Please note that it is EMBO Reports policy for the transcript of the editorial process (containing referee reports and your response letter) to be published as an online supplement to each paper. If you do NOT want this, you will need to inform the Editorial Office via email immediately. More information is available here: <https://link.springer.com/partners/embo-press/editorial-policies#Peer%20review>